# High Dimensional EM Algorithm:
# Statistical Optimization and Asymptotic Normality[*]

**Zhaoran Wang**
Princeton University

**Quanquan Gu**
University of Virginia

**Yang Ning**
Princeton University

**Han Liu**
Princeton University

## Abstract

We provide a general theory of the expectation-maximization (EM) algorithm for inferring high dimensional latent variable models. In particular, we make two contributions: (i) For parameter estimation, we propose a novel high dimensional EM algorithm which naturally incorporates sparsity structure into parameter estimation. With an appropriate initialization, this algorithm converges at a geometric rate and attains an estimator with the (near-)optimal statistical rate of convergence. (ii) Based on the obtained estimator, we propose a new inferential procedure for testing hypotheses for low dimensional components of high dimensional parameters. For a broad family of statistical models, our framework establishes the first computationally feasible approach for optimal estimation and asymptotic inference in high dimensions.

## 1 Introduction

The expectation-maximization (EM) algorithm [12] is the most popular approach for calculating the maximum likelihood estimator of latent variable models. Nevertheless, due to the nonconcavity of the likelihood function of latent variable models, the EM algorithm generally only converges to a local maximum rather than the global one [30]. On the other hand, existing statistical guarantees for latent variable models are only established for global optima [3]. Therefore, there exists a gap between computation and statistics.

Significant progress has been made toward closing the gap between the local maximum attained by the EM algorithm and the maximum likelihood estimator [2, 18, 25, 30]. In particular, [30] first establish general sufficient conditions for the convergence of the EM algorithm. [25] further improve this result by viewing the EM algorithm as a proximal point method applied to the Kullback-Leibler divergence. See [18] for a detailed survey. More recently, [2] establish the first result that characterizes explicit statistical and computational rates of convergence for the EM algorithm. They prove that, given a suitable initialization, the EM algorithm converges at a geometric rate to a local maximum close to the maximum likelihood estimator. All these results are established in the low dimensional regime where the dimension $d$ is much smaller than the sample size $n$.

In high dimensional regimes where the dimension $d$ is much larger than the sample size $n$, there exists no theoretical guarantee for the EM algorithm. In fact, when $d \gg n$, the maximum likelihood estimator is in general not well defined, unless the models are carefully regularized by sparsity-type assumptions. Furthermore, even if a regularized maximum likelihood estimator can be obtained in a computationally tractable manner, establishing the corresponding statistical properties, especially asymptotic normality, can still be challenging because of the existence of high dimensional nuisance parameters. To address such a challenge, we develop a general inferential theory of the EM algorithm for parameter estimation and uncertainty assessment of high dimensional latent variable models. In particular, we make two contributions in this paper:

- For high dimensional parameter estimation, we propose a novel high dimensional EM algorithm by attaching a truncation step to the expectation step (E-step) and maximization step (M-step). Such a

---

[*]Research supported by NSF IIS1116730, NSF IIS1332109, NSF IIS1408910, NSF IIS1546482-BIGDATA, NSF DMS1454377-CAREER, NIH R01GM083084, NIH R01HG06841, NIH R01MH102339, and FDA HHSF223201000072C.

truncation step effectively enforces the sparsity of the attained estimator and allows us to establish significantly improved statistical rate of convergence.

- Based upon the estimator attained by the high dimensional EM algorithm, we propose a decorrelated score statistic for testing hypotheses related to low dimensional components of the high dimensional parameter.

Under a unified analytic framework, we establish simultaneous statistical and computational guarantees for the proposed high dimensional EM algorithm and the respective uncertainty assessment procedure. Let $\boldsymbol{\beta}^* \in \mathbb{R}^d$ be the true parameter, $s^*$ be its sparsity level and $\{\boldsymbol{\beta}^{(t)}\}_{t=0}^T$ be the iterative solution sequence of the high dimensional EM algorithm with $T$ being the total number of iterations. In particular, we prove that:

- Given an appropriate initialization $\boldsymbol{\beta}^{\mathrm{init}}$ with relative error upper bounded by a constant $\kappa \in (0, 1)$, i.e., $\|\boldsymbol{\beta}^{\mathrm{init}} - \boldsymbol{\beta}^*\|_2 / \|\boldsymbol{\beta}^*\|_2 \leq \kappa$, the iterative solution sequence $\{\boldsymbol{\beta}^{(t)}\}_{t=0}^T$ satisfies

$$\left\|\boldsymbol{\beta}^{(t)} - \boldsymbol{\beta}^*\right\|_2 \leq \underbrace{\Delta_1 \cdot \rho^{t/2}}_{\text{Optimization Error}} + \underbrace{\Delta_2 \cdot \sqrt{s^* \cdot \log d / n}}_{\text{Statistical Error: Optimal Rate}} \tag{1.1}$$

with high probability. Here $\rho \in (0, 1)$, and $\Delta_1$, $\Delta_2$ are quantities that possibly depend on $\rho$, $\kappa$ and $\boldsymbol{\beta}^*$. As the optimization error term in (1.1) decreases to zero at a geometric rate with respect to $t$, the overall estimation error achieves the $\sqrt{s^* \cdot \log d / n}$ statistical rate of convergence (up to an extra factor of $\log n$), which is (near-)minimax-optimal. See Theorem 3.4 for details.

- The proposed decorrelated score statistic is asymptotically normal. Moreover, its limiting variance is optimal in the sense that it attains the semiparametric information bound for the low dimensional components of interest in the presence of high dimensional nuisance parameters. See Theorem 4.6 for details.

Our framework allows two implementations of the M-step: the exact maximization versus approximate maximization. The former one calculates the maximizer exactly, while the latter one conducts an approximate maximization through a gradient ascent step. Our framework is quite general. We illustrate its effectiveness by applying it to two high dimensional latent variable models, that is, Gaussian mixture model and mixture of regression model.

**Comparison with Related Work:** A closely related work is by [2], which considers the low dimensional regime where $d$ is much smaller than $n$. Under certain initialization conditions, they prove that the EM algorithm converges at a geometric rate to some local optimum that attains the $\sqrt{d/n}$ statistical rate of convergence. They cover both maximization and gradient ascent implementations of the M-step, and establish the consequences for the two latent variable models considered in our paper under low dimensional settings. Our framework adopts their view of treating the EM algorithm as a perturbed version of gradient methods. However, to handle the challenge of high dimensionality, the key ingredient of our framework is the truncation step that enforces the sparsity structure along the solution path. Such a truncation operation poses significant challenges for both computational and statistical analysis. In detail, for computational analysis we need to carefully characterize the evolution of each intermediate solution's support and its effects on the evolution of the entire iterative solution sequence. For statistical analysis, we need to establish a fine-grained characterization of the entrywise statistical error, which is technically more challenging than just establishing the $\ell_2$-norm error employed by [2]. In high dimensional regimes, we need to establish the $\sqrt{s^* \cdot \log d / n}$ statistical rate of convergence, which is much sharper than their $\sqrt{d/n}$ rate when $d \gg n$. In addition to point estimation, we further construct hypothesis tests for latent variable models in the high dimensional regime, which have not been established before.

High dimensionality poses significant challenges for assessing the uncertainty (e.g., testing hypotheses) of the constructed estimators. For example, [15] show that the limiting distribution of the Lasso estimator is not Gaussian even in the low dimensional regime. A variety of approaches have been proposed to correct the Lasso estimator to attain asymptotic normality, including the debiasing method [13], the desparsification methods [26, 32] as well as instrumental variable-based methods [4]. Meanwhile, [16, 17, 24] propose the post-selection procedures for exact inference. In addition, several authors propose methods based on data splitting [20, 29], stability selection [19] and $\ell_2$-confidence sets [22]. However, these approaches mainly focus on generalized linear models rather than latent variable models. In addition, their results heavily rely on the fact that the estimator is a global optimum of a convex program. In comparison, our approach applies to a much broader family of statistical models with latent structures. For these latent variable models, it is computationally infeasible to

obtain the global maximum of the penalized likelihood due to the nonconcavity of the likelihood function. Unlike existing approaches, our inferential theory is developed for the estimator attained by the proposed high dimensional EM algorithm, which is not necessarily a global optimum to any optimization formulation.

Another line of research for the estimation of latent variable models is the tensor method, which exploits the structures of third or higher order moments. See [1] and the references therein. However, existing tensor methods primarily focus on the low dimensional regime where $d \ll n$. In addition, since the high order sample moments generally have a slow statistical rate of convergence, the estimators obtained by the tensor methods usually have a suboptimal statistical rate even for $d \ll n$. For example, [9] establish the $\sqrt{d^6/n}$ statistical rate of convergence for mixture of regression model, which is suboptimal compared with the $\sqrt{d/n}$ minimax lower bound. Similarly, in high dimensional settings, the statistical rates of convergence attained by tensor methods are significantly slower than the statistical rate obtained in this paper.

The latent variable models considered in this paper have been well studied. Nevertheless, only a few works establish theoretical guarantees for the EM algorithm. In particular, for Gaussian mixture model, [10, 11] establish parameter estimation guarantees for the EM algorithm and its extensions. For mixture of regression model, [31] establish exact parameter recovery guarantees for the EM algorithm under a noiseless setting. For high dimensional mixture of regression model, [23] analyze the gradient EM algorithm for the $\ell_1$-penalized log-likelihood. They establish support recovery guarantees for the attained local optimum but have no parameter estimation guarantees. In comparison with existing works, this paper establishes a general inferential framework for simultaneous parameter estimation and uncertainty assessment based on a novel high dimensional EM algorithm. Our analysis provides the first theoretical guarantee of parameter estimation and asymptotic inference in high dimensional regimes for the EM algorithm and its applications to a broad family of latent variable models.

**Notation:** The matrix $(p, q)$-norm, i.e., $\| \cdot \|_{p,q}$, is obtained by taking the $\ell_p$-norm of each row and then taking the $\ell_q$-norm of the obtained row norms. We use $C, C', \dots$ to denote generic constants. Their values may vary from line to line. We will introduce more notations in §2.2.

## 2 Methodology

We first introduce the high dimensional EM Algorithm and then the respective inferential procedure. As examples, we consider their applications to Gaussian mixture model and mixture of regression model. For compactness, we defer the details to §A of the appendix. More models are included in the longer version of this paper.

---

**Algorithm 1** High Dimensional EM Algorithm

---

1: **Parameter:** Sparsity Parameter $\widehat{s}$, Maximum Number of Iterations $T$
2: **Initialization:** $\widehat{\mathcal{S}}^{\text{init}} \leftarrow \text{supp}(\boldsymbol{\beta}^{\text{init}}, \widehat{s})$, $\quad \boldsymbol{\beta}^{(0)} \leftarrow \text{trunc}(\boldsymbol{\beta}^{\text{init}}, \widehat{\mathcal{S}}^{\text{init}})$
   $\{\text{supp}(\cdot, \cdot) \text{ and } \text{trunc}(\cdot, \cdot) \text{ are defined in } (2.2) \text{ and } (2.3)\}$
3: **For** $t = 0$ to $T - 1$
4: $\quad$ **E-step:** Evaluate $Q_n(\boldsymbol{\beta}; \boldsymbol{\beta}^{(t)})$
5: $\quad$ **M-step:** $\boldsymbol{\beta}^{(t+0.5)} \leftarrow M_n(\boldsymbol{\beta}^{(t)})$ $\quad \{M_n(\cdot) \text{ is implemented as in Algorithm 2 or 3}\}$
6: $\quad$ **T-step:** $\widehat{\mathcal{S}}^{(t+0.5)} \leftarrow \text{supp}(\boldsymbol{\beta}^{(t+0.5)}, \widehat{s})$, $\quad \boldsymbol{\beta}^{(t+1)} \leftarrow \text{trunc}(\boldsymbol{\beta}^{(t+0.5)}, \widehat{\mathcal{S}}^{(t+0.5)})$
7: **End For**
8: **Output:** $\widehat{\boldsymbol{\beta}} \leftarrow \boldsymbol{\beta}^{(T)}$

---

**Algorithm 2** Maximization Implementation of the M-step

---

1: **Input:** $\boldsymbol{\beta}^{(t)}, Q_n(\boldsymbol{\beta}; \boldsymbol{\beta}^{(t)})$ $\quad$ **Output:** $M_n(\boldsymbol{\beta}^{(t)}) \leftarrow \text{argmax}_{\boldsymbol{\beta}} Q_n(\boldsymbol{\beta}; \boldsymbol{\beta}^{(t)})$

---

**Algorithm 3** Gradient Ascent Implementation of the M-step

---

1: **Input:** $\boldsymbol{\beta}^{(t)}, Q_n(\boldsymbol{\beta}; \boldsymbol{\beta}^{(t)})$ $\quad$ **Parameter:** Stepsize $\eta > 0$
2: **Output:** $M_n(\boldsymbol{\beta}^{(t)}) \leftarrow \boldsymbol{\beta}^{(t)} + \eta \cdot \nabla Q_n(\boldsymbol{\beta}^{(t)}; \boldsymbol{\beta}^{(t)})$

---

### 2.1 High Dimensional EM Algorithm

Before we introduce the proposed high dimensional EM Algorithm (Algorithm 1), we briefly review the classical EM algorithm. Let $h_{\boldsymbol{\beta}}(\mathbf{y})$ be the probability density function of $\boldsymbol{Y} \in \mathcal{Y}$, where $\boldsymbol{\beta} \in \mathbb{R}^d$ is the model parameter. For latent variable models, we assume that $h_{\boldsymbol{\beta}}(\mathbf{y})$ is obtained by marginalizing over an unobserved latent variable $\boldsymbol{Z} \in \mathcal{Z}$, i.e., $h_{\boldsymbol{\beta}}(\mathbf{y}) = \int_{\mathcal{Z}} f_{\boldsymbol{\beta}}(\mathbf{y}, \mathbf{z}) \, \text{d}\mathbf{z}$. Let $k_{\boldsymbol{\beta}}(\mathbf{z} \mid \mathbf{y})$ be the density

of $Z$ conditioning on the observed variable $\boldsymbol{Y} = \mathbf{y}$, i.e., $k_{\boldsymbol{\beta}}(\mathbf{z} \mid \mathbf{y}) = f_{\boldsymbol{\beta}}(\mathbf{y}, \mathbf{z})/h_{\boldsymbol{\beta}}(\mathbf{y})$. We define

$$Q_n(\boldsymbol{\beta}; \boldsymbol{\beta}') = \frac{1}{n} \sum_{i=1}^{n} \int_{\mathcal{Z}} k_{\boldsymbol{\beta}'}(\mathbf{z} \mid \mathbf{y}_i) \cdot \log f_{\boldsymbol{\beta}}(\mathbf{y}_i, \mathbf{z}) \, \mathrm{d}\mathbf{z}. \tag{2.1}$$

See §B of the appendix for a detailed derivation. At the $t$-th iteration of the classical EM algorithm, we evaluate $Q_n(\boldsymbol{\beta}; \boldsymbol{\beta}^{(t)})$ at the E-step and then perform $\max_{\boldsymbol{\beta}} Q_n(\boldsymbol{\beta}; \boldsymbol{\beta}^{(t)})$ at the M-step. The proposed high dimensional EM algorithm (Algorithm 1) is built upon the E-step and M-step (lines 4 and 5) of the classical EM algorithm. In addition to the exact maximization implementation of the M-step (Algorithm 2), we allow the gradient ascent implementation of the M-step (Algorithm 3), which performs an approximate maximization via a gradient ascent step. To handle the challenge of high dimensionality, in line 6 of Algorithm 1 we perform a truncation step (T-step) to enforce the sparsity structure. In detail, we define

$$\text{supp}(\boldsymbol{\beta}, s): \text{The set of index } j\text{'s corresponding to the top } s \text{ largest } |\beta_j|\text{'s.} \tag{2.2}$$

Also, for an index set $\mathcal{S} \subseteq \{1, \ldots, d\}$, we define the $\text{trunc}(\cdot, \cdot)$ function in line 6 as

$$\left[\text{trunc}(\boldsymbol{\beta}, \mathcal{S})\right]_j = \beta_j \cdot \mathbb{1}\{j \in \mathcal{S}\}. \tag{2.3}$$

Note that $\boldsymbol{\beta}^{(t+0.5)}$ is the output of the M-step (line 5) at the $t$-th iteration of the high dimensional EM algorithm. To obtain $\boldsymbol{\beta}^{(t+1)}$, the T-step (line 6) preserves the entries of $\boldsymbol{\beta}^{(t+0.5)}$ with the top $\widehat{s}$ large magnitudes and sets the rest to zero. Here $\widehat{s}$ is a tuning parameter that controls the sparsity level (line 1). By iteratively performing the E-step, M-step and T-step, the high dimensional EM algorithm attains an $\widehat{s}$-sparse estimator $\widehat{\boldsymbol{\beta}} = \boldsymbol{\beta}^{(T)}$ (line 8). Here $T$ is the total number of iterations.

## 2.2 Asymptotic Inference

**Notation:** Let $\nabla_1 Q(\boldsymbol{\beta}; \boldsymbol{\beta}')$ be the gradient with respect to $\boldsymbol{\beta}$ and $\nabla_2 Q(\boldsymbol{\beta}; \boldsymbol{\beta}')$ be the gradient with respect to $\boldsymbol{\beta}'$. If there is no confusion, we simply denote $\nabla Q(\boldsymbol{\beta}; \boldsymbol{\beta}') = \nabla_1 Q(\boldsymbol{\beta}; \boldsymbol{\beta}')$ as in the previous sections. We define the higher order derivatives in the same manner, e.g., $\nabla_{1,2}^2 Q(\boldsymbol{\beta}; \boldsymbol{\beta}')$ is calculated by first taking derivative with respect to $\boldsymbol{\beta}$ and then with respect to $\boldsymbol{\beta}'$. For $\boldsymbol{\beta} = (\boldsymbol{\beta}_1^\top, \boldsymbol{\beta}_2^\top)^\top \in \mathbb{R}^d$ with $\boldsymbol{\beta}_1 \in \mathbb{R}^{d_1}$, $\boldsymbol{\beta}_2 \in \mathbb{R}^{d_2}$ and $d_1 + d_2 = d$, we use notations such as $\mathbf{v}_{\boldsymbol{\beta}_1} \in \mathbb{R}^{d_1}$ and $\mathbf{A}_{\boldsymbol{\beta}_1, \boldsymbol{\beta}_2} \in \mathbb{R}^{d_1 \times d_2}$ to denote the corresponding subvector of $\mathbf{v} \in \mathbb{R}^d$ and the submatrix of $\mathbf{A} \in \mathbb{R}^{d \times d}$.

We aim to conduct asymptotic inference for low dimensional components of the high dimensional parameter $\boldsymbol{\beta}^*$. Without loss of generality, we consider a single entry of $\boldsymbol{\beta}^*$. In particular, we assume $\boldsymbol{\beta}^* = [\alpha^*, (\boldsymbol{\gamma}^*)^\top]^\top$, where $\alpha^* \in \mathbb{R}$ is the entry of interest, while $\boldsymbol{\gamma}^* \in \mathbb{R}^{d-1}$ is treated as the nuisance parameter. In the following, we construct a high dimensional score test named decorrelated score test. It is worth noting that, our method and theory can be easily generalized to perform statistical inference for an arbitrary low dimensional subvector of $\boldsymbol{\beta}^*$.

**Decorrelated Score Test:** For score test, we are primarily interested in testing $H_0 : \alpha^* = 0$, since this null hypothesis characterizes the uncertainty in variable selection. Our method easily generalizes to $H_0 : \alpha^* = \alpha_0$ with $\alpha_0 \neq 0$. For notational simplicity, we define the following key quantity

$$T_n(\boldsymbol{\beta}) = \nabla_{1,1}^2 Q_n(\boldsymbol{\beta}; \boldsymbol{\beta}) + \nabla_{1,2}^2 Q_n(\boldsymbol{\beta}; \boldsymbol{\beta}) \in \mathbb{R}^{d \times d}. \tag{2.4}$$

Let $\boldsymbol{\beta} = (\alpha, \boldsymbol{\gamma}^\top)^\top$. We define the decorrelated score function $S_n(\cdot, \cdot) \in \mathbb{R}$ as

$$S_n(\boldsymbol{\beta}, \lambda) = \left[\nabla_1 Q_n(\boldsymbol{\beta}; \boldsymbol{\beta})\right]_\alpha - w(\boldsymbol{\beta}, \lambda)^\top \cdot \left[\nabla_1 Q_n(\boldsymbol{\beta}; \boldsymbol{\beta})\right]_{\boldsymbol{\gamma}}. \tag{2.5}$$

Here $w(\boldsymbol{\beta}, \lambda) \in \mathbb{R}^{d-1}$ is obtained using the following Dantzig selector [8]

$$w(\boldsymbol{\beta}, \lambda) = \underset{\mathbf{w} \in \mathbb{R}^{d-1}}{\text{argmin}} \|\mathbf{w}\|_1, \quad \text{subject to } \left\| \left[T_n(\boldsymbol{\beta})\right]_{\boldsymbol{\gamma}, \alpha} - \left[T_n(\boldsymbol{\beta})\right]_{\boldsymbol{\gamma}, \boldsymbol{\gamma}} \cdot \mathbf{w} \right\|_\infty \leq \lambda, \tag{2.6}$$

where $\lambda > 0$ is a tuning parameter. Let $\widehat{\boldsymbol{\beta}} = (\widehat{\alpha}, \widehat{\boldsymbol{\gamma}}^\top)^\top$, where $\widehat{\boldsymbol{\beta}}$ is the estimator attained by the high dimensional EM algorithm (Algorithm 1). We define the decorrelated score statistic as

$$\sqrt{n} \cdot S_n(\widehat{\boldsymbol{\beta}}_0, \lambda) / \left\{ -\left[T_n(\widehat{\boldsymbol{\beta}}_0)\right]_{\alpha|\boldsymbol{\gamma}} \right\}^{1/2}, \tag{2.7}$$

where $\widehat{\boldsymbol{\beta}}_0 = (0, \widehat{\boldsymbol{\gamma}}^\top)^\top$, and $\left[T_n(\widehat{\boldsymbol{\beta}}_0)\right]_{\alpha|\boldsymbol{\gamma}} = \left[1, -w(\widehat{\boldsymbol{\beta}}_0, \lambda)^\top\right] \cdot T_n(\widehat{\boldsymbol{\beta}}_0) \cdot \left[1, -w(\widehat{\boldsymbol{\beta}}_0, \lambda)^\top\right]^\top$.

Here we use $\widehat{\boldsymbol{\beta}}_0$ instead of $\widehat{\boldsymbol{\beta}}$ since we are interested in the null hypothesis $H_0 : \alpha^* = 0$. We can also replace $\widehat{\boldsymbol{\beta}}_0$ with $\widehat{\boldsymbol{\beta}}$ and the theoretical results will remain the same. In §4 we will prove the proposed decorrelated score statistic in (2.7) is asymptotically $N(0, 1)$. Consequently, the decorrelated score

test with significance level $\delta \in (0,1)$ takes the form

$$\psi_S(\delta) = \mathbb{1}\big\{\sqrt{n} \cdot S_n\big(\widehat{\boldsymbol{\beta}}_0, \lambda\big)\big/\big\{-\big[T_n\big(\widehat{\boldsymbol{\beta}}_0\big)\big]_{\alpha|\gamma}\big\}^{1/2} \notin \big[-\Phi^{-1}(1 - \delta/2), \Phi^{-1}(1 - \delta/2)\big]\big\},$$

where $\Phi^{-1}(\cdot)$ is the inverse function of the Gaussian cumulative distribution function. If $\psi_S(\delta) = 1$, we reject the null hypothesis $H_0 : \alpha^* = 0$. The intuition of this decorrelated score test is explained in §D of the appendix. The key theoretical observation is Theorem 2.1, which connects $\nabla_1 Q_n(\cdot; \cdot)$ in (2.5) and $T_n(\cdot)$ in (2.7) with the score function and Fisher information in the presence of latent structures. Let $\ell_n(\boldsymbol{\beta})$ be the log-likelihood. Its score function is $\nabla \ell_n(\boldsymbol{\beta})$ and the Fisher information is $I(\boldsymbol{\beta}^*) = -\mathbb{E}_{\boldsymbol{\beta}^*}\big[\nabla^2 \ell_n(\boldsymbol{\beta}^*)\big]\big/n$, where $\mathbb{E}_{\boldsymbol{\beta}^*}(\cdot)$ is the expectation under the model with parameter $\boldsymbol{\beta}^*$.

**Theorem 2.1.** For the true parameter $\boldsymbol{\beta}^*$ and any $\boldsymbol{\beta} \in \mathbb{R}^d$, it holds that

$$\nabla_1 Q_n(\boldsymbol{\beta}; \boldsymbol{\beta}) = \nabla \ell_n(\boldsymbol{\beta})/n, \quad \text{and} \ \ \mathbb{E}_{\boldsymbol{\beta}^*}\big[T_n(\boldsymbol{\beta}^*)\big] = -I(\boldsymbol{\beta}^*) = \mathbb{E}_{\boldsymbol{\beta}^*}\big[\nabla^2 \ell_n(\boldsymbol{\beta}^*)\big]\big/n. \quad (2.8)$$

*Proof.* See §I.1 of the appendix for a detailed proof. $\qquad\qquad\qquad\qquad\qquad \square$

Based on the decorrelated score test, it is easy to establish the decorrelated Wald test, which allows us to construct confidence intervals. For compactness we defer it to the longer version of this paper.

# 3 Theory of Computation and Estimation

Before we present the main results, we introduce three technical conditions, which will significantly ease our presentation. They will be verified for specific latent variable models in §E of the appendix. The first two conditions, proposed by [2], characterize the properties of the population version lower bound function $Q(\cdot; \cdot)$, i.e., the expectation of $Q_n(\cdot; \cdot)$ defined in (2.1). We define the respective population version M-step as follows. For the M-step in Algorithm 2, we define

$$M(\boldsymbol{\beta}) = \operatorname*{argmax}_{\boldsymbol{\beta}'} Q(\boldsymbol{\beta}'; \boldsymbol{\beta}). \quad (3.1)$$

For the M-step in Algorithm 3, we define

$$M(\boldsymbol{\beta}) = \boldsymbol{\beta} + \eta \cdot \nabla_1 Q(\boldsymbol{\beta}; \boldsymbol{\beta}), \quad (3.2)$$

where $\eta > 0$ is the stepsize in Algorithm 3. We use $\mathcal{B}$ to denote the basin of attraction, i.e., the local region where the high dimensional EM algorithm enjoys desired guarantees.

**Condition 3.1.** We define two versions of this condition.

- *Lipschitz-Gradient*-$1(\gamma_1, \mathcal{B})$. For the true parameter $\boldsymbol{\beta}^*$ and any $\boldsymbol{\beta} \in \mathcal{B}$, we have

$$\big\|\nabla_1 Q\big[M(\boldsymbol{\beta}); \boldsymbol{\beta}^*\big] - \nabla_1 Q\big[M(\boldsymbol{\beta}); \boldsymbol{\beta}\big]\big\|_2 \leq \gamma_1 \cdot \|\boldsymbol{\beta} - \boldsymbol{\beta}^*\|_2, \quad (3.3)$$

  where $M(\cdot)$ is the population version M-step (maximization implementation) defined in (3.1).

- *Lipschitz-Gradient*-$2(\gamma_2, \mathcal{B})$. For the true parameter $\boldsymbol{\beta}^*$ and any $\boldsymbol{\beta} \in \mathcal{B}$, we have

$$\big\|\nabla_1 Q(\boldsymbol{\beta}; \boldsymbol{\beta}^*) - \nabla_1 Q(\boldsymbol{\beta}; \boldsymbol{\beta})\big\|_2 \leq \gamma_2 \cdot \|\boldsymbol{\beta} - \boldsymbol{\beta}^*\|_2. \quad (3.4)$$

Condition 3.1 defines a variant of Lipschitz continuity for $\nabla_1 Q(\cdot; \cdot)$. In the sequel, we will use (3.3) and (3.4) in the analysis of the two implementations of the M-step respectively.

**Condition 3.2** *Concavity-Smoothness*$(\mu, \nu, \mathcal{B})$. For any $\boldsymbol{\beta}_1, \boldsymbol{\beta}_2 \in \mathcal{B}$, $Q(\cdot; \boldsymbol{\beta}^*)$ is $\mu$-smooth, i.e.,

$$Q(\boldsymbol{\beta}_1; \boldsymbol{\beta}^*) \geq Q(\boldsymbol{\beta}_2; \boldsymbol{\beta}^*) + (\boldsymbol{\beta}_1 - \boldsymbol{\beta}_2)^\top \cdot \nabla_1 Q(\boldsymbol{\beta}_2; \boldsymbol{\beta}^*) - \mu/2 \cdot \|\boldsymbol{\beta}_2 - \boldsymbol{\beta}_1\|_2^2, \quad (3.5)$$

and $\nu$-strongly concave, i.e.,

$$Q(\boldsymbol{\beta}_1; \boldsymbol{\beta}^*) \leq Q(\boldsymbol{\beta}_2; \boldsymbol{\beta}^*) + (\boldsymbol{\beta}_1 - \boldsymbol{\beta}_2)^\top \cdot \nabla_1 Q(\boldsymbol{\beta}_2; \boldsymbol{\beta}^*) - \nu/2 \cdot \|\boldsymbol{\beta}_2 - \boldsymbol{\beta}_1\|_2^2. \quad (3.6)$$

This condition indicates that, when the second variable of $Q(\cdot; \cdot)$ is fixed to be $\boldsymbol{\beta}^*$, the function is 'sandwiched' between two quadratic functions. The third condition characterizes the statistical error between the sample version and population version M-steps, i.e., $M_n(\cdot)$ defined in Algorithms 2 and 3, and $M(\cdot)$ in (3.1) and (3.2). Recall $\|\cdot\|_0$ denotes the total number of nonzero entries in a vector.

**Condition 3.3** *Statistical-Error*$(\epsilon, \delta, s, n, \mathcal{B})$. For any fixed $\boldsymbol{\beta} \in \mathcal{B}$ with $\|\boldsymbol{\beta}\|_0 \leq s$, we have that

$$\big\|M(\boldsymbol{\beta}) - M_n(\boldsymbol{\beta})\big\|_\infty \leq \epsilon \quad (3.7)$$

holds with probability at least $1 - \delta$. Here $\epsilon > 0$ possibly depends on $\delta$, sparsity level $s$, sample size $n$, dimension $d$, as well as the basin of attraction $\mathcal{B}$.

In (3.7) the statistical error $\epsilon$ quantifies the $\ell_\infty$-norm of the difference between the population version and sample version M-steps. Particularly, we constrain the input $\boldsymbol{\beta}$ of $M(\cdot)$ and $M_n(\cdot)$ to be $s$-sparse. Such a condition is different from the one used by [2]. In detail, they quantify the statistical error

with the $\ell_2$-norm and do not constrain the input of $M(\cdot)$ and $M_n(\cdot)$ to be sparse. Consequently, our subsequent statistical analysis is different from theirs. The reason we use the $\ell_\infty$-norm is that, it characterizes the more refined entrywise statistical error, which converges at a fast rate of $\sqrt{\log d/n}$ (possibly with extra factors depending on specific models). In comparison, the $\ell_2$-norm statistical error converges at a slow rate of $\sqrt{d/n}$, which does not decrease to zero as $n$ increases with $d \gg n$. Furthermore, the fine-grained entrywise statistical error is crucial to our key proof for quantifying the effects of the truncation step (line 6 of Algorithm 1) on the iterative solution sequence.

## 3.1 Main Results

To simplify the technical analysis of the high dimensional EM algorithm, we focus on its resampling version, which is illustrated in Algorithm 4 in §C of the appendix.

**Theorem 3.4.** We define $\mathcal{B} = \{\boldsymbol{\beta} : \|\boldsymbol{\beta} - \boldsymbol{\beta}^*\|_2 \le R\}$, where $R = \kappa \cdot \|\boldsymbol{\beta}^*\|_2$ for some $\kappa \in (0, 1)$. We assume Condition *Concavity-Smoothness*$(\mu, \nu, \mathcal{B})$ holds and $\|\boldsymbol{\beta}^{\mathrm{init}} - \boldsymbol{\beta}^*\|_2 \le R/2$.

- For the maximization implementation of the M-step (Algorithm 2), we suppose that Condition *Lipschitz-Gradient*-1$(\gamma_1, \mathcal{B})$ holds with $\rho_1 := \gamma_1/\nu \in (0, 1)$ and

$$\widehat{s} = \lceil C \cdot \max \{16/(1/\rho_1 - 1)^2, \ 4 \cdot (1+\kappa)^2/(1-\kappa)^2\} \cdot s^* \rceil, \tag{3.8}$$

$$(\sqrt{\widehat{s}} + C'/\sqrt{1-\kappa} \cdot \sqrt{s^*}) \cdot \epsilon \le \min\{(1 - \sqrt{\rho_1})^2 \cdot R, \ (1-\kappa)^2/[2 \cdot (1+\kappa)] \cdot \|\boldsymbol{\beta}^*\|_2\}. \tag{3.9}$$

  Here $C \ge 1$ and $C' > 0$ are constants. Under Condition *Statistical-Error*$(\epsilon, \delta/T, \widehat{s}, n/T, \mathcal{B})$ we have that, for $t = 1, \ldots, T$,

$$\|\boldsymbol{\beta}^{(t)} - \boldsymbol{\beta}^*\|_2 \le \underbrace{\rho_1^{t/2} \cdot R}_{\text{Optimization Error}} + \underbrace{(\sqrt{\widehat{s}} + C'/\sqrt{1-\kappa} \cdot \sqrt{s^*})/(1 - \sqrt{\rho_1}) \cdot \epsilon}_{\text{Statistical Error}} \tag{3.10}$$

  holds with probability at least $1 - \delta$, where $C'$ is the same constant as in (3.9).

- For the gradient ascent implementation of the M-step (Algorithm 3), we suppose that Condition *Lipschitz-Gradient*-2$(\gamma_2, \mathcal{B})$ holds with $\rho_2 := 1 - 2 \cdot (\nu - \gamma_2)/(\nu + \mu) \in (0, 1)$ and the stepsize in Algorithm 3 is set to $\eta = 2/(\nu + \mu)$. Meanwhile, we assume (3.8) and (3.9) hold with $\rho_1$ replaced by $\rho_2$. Under Condition *Statistical-Error*$(\epsilon, \delta/T, \widehat{s}, n/T, \mathcal{B})$ we have that, for $t = 1, \ldots, T$, (3.10) holds with probability at least $1 - \delta$, in which $\rho_1$ is replaced with $\rho_2$.

*Proof.* See §G.1 of the appendix for a detailed proof. $\square$

The assumption in (3.8) states that the sparsity parameter $\widehat{s}$ is chosen to be sufficiently large and also of the same order as the true sparsity level $s^*$. This assumption ensures that the error incurred by the truncation step can be upper bounded. In addition, as is shown for specific latent variable models in §E of the appendix, the error term $\epsilon$ in Condition *Statistical-Error*$(\epsilon, \delta/T, \widehat{s}, n/T, \mathcal{B})$ decreases as sample size $n$ increases. By the assumption in (3.8), $(\sqrt{\widehat{s}} + C'/\sqrt{1-\kappa} \cdot \sqrt{s^*})$ is of the same order as $\sqrt{s^*}$. Therefore, the assumption in (3.9) suggests the sample size $n$ is sufficiently large such that $\sqrt{s^*} \cdot \epsilon$ is sufficiently small. These assumptions guarantee that the entire iterative solution sequence remains within the basin of attraction $\mathcal{B}$ in the presence of statistical error.

Theorem 3.4 illustrates that, the upper bound of the overall estimation error can be decomposed into two terms. The first term is the upper bound of optimization error, which decreases to zero at a geometric rate of convergence, because we have $\rho_1, \rho_2 < 1$. Meanwhile, the second term is the upper bound of statistical error, which does not depend on $t$. Since $(\sqrt{\widehat{s}} + C'/\sqrt{1-\kappa} \cdot \sqrt{s^*})$ is of the same order as $\sqrt{s^*}$, this term is proportional to $\sqrt{s^*} \cdot \epsilon$, where $\epsilon$ is the entrywise statistical error between $M(\cdot)$ and $M_n(\cdot)$. In §E of the appendix we prove that, for each specific latent variable model, $\epsilon$ is roughly of the order $\sqrt{\log d/n}$. (There may be extra factors attached to $\epsilon$ depending on each specific model.) Therefore, the statistical error term is roughly of the order $\sqrt{s^* \cdot \log d/n}$. Consequently, for a sufficiently large $t = T$ such that the optimization and statistical error terms in (3.10) are of the same order, the final estimator $\widehat{\boldsymbol{\beta}} = \boldsymbol{\beta}^{(T)}$ attains a (near-)optimal $\sqrt{s^* \cdot \log d/n}$ (possibly with extra factors) statistical rate. For compactness, we give the following example and defer the details to §E.

**Implications for Gaussian Mixture Model:** We assume $\mathbf{y}_1, \ldots, \mathbf{y}_n$ are the $n$ i.i.d. realizations of $\boldsymbol{Y} = Z \cdot \boldsymbol{\beta}^* + \boldsymbol{V}$. Here $Z$ is a Rademacher random variable, i.e., $\mathbb{P}(Z = +1) = \mathbb{P}(Z = -1) = 1/2$, and $\boldsymbol{V} \sim N(\mathbf{0}, \sigma^2 \cdot \mathbf{I}_d)$ is independent of $Z$, where $\sigma$ is the standard deviation. Suppose that we have $\|\boldsymbol{\beta}^*\|_2/\sigma \ge r$, where $r > 0$ is a sufficiently large constant that denotes the minimum signal-to-noise ratio. In §E of the appendix we prove that there exists some constant $C > 0$ such that Conditions

*Lipschitz-Gradient*-$1(\gamma_1, \mathcal{B})$ and *Concavity-Smoothness*$(\mu, \nu, \mathcal{B})$ hold with

$$\gamma_1 = \exp(-C \cdot r^2), \quad \mu = \nu = 1, \quad \mathcal{B} = \{\boldsymbol{\beta} : \|\boldsymbol{\beta} - \boldsymbol{\beta}^*\|_2 \leq R\} \text{ with } R = \kappa \cdot \|\boldsymbol{\beta}^*\|_2, \; \kappa = 1/4.$$

For a sufficiently large $n$, we have that Condition *Statistical-Error*$(\epsilon, \delta, s, n, \mathcal{B})$ holds with

$$\epsilon = C \cdot \left(\|\boldsymbol{\beta}^*\|_\infty + \sigma\right) \cdot \sqrt{\left[\log d + \log(2/\delta)\right]/n}.$$

Then the first part of Theorem 3.4 implies $\|\widehat{\boldsymbol{\beta}} - \boldsymbol{\beta}^*\|_2 \leq C \cdot \sqrt{s^* \cdot \log d \cdot \log n / n}$ for a sufficiently large $T$, which is near-optimal with respect to the minimax lower bound $\sqrt{s^* \log d / n}$.

# 4 Theory of Inference

To simplify the presentation of the unified framework, we lay out several technical conditions, which will be verified for each model. Let $\zeta^{\mathrm{EM}}$, $\zeta^{\mathrm{G}}$, $\zeta^{\mathrm{T}}$ and $\zeta^{\mathrm{L}}$ be four quantities that scale with $s^*$, $d$ and $n$. These conditions will be verified for specific latent variable models in §F of the appendix.

**Condition 4.1** *Parameter-Estimation*$(\zeta^{\mathrm{EM}})$. We have $\|\widehat{\boldsymbol{\beta}} - \boldsymbol{\beta}^*\|_1 = O_{\mathbb{P}}(\zeta^{\mathrm{EM}})$.

**Condition 4.2** *Gradient-Statistical-Error*$(\zeta^{\mathrm{G}})$. We have

$$\left\|\nabla_1 Q_n(\boldsymbol{\beta}^*; \boldsymbol{\beta}^*) - \nabla_1 Q(\boldsymbol{\beta}^*; \boldsymbol{\beta}^*)\right\|_\infty = O_{\mathbb{P}}(\zeta^{\mathrm{G}}).$$

**Condition 4.3** $T_n(\cdot)$-*Concentration*$(\zeta^{\mathrm{T}})$. We have $\left\|T_n(\boldsymbol{\beta}^*) - \mathbb{E}_{\boldsymbol{\beta}^*}\left[T_n(\boldsymbol{\beta}^*)\right]\right\|_{\infty,\infty} = O_{\mathbb{P}}(\zeta^{\mathrm{T}})$.

**Condition 4.4** $T_n(\cdot)$-*Lipschitz*$(\zeta^{\mathrm{L}})$. For any $\boldsymbol{\beta}$, we have

$$\left\|T_n(\boldsymbol{\beta}) - T_n(\boldsymbol{\beta}^*)\right\|_{\infty,\infty} = O_{\mathbb{P}}(\zeta^{\mathrm{L}}) \cdot \|\boldsymbol{\beta} - \boldsymbol{\beta}^*\|_1.$$

In the sequel, we lay out an assumption on several population quantities and the sample size $n$. Recall that $\boldsymbol{\beta}^* = [\alpha^*, (\boldsymbol{\gamma}^*)^\top]^\top$, where $\alpha^* \in \mathbb{R}$ is the entry of interest, while $\boldsymbol{\gamma}^* \in \mathbb{R}^{d-1}$ is the nuisance parameter. By the notations in §2.2, $[I(\boldsymbol{\beta}^*)]_{\boldsymbol{\gamma},\boldsymbol{\gamma}} \in \mathbb{R}^{(d-1)\times(d-1)}$ and $[I(\boldsymbol{\beta}^*)]_{\boldsymbol{\gamma},\alpha} \in \mathbb{R}^{(d-1)\times 1}$ denote the submatrices of the Fisher information matrix $I(\boldsymbol{\beta}^*) \in \mathbb{R}^{d \times d}$. We define $\mathbf{w}^*$, $s_{\mathbf{w}}^*$ and $\mathcal{S}_{\mathbf{w}}^*$ as

$$\mathbf{w}^* = \left[I(\boldsymbol{\beta}^*)\right]_{\boldsymbol{\gamma},\boldsymbol{\gamma}}^{-1} \cdot \left[I(\boldsymbol{\beta}^*)\right]_{\boldsymbol{\gamma},\alpha} \in \mathbb{R}^{d-1}, \quad s_{\mathbf{w}}^* = \|\mathbf{w}^*\|_0, \quad \text{and} \quad \mathcal{S}_{\mathbf{w}}^* = \mathrm{supp}(\mathbf{w}^*). \tag{4.1}$$

We define $\lambda_1[I(\boldsymbol{\beta}^*)]$ and $\lambda_d[I(\boldsymbol{\beta}^*)]$ as the largest and smallest eigenvalues of $I(\boldsymbol{\beta}^*)$, and

$$\left[I(\boldsymbol{\beta}^*)\right]_{\alpha|\boldsymbol{\gamma}} = \left[I(\boldsymbol{\beta}^*)\right]_{\alpha,\alpha} - \left[I(\boldsymbol{\beta}^*)\right]_{\boldsymbol{\gamma},\alpha}^\top \cdot \left[I(\boldsymbol{\beta}^*)\right]_{\boldsymbol{\gamma},\boldsymbol{\gamma}}^{-1} \cdot \left[I(\boldsymbol{\beta}^*)\right]_{\boldsymbol{\gamma},\alpha} \in \mathbb{R}. \tag{4.2}$$

According to (4.1) and (4.2), we can easily verify that

$$\left[I(\boldsymbol{\beta}^*)\right]_{\alpha|\boldsymbol{\gamma}} = \left[1, -(\mathbf{w}^*)^\top\right] \cdot I(\boldsymbol{\beta}^*) \cdot \left[1, -(\mathbf{w}^*)^\top\right]^\top. \tag{4.3}$$

The following assumption ensures that $\lambda_d[I(\boldsymbol{\beta}^*)] > 0$. Hence, $[I(\boldsymbol{\beta}^*)]_{\boldsymbol{\gamma},\boldsymbol{\gamma}}$ in (4.1) is invertible. Also, according to (4.3) and the fact that $\lambda_d[I(\boldsymbol{\beta}^*)] > 0$, we have $[I(\boldsymbol{\beta}^*)]_{\alpha|\boldsymbol{\gamma}} > 0$.

**Assumption 4.5** . We impose the following assumptions.

• For positive constants $\rho_{\max}$ and $\rho_{\min}$, we assume

$$\rho_{\max} \geq \lambda_1[I(\boldsymbol{\beta}^*)] \geq \lambda_d[I(\boldsymbol{\beta}^*)] \geq \rho_{\min}, \quad [I(\boldsymbol{\beta}^*)]_{\alpha|\boldsymbol{\gamma}} = O(1), \quad [I(\boldsymbol{\beta}^*)]_{\alpha|\boldsymbol{\gamma}}^{-1} = O(1). \tag{4.4}$$

• The tuning parameter $\lambda$ of the Dantzig selector in (2.6) is set to

$$\lambda = C \cdot \left(\zeta^{\mathrm{T}} + \zeta^{\mathrm{L}} \cdot \zeta^{\mathrm{EM}}\right) \cdot \left(1 + \|\mathbf{w}^*\|_1\right), \tag{4.5}$$

where $C \geq 1$ is a sufficiently large constant. The sample size $n$ is sufficiently large such that

$$\max\{\|\mathbf{w}^*\|_1, \, 1\} \cdot s_{\mathbf{w}}^* \cdot \lambda = o(1), \quad \zeta^{\mathrm{EM}} = o(1), \quad s_{\mathbf{w}}^* \cdot \lambda \cdot \zeta^{\mathrm{G}} = o(1/\sqrt{n}), \tag{4.6}$$

$$\lambda \cdot \zeta^{\mathrm{EM}} = o(1/\sqrt{n}), \quad \max\{1, \, \|\mathbf{w}^*\|_1\} \cdot \zeta^{\mathrm{L}} \cdot \left(\zeta^{\mathrm{EM}}\right)^2 = o(1/\sqrt{n}).$$

The assumption on $\lambda_d[I(\boldsymbol{\beta}^*)]$ guarantees that the Fisher information matrix is positive definite. The other assumptions in (4.4) guarantee the existence of the asymptotic variance of $\sqrt{n} \cdot S_n(\widehat{\boldsymbol{\beta}}_0, \lambda)$ in the score statistic defined in (2.7). Similar assumptions are standard in existing asymptotic inference results. For example, for mixture of regression model, [14] impose variants of these assumptions. For specific models, we will show that $\zeta^{\mathrm{EM}}$, $\zeta^{\mathrm{G}}$, $\zeta^{\mathrm{T}}$ and $\lambda$ all decrease with $n$, while $\zeta^{\mathrm{L}}$ increases with $n$ at a slow rate. Therefore, the assumptions in (4.6) ensure that the sample size $n$ is sufficiently large. We will make these assumptions more explicit after we specify $\zeta^{\mathrm{EM}}$, $\zeta^{\mathrm{G}}$, $\zeta^{\mathrm{T}}$ and $\zeta^{\mathrm{L}}$ for each

model. Note the assumptions in (4.6) imply that $s_{\mathbf{w}}^* = \|\mathbf{w}^*\|_0$ needs to be small. For instance, for $\lambda$ specified in (4.5), $\max\{\|\mathbf{w}^*\|_1, 1\} \cdot s_{\mathbf{w}}^* \cdot \lambda = o(1)$ in (4.6) implies $s_{\mathbf{w}}^* \cdot \zeta^{\mathrm{T}} = o(1)$. In the following, we will prove that $\zeta^{\mathrm{T}}$ is of the order $\sqrt{\log d/n}$. Hence, we require that $s_{\mathbf{w}}^* = o(\sqrt{n/\log d}) \ll d - 1$, i.e., $\mathbf{w}^* \in \mathbb{R}^{d-1}$ is sparse. Such a sparsity assumption can be understood as follows. According to the definition of $\mathbf{w}^*$ in (4.1), we have $\left[I(\boldsymbol{\beta}^*)\right]_{\boldsymbol{\gamma},\boldsymbol{\gamma}} \cdot \mathbf{w}^* = \left[I(\boldsymbol{\beta}^*)\right]_{\boldsymbol{\gamma},\alpha}$. Therefore, such a sparsity assumption suggests $\left[I(\boldsymbol{\beta}^*)\right]_{\boldsymbol{\gamma},\alpha}$ lies within the span of a few columns of $\left[I(\boldsymbol{\beta}^*)\right]_{\boldsymbol{\gamma},\boldsymbol{\gamma}}$. Such a sparsity assumption on $\mathbf{w}^*$ is necessary, because otherwise it is difficult to accurately estimate $\mathbf{w}^*$ in high dimensional regimes. In the context of high dimensional generalized linear models, [26, 32] impose similar sparsity assumptions.

## 4.1 Main Results

**Decorrelated Score Test:** The next theorem establishes the asymptotic normality of the decorrelated score statistic defined in (2.7).

**Theorem 4.6.** We consider $\boldsymbol{\beta}^* = \left[\alpha^*, (\boldsymbol{\gamma}^*)^\top\right]^\top$ with $\alpha^* = 0$. Under Assumption 4.5 and Conditions 4.1-4.4, we have that for $n \to \infty$,

$$\sqrt{n} \cdot S_n(\widehat{\boldsymbol{\beta}}_0, \lambda) / \left\{-\left[T_n(\widehat{\boldsymbol{\beta}}_0)\right]_{\alpha|\boldsymbol{\gamma}}\right\}^{1/2} \xrightarrow{D} N(0, 1), \tag{4.7}$$

where $\widehat{\boldsymbol{\beta}}_0$ and $\left[T_n(\widehat{\boldsymbol{\beta}}_0)\right]_{\alpha|\boldsymbol{\gamma}} \in \mathbb{R}$ are defined in (2.7). The limiting variance of the decorrelated score function $\sqrt{n} \cdot S_n(\widehat{\boldsymbol{\beta}}_0, \lambda)$ is $\left[I(\boldsymbol{\beta}^*)\right]_{\alpha|\boldsymbol{\gamma}}$, which is defined in (4.2).

*Proof.* See §G.2 of the appendix for a detailed proof. $\qquad\square$

**Optimality:** [27] prove that for inferring $\alpha^*$ in the presence of nuisance parameter $\boldsymbol{\gamma}^*$, $\left[I(\boldsymbol{\beta}^*)\right]_{\alpha|\boldsymbol{\gamma}}$ is the semiparametric efficient information, i.e., the minimum limiting variance of the (rescaled) score function. Our proposed decorrelated score function achieves such a semiparametric information lower bound and is therefore in this sense optimal.

In the following, we use Gaussian mixture model to illustrate the effectiveness of Theorem 4.6. We defer the details and the implications for mixture of regression to §F of the appendix.

**Implications for Gaussian Mixture Model:** Under the same model considered in §3.1, if we assume all quantities except $s_{\mathbf{w}}^*$, $s^*$, $d$ and $n$ are constant, then we have that Conditions 4.1-4.4 hold with $\zeta^{\mathrm{EM}} = s^* \sqrt{\log d \cdot \log n/n}$, $\zeta^{\mathrm{G}} = \sqrt{\log d/n}$, $\zeta^{\mathrm{T}} = \sqrt{\log d/n}$ and $\zeta^{\mathrm{L}} = (\log d + \log n)^{3/2}$. Thus, under Assumption 4.5, (4.7) holds when $n \to \infty$. Also, we can verify that (4.6) in Assumption 4.5 holds if $\max\{s_{\mathbf{w}}^*, s^*\}^2 \cdot (s^*)^2 \cdot (\log d)^5 = o[n/(\log n)^2]$.

## 5 Conclusion

We propose a novel high dimensional EM algorithm which naturally incorporates sparsity structure. Our theory shows that, with a suitable initialization, the proposed algorithm converges at a geometric rate and achieves an estimator with the (near-)optimal statistical rate of convergence. Beyond point estimation, we further propose the decorrelated score and Wald statistics for testing hypotheses and constructing confidence intervals for low dimensional components of high dimensional parameters. We apply the proposed algorithmic framework to a broad family of high dimensional latent variable models. For these models, our framework establishes the first computationally feasible approach for optimal parameter estimation and asymptotic inference under high dimensional settings.

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
