[Supplementary Material]

# A    Applications to Latent Variable Models

In the sequel, we introduce two latent variable models as examples. To apply the high dimensional EM algorithm in §2.1 and the methods for asymptotic inference in §2.2, we only need to specify the forms of $Q_n(\cdot;\cdot)$ defined in (2.1), $M_n(\cdot)$ in Algorithms 2 and 3, and $T_n(\cdot)$ in (2.4) for each model.

**Gaussian Mixture Model:** Let $\mathbf{y}_1,\ldots,\mathbf{y}_n$ be the $n$ i.i.d. realizations of $\mathbf{Y} \in \mathbb{R}^d$ and

$$\mathbf{Y} = Z \cdot \boldsymbol{\beta}^* + \mathbf{V}. \tag{A.1}$$

Here $Z$ is a Rademacher random variable, i.e., $\mathbb{P}(Z = +1) = \mathbb{P}(Z = -1) = 1/2$, and $\mathbf{V} \sim N(\mathbf{0}, \sigma^2 \cdot \mathbf{I}_d)$ is independent of $Z$, where $\sigma$ is the standard deviation. We suppose $\sigma$ is known. In high dimensional settings, we assume that $\boldsymbol{\beta}^* \in \mathbb{R}^d$ is sparse. To avoid the degenerate case in which the two Gaussians in the mixture are identical, here we suppose that $\boldsymbol{\beta}^* \neq \mathbf{0}$.

For the E-step (line 4 of Algorithm 1), we have

$$Q_n(\boldsymbol{\beta}';\boldsymbol{\beta}) = -\frac{1}{2n}\sum_{i=1}^n \omega_{\boldsymbol{\beta}}(\mathbf{y}_i) \cdot \|\mathbf{y}_i - \boldsymbol{\beta}'\|_2^2 + \left[1 - \omega_{\boldsymbol{\beta}}(\mathbf{y}_i)\right] \cdot \|\mathbf{y}_i + \boldsymbol{\beta}'\|_2^2, \tag{A.2}$$

$$\text{where } \omega_{\boldsymbol{\beta}}(\mathbf{y}) = \frac{1}{1 + \exp\left(-\langle\boldsymbol{\beta},\mathbf{y}\rangle/\sigma^2\right)}.$$

The maximization implementation (Algorithm 2) of the M-step takes the form

$$M_n(\boldsymbol{\beta}) = \frac{2}{n}\sum_{i=1}^n \omega_{\boldsymbol{\beta}}(\mathbf{y}_i) \cdot \mathbf{y}_i - \frac{1}{n}\sum_{i=1}^n \mathbf{y}_i. \tag{A.3}$$

Meanwhile, for the gradient ascent implementation (Algorithm 3) of the M-step, we have

$$M_n(\boldsymbol{\beta}) = \boldsymbol{\beta} + \eta \cdot \nabla_1 Q_n(\boldsymbol{\beta};\boldsymbol{\beta}), \quad \text{where } \nabla_1 Q_n(\boldsymbol{\beta};\boldsymbol{\beta}) = \frac{1}{n}\sum_{i=1}^n \left[2\cdot\omega_{\boldsymbol{\beta}}(\mathbf{y}_i) - 1\right]\cdot\mathbf{y}_i - \boldsymbol{\beta}.$$

Here $\eta > 0$ is the stepsize. For asymptotic inference, $T_n(\cdot)$ in (2.4) takes the form

$$T_n(\boldsymbol{\beta}) = \frac{1}{n}\sum_{i=1}^n \nu_{\boldsymbol{\beta}}(\mathbf{y}_i)\cdot\mathbf{y}_i\cdot\mathbf{y}_i^\top - \mathbf{I}_d,$$

$$\text{where } \nu_{\boldsymbol{\beta}}(\mathbf{y}) = \frac{4/\sigma^2}{\left[1 + \exp\left(-2\cdot\langle\boldsymbol{\beta},\mathbf{y}\rangle/\sigma^2\right)\right]\cdot\left[1 + \exp\left(2\cdot\langle\boldsymbol{\beta},\mathbf{y}\rangle/\sigma^2\right)\right]}.$$

**Mixture of Regression Model:** We assume that $Y \in \mathbb{R}$ and $\mathbf{X} \in \mathbb{R}^d$ satisfy

$$Y = Z\cdot\mathbf{X}^\top\boldsymbol{\beta}^* + V, \tag{A.4}$$

where $\mathbf{X} \sim N(\mathbf{0},\mathbf{I}_d)$, $V \sim N(0,\sigma^2)$ and $Z$ is a Rademacher random variable. Here $\mathbf{X}$, $V$ and $Z$ are independent. In the high dimensional regime, we assume $\boldsymbol{\beta}^* \in \mathbb{R}^d$ is sparse. To avoid the degenerate case, we suppose $\boldsymbol{\beta}^* \neq \mathbf{0}$. In addition, we assume that $\sigma$ is known. For the E-step (line 4 of Algorithm 1), we have

$$Q_n(\boldsymbol{\beta}';\boldsymbol{\beta}) = -\frac{1}{2n}\sum_{i=1}^n \omega_{\boldsymbol{\beta}}(\mathbf{x}_i,y_i)\cdot\left(y_i - \langle\mathbf{x}_i,\boldsymbol{\beta}'\rangle\right)^2 + \left[1 - \omega_{\boldsymbol{\beta}}(\mathbf{x}_i,y_i)\right]\cdot\left(y_i + \langle\mathbf{x}_i,\boldsymbol{\beta}'\rangle\right)^2, \tag{A.5}$$

$$\text{where } \omega_{\boldsymbol{\beta}}(\mathbf{x},y) = \frac{1}{1 + \exp\left(-y\cdot\langle\boldsymbol{\beta},\mathbf{x}\rangle/\sigma^2\right)}.$$

For the maximization implementation (Algorithm 2) of the M-step (line 5 of Algorithm 1), we have that $M_n(\boldsymbol{\beta}) = \operatorname{argmax}_{\boldsymbol{\beta}'} Q_n(\boldsymbol{\beta}';\boldsymbol{\beta})$ satisfies

$$\widehat{\boldsymbol{\Sigma}}\cdot M_n(\boldsymbol{\beta}) = \frac{1}{n}\sum_{i=1}^n \left[2\cdot\omega_{\boldsymbol{\beta}}(\mathbf{x}_i,y_i) - 1\right]\cdot y_i\cdot\mathbf{x}_i, \quad \text{where } \widehat{\boldsymbol{\Sigma}} = \frac{1}{n}\sum_{i=1}^n \mathbf{x}_i\cdot\mathbf{x}_i^\top. \tag{A.6}$$

However, in high dimensional regimes, the sample covariance matrix $\widehat{\boldsymbol{\Sigma}}$ is not invertible. To estimate the inverse covariance matrix of $\mathbf{X}$, we use the CLIME estimator proposed by [7], i.e.,

$$\widehat{\boldsymbol{\Theta}} = \operatorname*{argmin}_{\boldsymbol{\Theta}\in\mathbb{R}^{d\times d}} \|\boldsymbol{\Theta}\|_{1,1}, \quad \text{subject to } \left\|\widehat{\boldsymbol{\Sigma}}\cdot\boldsymbol{\Theta} - \mathbf{I}_d\right\|_{\infty,\infty} \leq \lambda^{\mathrm{CLIME}}, \tag{A.7}$$

where $\|\cdot\|_{1,1}$ and $\|\cdot\|_{\infty,\infty}$ are the sum and maximum of the absolute values of all entries respectively, and $\lambda^{\mathrm{CLIME}} > 0$ is a tuning parameter. Based on (A.6), we modify the maximization implementation

of the M-step to be

$$M_n(\boldsymbol{\beta}) = \widehat{\boldsymbol{\Theta}} \cdot \frac{1}{n} \sum_{i=1}^{n} \left[ 2 \cdot \omega_{\boldsymbol{\beta}}(\mathbf{x}_i, y_i) - 1 \right] \cdot y_i \cdot \mathbf{x}_i. \tag{A.8}$$

For the gradient ascent implementation (Algorithm 3) of the M-step, we have

$$M_n(\boldsymbol{\beta}) = \boldsymbol{\beta} + \eta \cdot \nabla_1 Q_n(\boldsymbol{\beta}; \boldsymbol{\beta}), \tag{A.9}$$

$$\text{where} \quad \nabla_1 Q_n(\boldsymbol{\beta}, \boldsymbol{\beta}) = \frac{1}{n} \sum_{i=1}^{n} \left[ 2 \cdot \omega_{\boldsymbol{\beta}}(\mathbf{x}_i, y_i) \cdot y_i \cdot \mathbf{x}_i - \mathbf{x}_i \cdot \mathbf{x}_i^\top \cdot \boldsymbol{\beta} \right].$$

Here $\eta > 0$ is the stepsize. For asymptotic inference, $T_n(\cdot)$ in (2.4) takes the form

$$T_n(\boldsymbol{\beta}) = \frac{1}{n} \sum_{i=1}^{n} \nu_{\boldsymbol{\beta}}(\mathbf{x}_i, y_i) \cdot \mathbf{x}_i \cdot \mathbf{x}_i^\top \cdot y_i^2 - \frac{1}{n} \sum_{i=1}^{n} \mathbf{x}_i \cdot \mathbf{x}_i^\top,$$

$$\text{where} \quad \nu_{\boldsymbol{\beta}}(\mathbf{x}, y) = \frac{4/\sigma^2}{\left[ 1 + \exp\left( -2 \cdot y \cdot \langle \boldsymbol{\beta}, \mathbf{x} \rangle / \sigma^2 \right) \right] \cdot \left[ 1 + \exp\left( 2 \cdot y \cdot \langle \boldsymbol{\beta}, \mathbf{x} \rangle / \sigma^2 \right) \right]}.$$

It is worth noting that, for the maximization implementation of the M-step, the CLIME estimator in (A.7) requires that $\boldsymbol{\Sigma}^{-1}$ is sparse, where $\boldsymbol{\Sigma}$ is the population covariance of $\boldsymbol{X}$. Since we assume $\boldsymbol{X} \sim N(\mathbf{0}, \mathbf{I}_d)$, this requirement is satisfied. Nevertheless, for more general settings where $\boldsymbol{\Sigma}$ does not possess such a structure, the gradient ascent implementation of the M-step is a better choice, since it does not require inverse covariance estimation and is also more efficient in computation.

## B  Derivation of the EM Algorithm

Recall that in §2.1, we assume that $h_{\boldsymbol{\beta}}(\mathbf{y})$ is obtained by marginalizing over an unobserved latent variable $\boldsymbol{Z} \in \mathcal{Z}$, i.e.,

$$h_{\boldsymbol{\beta}}(\mathbf{y}) = \int_{\mathcal{Z}} f_{\boldsymbol{\beta}}(\mathbf{y}, \mathbf{z}) \, d\mathbf{z}. \tag{B.1}$$

Let $k_{\boldsymbol{\beta}}(\mathbf{z} \mid \mathbf{y})$ be the density of $\boldsymbol{Z}$ conditioning on the observed variable $\boldsymbol{Y} = \mathbf{y}$, i.e.,

$$k_{\boldsymbol{\beta}}(\mathbf{z} \mid \mathbf{y}) = f_{\boldsymbol{\beta}}(\mathbf{y}, \mathbf{z}) / h_{\boldsymbol{\beta}}(\mathbf{y}). \tag{B.2}$$

Given the $n$ observations $\mathbf{y}_1, \dots, \mathbf{y}_n$ of $\boldsymbol{Y}$, the EM algorithm aims at maximizing the log-likelihood

$$\ell_n(\boldsymbol{\beta}) = \sum_{i=1}^{n} \log h_{\boldsymbol{\beta}}(\mathbf{y}_i). \tag{B.3}$$

Due to the unobserved latent variable $\boldsymbol{Z}$, it is difficult to directly evaluate $\ell_n(\boldsymbol{\beta})$. Instead, we turn to consider the difference between $\ell_n(\boldsymbol{\beta})$ and $\ell_n(\boldsymbol{\beta}')$. Let $k_{\boldsymbol{\beta}}(\mathbf{z} \mid \mathbf{y})$ be the density of $\boldsymbol{Z}$ conditioning on the observed variable $\boldsymbol{Y} = \mathbf{y}$, i.e.,

$$k_{\boldsymbol{\beta}}(\mathbf{z} \mid \mathbf{y}) = f_{\boldsymbol{\beta}}(\mathbf{y}, \mathbf{z}) / h_{\boldsymbol{\beta}}(\mathbf{y}). \tag{B.4}$$

According to (B.1) and (B.3), we have

$$\frac{1}{n} \cdot \left[ \ell_n(\boldsymbol{\beta}) - \ell_n(\boldsymbol{\beta}') \right] = \frac{1}{n} \sum_{i=1}^{n} \log \left[ h_{\boldsymbol{\beta}}(\mathbf{y}_i) / h_{\boldsymbol{\beta}'}(\mathbf{y}_i) \right] = \frac{1}{n} \sum_{i=1}^{n} \log \left[ \int_{\mathcal{Z}} \frac{f_{\boldsymbol{\beta}}(\mathbf{y}_i, \mathbf{z})}{h_{\boldsymbol{\beta}'}(\mathbf{y}_i)} \, d\mathbf{z} \right]$$

$$= \frac{1}{n} \sum_{i=1}^{n} \log \left[ \int_{\mathcal{Z}} k_{\boldsymbol{\beta}'}(\mathbf{z} \mid \mathbf{y}_i) \cdot \frac{f_{\boldsymbol{\beta}}(\mathbf{y}_i, \mathbf{z})}{f_{\boldsymbol{\beta}'}(\mathbf{y}_i, \mathbf{z})} \, d\mathbf{z} \right] \geq \frac{1}{n} \sum_{i=1}^{n} \int_{\mathcal{Z}} k_{\boldsymbol{\beta}'}(\mathbf{z} \mid \mathbf{y}_i) \cdot \log \left[ \frac{f_{\boldsymbol{\beta}}(\mathbf{y}_i, \mathbf{z})}{f_{\boldsymbol{\beta}'}(\mathbf{y}_i, \mathbf{z})} \right] \, d\mathbf{z}, \tag{B.5}$$

where the third equality follows from (B.4) and the inequality is obtained from Jensen's inequality. On the right-hand side of (B.5) we have

$$\frac{1}{n} \sum_{i=1}^{n} \int_{\mathcal{Z}} k_{\boldsymbol{\beta}'}(\mathbf{z} \mid \mathbf{y}_i) \cdot \log \left[ \frac{f_{\boldsymbol{\beta}}(\mathbf{y}_i, \mathbf{z})}{f_{\boldsymbol{\beta}'}(\mathbf{y}_i, \mathbf{z})} \right] \, d\mathbf{z}$$

$$= \underbrace{\frac{1}{n} \sum_{i=1}^{n} \int_{\mathcal{Z}} k_{\boldsymbol{\beta}'}(\mathbf{z} \mid \mathbf{y}_i) \cdot \log f_{\boldsymbol{\beta}}(\mathbf{y}_i, \mathbf{z}) \, d\mathbf{z}}_{Q_n(\boldsymbol{\beta}; \boldsymbol{\beta}')} - \frac{1}{n} \sum_{i=1}^{n} \int_{\mathcal{Z}} k_{\boldsymbol{\beta}'}(\mathbf{z} \mid \mathbf{y}_i) \cdot \log f_{\boldsymbol{\beta}'}(\mathbf{y}_i, \mathbf{z}) \, d\mathbf{z}. \tag{B.6}$$

We define the first term on the right-hand side of (B.6) to be $Q_n(\boldsymbol{\beta}; \boldsymbol{\beta}')$. Correspondingly, we define its expectation to be $Q(\boldsymbol{\beta}; \boldsymbol{\beta}')$. Note the second term on the right-hand side of (B.6) does not depend on $\boldsymbol{\beta}$. Hence, given some fixed $\boldsymbol{\beta}'$, we can maximize the lower bound function $Q_n(\boldsymbol{\beta}; \boldsymbol{\beta}')$ over $\boldsymbol{\beta}$ to obtain a sufficiently large $\ell_n(\boldsymbol{\beta}) - \ell_n(\boldsymbol{\beta}')$. Based on such an observation, at the $t$-th iteration of the classical EM algorithm, we evaluate $Q_n(\boldsymbol{\beta}; \boldsymbol{\beta}^{(t)})$ at the E-step and then perform $\max_{\boldsymbol{\beta}} Q_n(\boldsymbol{\beta}; \boldsymbol{\beta}^{(t)})$ at the M-step. See [18] for more details.

## C  High Dimensional EM Algorithm with Resampling

To simplify the technical analysis of the high dimensional algorithm, here we introduce its resampling version (Algorithm 4).

---

**Algorithm 4** High Dimensional EM Algorithm with Resampling.

---

1: **Parameter:** Sparsity Parameter $\widehat{s}$, Maximum Number of Iterations $T$
2: **Initialization:** $\widehat{\mathcal{S}}^{\text{init}} \leftarrow \text{supp}(\boldsymbol{\beta}^{\text{init}}, \widehat{s})$, $\quad \boldsymbol{\beta}^{(0)} \leftarrow \text{trunc}(\boldsymbol{\beta}^{\text{init}}, \widehat{\mathcal{S}}^{\text{init}})$,
$\quad\quad\quad \big\{ \text{supp}(\cdot, \cdot) \text{ and } \text{trunc}(\cdot, \cdot) \text{ are defined in (2.2) and (2.3)} \big\}$
$\quad\quad\quad$ Split the Dataset into $T$ Subsets of Size $n/T$
$\quad\quad\quad \big\{ \text{Without loss of generality, we assume } n/T \text{ is an integer} \big\}$
3: **For** $t = 0$ to $T - 1$
4: $\quad$ **E-step:** Evaluate $Q_{n/T}(\boldsymbol{\beta}; \boldsymbol{\beta}^{(t)})$ with the $t$-th Data Subset
5: $\quad$ **M-step:** $\boldsymbol{\beta}^{(t+0.5)} \leftarrow M_{n/T}(\boldsymbol{\beta}^{(t)})$
$\quad\quad\quad \big\{ M_{n/T}(\cdot) \text{ is implemented as in Algorithm 2 or 3 with } Q_n(\cdot; \cdot) \text{ replaced by } Q_{n/T}(\cdot; \cdot) \big\}$
6: $\quad$ **T-step:** $\widehat{\mathcal{S}}^{(t+0.5)} \leftarrow \text{supp}(\boldsymbol{\beta}^{(t+0.5)}, \widehat{s})$, $\quad \boldsymbol{\beta}^{(t+1)} \leftarrow \text{trunc}(\boldsymbol{\beta}^{(t+0.5)}, \widehat{\mathcal{S}}^{(t+0.5)})$
7: **End For**
8: **Output:** $\widehat{\boldsymbol{\beta}} \leftarrow \boldsymbol{\beta}^{(T)}$

---

## D  Decorrelated Score Statistic: An Intuitive Explanation

The intuition for the decorrelated score statistic in (2.7) can be understood as follows. Since $\ell_n(\boldsymbol{\beta})$ is the log-likelihood, its score function is $\nabla \ell_n(\boldsymbol{\beta})$ and the Fisher information at $\boldsymbol{\beta}^*$ is $I(\boldsymbol{\beta}^*) = -\mathbb{E}_{\boldsymbol{\beta}^*}[\nabla^2 \ell_n(\boldsymbol{\beta}^*)]/n$, where $\mathbb{E}_{\boldsymbol{\beta}^*}(\cdot)$ means the expectation is taken under the model with parameter $\boldsymbol{\beta}^*$. The following key theorem, which restates Theorem 2.1, reveals the connection of $\nabla_1 Q_n(\cdot; \cdot)$ in (2.5) and $T_n(\cdot)$ in (2.7) with the score function and Fisher information, which forms the foundation of our inferential method.

**Theorem D.1.** For the true parameter $\boldsymbol{\beta}^*$ and any $\boldsymbol{\beta} \in \mathbb{R}^d$, it holds that

$$\nabla_1 Q_n(\boldsymbol{\beta}; \boldsymbol{\beta}) = \nabla \ell_n(\boldsymbol{\beta})/n, \quad \text{and} \quad \mathbb{E}_{\boldsymbol{\beta}^*}[T_n(\boldsymbol{\beta}^*)] = -I(\boldsymbol{\beta}^*) = \mathbb{E}_{\boldsymbol{\beta}^*}[\nabla^2 \ell_n(\boldsymbol{\beta}^*)]/n. \quad \text{(D.1)}$$

*Proof.* See §I.1 for details. $\quad\quad\quad\quad\quad\quad\quad\quad\quad\quad\quad\quad\quad\quad\quad\quad\quad\quad\quad\quad\quad\quad\quad\quad\quad$ $\square$

Recall that the log-likelihood $\ell_n(\boldsymbol{\beta})$ defined in (B.3) is difficult to evaluate due to the unobserved latent variable. Theorem D.1 provides a feasible way to calculate or estimate the corresponding score function and Fisher information, since $Q_n(\cdot; \cdot)$ and $T_n(\cdot)$ have closed forms. The geometric intuition behind Theorem D.1 can be understood as follows. By (B.5) and (B.6) we have

$$\ell_n(\boldsymbol{\beta}) \geq \ell_n(\boldsymbol{\beta}') + n \cdot Q_n(\boldsymbol{\beta}; \boldsymbol{\beta}') - \sum_{i=1}^{n} \int_{\mathcal{Z}} k_{\boldsymbol{\beta}'}(\mathbf{z} \mid \mathbf{y}_i) \cdot \log f_{\boldsymbol{\beta}'}(\mathbf{y}_i, \mathbf{z}) \, d\mathbf{z}. \quad \text{(D.2)}$$

By (D.1), both sides of (D.2) have the same gradient with respect to $\boldsymbol{\beta}$ at $\boldsymbol{\beta}' = \boldsymbol{\beta}$. Furthermore, by (B.6), (D.2) becomes an equality for $\boldsymbol{\beta}' = \boldsymbol{\beta}$. Therefore, the lower bound function on the right-hand side of (D.2) is tangent to $\ell_n(\boldsymbol{\beta})$ at $\boldsymbol{\beta}' = \boldsymbol{\beta}$. Meanwhile, according to (2.4), $T_n(\boldsymbol{\beta})$ defines a modified curvature for the right-hand side of (D.2), which is obtained by taking derivative with respect to $\boldsymbol{\beta}$, then setting $\boldsymbol{\beta}' = \boldsymbol{\beta}$ and taking the second order derivative with respect to $\boldsymbol{\beta}$. The second equation in (D.1) shows that the obtained curvature equals the curvature of $\ell_n(\boldsymbol{\beta})$ at $\boldsymbol{\beta} = \boldsymbol{\beta}^*$ in expectation (up to a renormalization factor of $n$). Therefore, $\nabla_1 Q_n(\boldsymbol{\beta}; \boldsymbol{\beta})$ gives the score function and $T_n(\boldsymbol{\beta}^*)$ gives a good estimate of the Fisher information $I(\boldsymbol{\beta}^*)$. Since $\boldsymbol{\beta}^*$ is unknown in practice, later we will use $T_n(\widehat{\boldsymbol{\beta}})$ or $T_n(\widehat{\boldsymbol{\beta}}_0)$ to approximate $T_n(\boldsymbol{\beta}^*)$.

In the presence of the high dimensional nuisance parameter $\boldsymbol{\gamma}^* \in \mathbb{R}^{d-1}$, the classical score test is no longer applicable. In detail, the score test for $H_0 : \alpha^* = 0$ relies on the following Taylor expansion

of the score function $\partial \ell_n(\cdot)/\partial \alpha$

$$\frac{1}{\sqrt{n}} \cdot \frac{\partial \ell_n(\bar{\boldsymbol{\beta}}_0)}{\partial \alpha} = \frac{1}{\sqrt{n}} \cdot \frac{\partial \ell_n(\boldsymbol{\beta}^*)}{\partial \alpha} + \frac{1}{\sqrt{n}} \cdot \frac{\partial^2 \ell_n(\boldsymbol{\beta}^*)}{\partial \alpha \partial \boldsymbol{\gamma}} \cdot (\bar{\boldsymbol{\gamma}} - \boldsymbol{\gamma}^*) + \bar{R}. \tag{D.3}$$

Here $\boldsymbol{\beta}^* = \left[0, (\boldsymbol{\gamma}^*)^\top\right]^\top$, $\bar{R}$ denotes the remainder term and $\bar{\boldsymbol{\beta}}_0 = \left(0, \bar{\boldsymbol{\gamma}}^\top\right)^\top$, where $\bar{\boldsymbol{\gamma}}$ is an estimator of the nuisance parameter $\boldsymbol{\gamma}^*$. The asymptotic normality of $1/\sqrt{n} \cdot \partial \ell_n(\bar{\boldsymbol{\beta}}_0)/\partial \alpha$ in (D.3) relies on the fact that $1/\sqrt{n} \cdot \partial \ell_n(\boldsymbol{\beta}_0^*)/\partial \alpha$ and $\sqrt{n} \cdot (\bar{\boldsymbol{\gamma}} - \boldsymbol{\gamma}^*)$ are jointly normal asymptotically and $\bar{R}$ is $o_{\mathbb{P}}(1)$. In low dimensional settings, such a necessary condition holds for $\bar{\boldsymbol{\gamma}}$ being the maximum likelihood estimator. However, in high dimensional settings, the maximum likelihood estimator cannot guarantee that $\bar{R}$ is $o_{\mathbb{P}}(1)$, since $\|\bar{\boldsymbol{\gamma}} - \boldsymbol{\gamma}^*\|_2$ can be large due to the curse of dimensionality. Meanwhile, for $\bar{\boldsymbol{\gamma}}$ being sparsity-type estimators, in general the asymptotic normality of $\sqrt{n} \cdot (\bar{\boldsymbol{\gamma}} - \boldsymbol{\gamma}^*)$ does not hold. For example, let $\bar{\boldsymbol{\gamma}}$ be $\widehat{\boldsymbol{\gamma}}$, where $\widehat{\boldsymbol{\gamma}} \in \mathbb{R}^{d-1}$ is the subvector of $\widehat{\boldsymbol{\beta}}$, i.e., the estimator attained by the proposed high dimensional EM algorithm. Note that $\widehat{\boldsymbol{\gamma}}$ has many zero entries due to the truncation step. As $n \to \infty$, some entries of $\sqrt{n} \cdot (\widehat{\boldsymbol{\gamma}} - \boldsymbol{\gamma}^*)$ have limiting distributions with point mass at zero. Clearly, this limiting distribution is not Gaussian with nonzero variance. In fact, for a similar setting of high dimensional linear regression, [15] illustrate that for $\boldsymbol{\gamma}^\sharp$ being a subvector of the Lasso estimator and $\boldsymbol{\gamma}^*$ being the corresponding subvector of the true parameter, the limiting distribution of $\sqrt{n} \cdot (\boldsymbol{\gamma}^\sharp - \boldsymbol{\gamma}^*)$ is not Gaussian.

The decorrelated score function defined in (2.5) successfully addresses the above issues. In detail, according to (D.1) in Theorem D.1 we have

$$\sqrt{n} \cdot S_n(\widehat{\boldsymbol{\beta}}_0, \lambda) = \frac{1}{\sqrt{n}} \cdot \frac{\partial \ell_n(\widehat{\boldsymbol{\beta}}_0)}{\partial \alpha} - \frac{1}{\sqrt{n}} \cdot w(\widehat{\boldsymbol{\beta}}_0, \lambda)^\top \cdot \frac{\partial \ell_n(\widehat{\boldsymbol{\beta}}_0)}{\partial \boldsymbol{\gamma}}. \tag{D.4}$$

Intuitively, if we replace $w(\widehat{\boldsymbol{\beta}}_0, \lambda)$ with $\mathbf{w} \in \mathbb{R}^{d-1}$ that satisfies

$$\mathbf{w}^\top \cdot \frac{\partial^2 \ell_n(\boldsymbol{\beta}^*)}{\partial^2 \boldsymbol{\gamma}} = \frac{\partial^2 \ell_n(\boldsymbol{\beta}^*)}{\partial \alpha \partial \boldsymbol{\gamma}}, \tag{D.5}$$

we have the following Taylor expansion of the decorrelated score function

$$\frac{1}{\sqrt{n}} \cdot \frac{\partial \ell_n(\widehat{\boldsymbol{\beta}}_0)}{\partial \alpha} - \frac{\mathbf{w}^\top}{\sqrt{n}} \cdot \frac{\partial \ell_n(\widehat{\boldsymbol{\beta}}_0)}{\partial \boldsymbol{\gamma}} = \overbrace{\frac{1}{\sqrt{n}} \cdot \frac{\partial \ell_n(\boldsymbol{\beta}^*)}{\partial \alpha} - \frac{\mathbf{w}^\top}{\sqrt{n}} \cdot \frac{\partial \ell_n(\boldsymbol{\beta}^*)}{\partial \boldsymbol{\gamma}}}^{(i)} \tag{D.6}$$

$$+ \underbrace{\frac{1}{\sqrt{n}} \cdot \frac{\partial^2 \ell_n(\boldsymbol{\beta}^*)}{\partial \alpha \partial \boldsymbol{\gamma}} \cdot (\widehat{\boldsymbol{\gamma}} - \boldsymbol{\gamma}^*) - \frac{\mathbf{w}^\top}{\sqrt{n}} \cdot \frac{\partial^2 \ell_n(\boldsymbol{\beta}^*)}{\partial^2 \boldsymbol{\gamma}} \cdot (\widehat{\boldsymbol{\gamma}} - \boldsymbol{\gamma}^*)}_{(ii)} + \widetilde{R},$$

where term (ii) is zero by (D.5). Therefore, we no longer require the asymptotic normality of $\widehat{\boldsymbol{\gamma}} - \boldsymbol{\gamma}^*$. Also, we will prove that the new remainder term $\widetilde{R}$ in (D.6) is $o_{\mathbb{P}}(1)$, since $\widehat{\boldsymbol{\gamma}}$ has a fast statistical rate of convergence. Now we only need to find the $\mathbf{w}$ that satisfies (D.5). Nevertheless, it is difficult to calculate the second order derivatives in (D.5), because it is hard to evaluate $\ell_n(\cdot)$. According to (D.1), we use the submatrices of $T_n(\cdot)$ to approximate the derivatives in (D.5). Since $\left[T_n(\boldsymbol{\beta})\right]_{\boldsymbol{\gamma}, \boldsymbol{\gamma}}$ is not invertible in high dimensions, we use the Dantzig selector in (2.6) to approximately solve the linear system in (D.5). Based on this intuition, we can expect that $\sqrt{n} \cdot S_n(\widehat{\boldsymbol{\beta}}_0, \lambda)$ is asymptotically normal, since term (i) in (D.6) is a (rescaled) average of $n$ i.i.d. random variables for which we can apply the central limit theorem. Besides, we will prove that $-\left[T_n(\widehat{\boldsymbol{\beta}}_0)\right]_{\alpha|\boldsymbol{\gamma}}$ in (2.7) is a consistent estimator of $\sqrt{n} \cdot S_n(\widehat{\boldsymbol{\beta}}_0, \lambda)$'s asymptotic variance. Hence, we can expect that the decorrelated score statistic in (2.7) is asymptotically $N(0, 1)$.

From a high-level perspective, we can view $w(\widehat{\boldsymbol{\beta}}_0, \lambda)^\top \cdot \partial \ell_n(\widehat{\boldsymbol{\beta}}_0)/\partial \boldsymbol{\gamma}$ in (D.4) as the projection of $\partial \ell_n(\widehat{\boldsymbol{\beta}}_0)/\partial \alpha$ onto the span of $\partial \ell_n(\widehat{\boldsymbol{\beta}}_0)/\partial \boldsymbol{\gamma}$, where $w(\widehat{\boldsymbol{\beta}}_0, \lambda)$ is the projection coefficient. Intuitively, such a projection guarantees that in (D.4), $S_n(\widehat{\boldsymbol{\beta}}_0, \lambda)$ is orthogonal (uncorrelated) with $\partial \ell_n(\widehat{\boldsymbol{\beta}}_0)/\partial \boldsymbol{\gamma}$, i.e., the score function with respect to the nuisance parameter $\boldsymbol{\gamma}$. In this way, the projection corrects the effects of the high dimensional nuisance parameter. According to this intuition of decorrelation, we name $S_n(\widehat{\boldsymbol{\beta}}_0, \lambda)$ as the decorrelated score function.

# E    Implications for Specific Models: Computation and Estimation

To establish the corresponding results for specific models under the unified framework, we only need to establish Conditions 3.1-3.3 and determine the key quantities $R$, $\gamma_1$, $\gamma_2$, $\nu$, $\mu$, $\kappa$ and $\epsilon$. Recall that Conditions 3.1 and 3.2 and the models analyzed in our paper are identical to those in [2]. Meanwhile, note that Conditions 3.1 and 3.2 only involve the population version lower bound function $Q(\cdot;\cdot)$ and M-step $M(\cdot)$. Since [2] prove that the quantities in Conditions 3.1 and 3.2 are independent of the dimension $d$ and sample size $n$, their corresponding results can be directly adapted. To establish the final results, it still remains to verify Condition 3.3 for each high dimensional latent variable model.

**Gaussian Mixture Model:** The following lemma, which is proved by [2], verifies Conditions 3.1 and 3.2 for Gaussian mixture model. Recall that $\sigma$ is the standard deviation of each individual Gaussian distribution within the mixture.

**Lemma E.1.** Suppose that we have $\|\boldsymbol{\beta}^*\|_2/\sigma \geq r$, where $r > 0$ is a sufficiently large constant that denotes the minimum signal-to-noise ratio. There exists some constant $C > 0$ such that Conditions *Lipschitz-Gradient*-$1(\gamma_1, \mathcal{B})$ and *Concavity-Smoothness*$(\mu, \nu, \mathcal{B})$ hold with

$$\gamma_1 = \exp(-C \cdot r^2), \quad \mu = \nu = 1, \quad \mathcal{B} = \{\boldsymbol{\beta} : \|\boldsymbol{\beta} - \boldsymbol{\beta}^*\|_2 \leq R\} \text{ with } R = \kappa \cdot \|\boldsymbol{\beta}^*\|_2, \; \kappa = 1/4. \tag{E.1}$$

*Proof.* See the proof of Corollary 1 in [2] for details. $\qquad\square$

Now we verify Condition 3.3 for the maximization implementation of the M-step (Algorithm 2).

**Lemma E.2.** For the maximization implementation of the M-step (Algorithm 2), we have that for a sufficiently large $n$ and $\mathcal{B}$ specified in (E.1), Condition *Statistical-Error*$(\epsilon, \delta, s, n, \mathcal{B})$ holds with

$$\epsilon = C \cdot \left(\|\boldsymbol{\beta}^*\|_\infty + \sigma\right) \cdot \sqrt{\frac{\log d + \log(2/\delta)}{n}}. \tag{E.2}$$

*Proof.* See §H.4 for a detailed proof. $\qquad\square$

The next theorem establishes the implication of Theorem 3.4 for Gaussian mixture model.

**Theorem E.3.** We consider the maximization implementation of M-step (Algorithm 2). We assume $\|\boldsymbol{\beta}^*\|_2/\sigma \geq r$ holds with a sufficiently large $r > 0$, and $\mathcal{B}$ and $R$ are as defined in (E.1). We suppose the initialization $\boldsymbol{\beta}^{\text{init}}$ of Algorithm 4 satisfies $\left\|\boldsymbol{\beta}^{\text{init}} - \boldsymbol{\beta}^*\right\|_2 \leq R/2$. Let the sparsity parameter $\widehat{s}$ be

$$\widehat{s} = \left\lceil C' \cdot \max\left\{16 \cdot \left[\exp(C \cdot r^2) - 1\right]^{-2}, \; 100/9\right\} \cdot s^*\right\rceil \tag{E.3}$$

with $C$ specified in (E.1) and $C' \geq 1$. Let the total number of iterations $T$ in Algorithm 4 be

$$T = \left\lceil \frac{\log\left\{C' \cdot R / \left[\Delta^{\text{GMM}}(s^*) \cdot \sqrt{\log d/n}\right]\right\}}{C \cdot r^2/2} \right\rceil, \tag{E.4}$$

$$\text{where } \Delta^{\text{GMM}}(s^*) = \left(\sqrt{\widehat{s}} + C'' \cdot \sqrt{s^*}\right) \cdot \left(\|\boldsymbol{\beta}^*\|_\infty + \sigma\right).$$

Meanwhile, we suppose the dimension $d$ is sufficiently large such that $T$ in (E.4) is upper bounded by $\sqrt{d}$, and the sample size $n$ is sufficiently large such that

$$C' \cdot \Delta^{\text{GMM}}(s^*) \cdot \sqrt{\frac{\log d \cdot T}{n}} \leq \min\left\{\left[1 - \exp(-C \cdot r^2/2)\right]^2 \cdot R, \; 9/40 \cdot \|\boldsymbol{\beta}^*\|_2\right\}. \tag{E.5}$$

We have that, with probability at least $1 - 2 \cdot d^{-1/2}$, the final estimator $\widehat{\boldsymbol{\beta}} = \boldsymbol{\beta}^{(T)}$ satisfies

$$\left\|\widehat{\boldsymbol{\beta}} - \boldsymbol{\beta}^*\right\|_2 \leq \frac{C' \cdot \Delta^{\text{GMM}}(s^*)}{1 - \exp(-C \cdot r^2/2)} \cdot \sqrt{\frac{\log d \cdot T}{n}}. \tag{E.6}$$

*Proof.* First we plug the quantities in (E.1) and (E.2) into Theorem 3.4. Recall that Theorem 3.4 requires Condition *Statistical-Error*$(\epsilon, \delta/T, \widehat{s}, n/T, \mathcal{B})$. Thus we need to replace $\delta$ and $n$ with $\delta/T$ and $n/T$ in the definition of $\epsilon$ in (E.2). Then we set $\delta = 2 \cdot d^{-1/2}$. Since $T$ is specified in (E.4) and the dimension $d$ is sufficiently large such that $T \leq \sqrt{d}$, we have $\log\left[2/(\delta/T)\right] \leq \log d$ in the definition of $\epsilon$. By (E.3) and (E.5), we can then verify the assumptions in (3.8) and (3.9). Finally, by plugging in $T$ in (E.4) into (3.10) and taking $t = T$, we can verify that in (3.9) the optimization error term is smaller than the statistical error term up to a constant factor. Therefore, we obtain (E.6). $\qquad\square$

To see the statistical rate of convergence with respect to $s^*$, $d$ and $n$, for the moment we assume that $R$, $r$, $\|\boldsymbol{\beta}^*\|_\infty$, $\|\boldsymbol{\beta}^*\|_2$ and $\sigma$ are constants. From (E.3) and (E.4), we obtain $\widehat{s} = C \cdot s^*$ and therefore $\Delta^{\mathrm{GMM}}(s^*) = C' \cdot \sqrt{s^*}$, which implies $T = C''' \cdot \log\big[C'' \cdot \sqrt{n/(s^* \cdot \log d)}\big]$. Hence, by (E.6) we have that, with high probability,

$$\big\|\widehat{\boldsymbol{\beta}} - \boldsymbol{\beta}^*\big\|_2 \leq C \cdot \sqrt{\frac{s^* \cdot \log d \cdot \log n}{n}}.$$

Because the minimax lower bound for estimating an $s^*$-sparse $d$-dimensional vector is $\sqrt{s^* \cdot \log d/n}$, the rate of convergence in (E.6) is optimal up to a factor of $\log n$. Such a logarithmic factor results from the resampling scheme in Algorithm 4, since we only utilize $n/T$ samples within each iteration. We expect that by directly analyzing Algorithm 1 we can eliminate such a logarithmic factor, which however incurs extra technical complexity for the analysis.

**Mixture of Regression Model:** The next lemma, proved by [2], verifies Conditions 3.1 and 3.2 for mixture of regression model. Recall that $\boldsymbol{\beta}^*$ is the regression coefficient and $\sigma$ is the standard deviation of the Gaussian noise.

**Lemma E.4.** Suppose $\|\boldsymbol{\beta}^*\|_2/\sigma \geq r$, where $r > 0$ is a sufficiently large constant that denotes the required minimum signal-to-noise ratio. Conditions *Lipschitz-Gradient*-$1(\gamma_1, \mathcal{B})$, *Lipschitz-Gradient*-$2(\gamma_2, \mathcal{B})$ and *Concavity-Smoothness*$(\mu, \nu, \mathcal{B})$ hold with

$$\gamma_1 \in (0, 1/2), \quad \gamma_2 \in (0, 1/4), \quad \mu = \nu = 1,$$
$$\mathcal{B} = \big\{\boldsymbol{\beta} : \|\boldsymbol{\beta} - \boldsymbol{\beta}^*\|_2 \leq R\big\} \quad \text{with} \quad R = \kappa \cdot \|\boldsymbol{\beta}^*\|_2, \ \kappa = 1/32. \tag{E.7}$$

*Proof.* See the proof of Corollary 3 in [2] for details. $\qquad\square$

The following lemma establishes Condition 3.3 for the two implementations of the M-step.

**Lemma E.5.** For $\mathcal{B}$ specified in (E.7), we have the following results.

- For the maximization implementation of the M-step (line 5 of Algorithm 4), we have that Condition *Statistical-Error*$(\epsilon, \delta, s, n, \mathcal{B})$ holds with

$$\epsilon = C \cdot \Big[\max\big\{\|\boldsymbol{\beta}^*\|_2^2 + \sigma^2, \ 1\big\} + \|\boldsymbol{\beta}^*\|_2\Big] \cdot \sqrt{\frac{\log d + \log(4/\delta)}{n}} \tag{E.8}$$

  for sufficiently large sample size $n$ and constant $C > 0$.

- For the gradient ascent implementation, Condition *Statistical-Error*$(\epsilon, \delta, s, n, \mathcal{B})$ holds with

$$\epsilon = C \cdot \eta \cdot \max\big\{\|\boldsymbol{\beta}^*\|_2^2 + \sigma^2, \ 1, \ \sqrt{s} \cdot \|\boldsymbol{\beta}^*\|_2\big\} \cdot \sqrt{\frac{\log d + \log(4/\delta)}{n}} \tag{E.9}$$

  for sufficiently large sample size $n$ and $C > 0$, where $\eta$ denotes the stepsize in Algorithm 3.

*Proof.* See §H.5 for a detailed proof. $\qquad\square$

The next theorem establishes the implication of Theorem 3.4 for mixture of regression model.

**Theorem E.6.** Let $\|\boldsymbol{\beta}^*\|_2/\sigma \geq r$ with $r > 0$ sufficiently large. Assuming that $\mathcal{B}$ and $R$ are specified in (E.7) and the initialization $\boldsymbol{\beta}^{\mathrm{init}}$ satisfies $\big\|\boldsymbol{\beta}^{\mathrm{init}} - \boldsymbol{\beta}^*\big\|_2 \leq R/2$, we have the following results.

- For the maximization implementation of the M-step (Algorithm 2), let $\widehat{s}$ and $T$ be

$$\widehat{s} = \big\lceil C \cdot \max\big\{16, \ 132/31\big\} \cdot s^* \big\rceil, \quad T = \left\lceil \frac{\log\big\{C' \cdot R/\big[\Delta_1^{\mathrm{MR}}(s^*) \cdot \sqrt{\log d/n}\big]\big\}}{\log\sqrt{2}} \right\rceil,$$

  where $\Delta_1^{\mathrm{MR}}(s^*) = \big(\sqrt{\widehat{s}} + C'' \cdot \sqrt{s^*}\big) \cdot \Big[\max\big\{\|\boldsymbol{\beta}^*\|_2^2 + \sigma^2, \ 1\big\} + \|\boldsymbol{\beta}^*\|_2\Big]$, and $C \geq 1$.

  We suppose $d$ and $n$ are sufficiently large such that $T \leq \sqrt{d}$ and

$$C \cdot \Delta_1^{\mathrm{MR}}(s^*) \cdot \sqrt{\frac{\log d \cdot T}{n}} \leq \min\Big\{\big(1 - 1/\sqrt{2}\big)^2 \cdot R, \ 3/8 \cdot \|\boldsymbol{\beta}^*\|_2\Big\}.$$

  Then with probability at least $1 - 4 \cdot d^{-1/2}$, the final estimator $\widehat{\boldsymbol{\beta}} = \boldsymbol{\beta}^{(T)}$ satisfies

$$\big\|\widehat{\boldsymbol{\beta}} - \boldsymbol{\beta}^*\big\|_2 \leq C' \cdot \Delta_1^{\mathrm{MR}}(s^*) \cdot \sqrt{\frac{\log d \cdot T}{n}}. \tag{E.10}$$

- For the gradient ascent implementation of the M-step (Algorithm 3) with stepsize set to $\eta = 1$, let $\widehat{s}$ and $T$ be

$$\widehat{s} = \big\lceil C \cdot \max\{16/9,\ 132/31\} \cdot s^* \big\rceil, \quad T = \left\lceil \frac{\log\big\{ C' \cdot R / \big[ \Delta_2^{\mathrm{MR}}(s^*) \cdot \sqrt{\log d/n}\,\big] \big\}}{\log 2} \right\rceil,$$

where $\Delta_2^{\mathrm{MR}}(s^*) = \big(\sqrt{\widehat{s}} + C'' \cdot \sqrt{s^*}\big) \cdot \max\big\{ \|\boldsymbol{\beta}^*\|_2^2 + \sigma^2,\ 1,\ \sqrt{\widehat{s}} \cdot \|\boldsymbol{\beta}^*\|_2 \big\}$, and $C \geq 1$.

We suppose $d$ and $n$ are sufficiently large such that $T \leq \sqrt{d}$ and

$$C \cdot \Delta_2^{\mathrm{MR}}(s^*) \cdot \sqrt{\frac{\log d \cdot T}{n}} \leq \min\big\{ R/4,\ 3/8 \cdot \|\boldsymbol{\beta}^*\|_2 \big\}.$$

Then with probability at least $1 - 4 \cdot d^{-1/2}$, the final estimator $\widehat{\boldsymbol{\beta}} = \boldsymbol{\beta}^{(T)}$ satisfies

$$\big\| \widehat{\boldsymbol{\beta}} - \boldsymbol{\beta}^* \big\|_2 \leq C' \cdot \Delta_2^{\mathrm{MR}}(s^*) \cdot \sqrt{\frac{\log d \cdot T}{n}}. \tag{E.11}$$

*Proof.* The proof is similar to Theorem E.3. $\qquad\square$

To understand the intuition of Theorem E.6, we suppose that $\|\boldsymbol{\beta}^*\|_2$, $\sigma$, $R$ and $r$ are constants, which implies $\widehat{s} = C \cdot s^*$ and $\Delta_1^{\mathrm{MR}}(s^*) = C' \cdot \sqrt{s^*}$, $\Delta_2^{\mathrm{MR}}(s^*) = C'' \cdot s^*$. Therefore, for the maximization and gradient ascent implementations of the M-step, we have $T = C' \cdot \log\big[ n/(s^* \cdot \log d) \big]$ and $T = C'' \cdot \log\big\{ n / \big[ (s^*)^2 \cdot \log d \big] \big\}$ correspondingly. Hence, by (E.10) in Theorem E.6 we have that, for the maximization implementation of the M-step,

$$\big\| \widehat{\boldsymbol{\beta}} - \boldsymbol{\beta}^* \big\|_2 \leq C \cdot \sqrt{\frac{s^* \cdot \log d \cdot \log n}{n}} \tag{E.12}$$

holds with high probability. Meanwhile, from (E.11) in Theorem E.6 we have that, for the gradient ascent implementation of the M-step,

$$\big\| \widehat{\boldsymbol{\beta}} - \boldsymbol{\beta}^* \big\|_2 \leq C' \cdot s^* \cdot \sqrt{\frac{\log d \cdot \log n}{n}} \tag{E.13}$$

holds with high probability. The statistical rates in (E.12) and (E.13) attain the $\sqrt{s^* \cdot \log d/n}$ minimax lower bound up to factors of $\sqrt{\log n}$ and $\sqrt{s^* \cdot \log n}$ respectively and are therefore near-optimal. Note that the statistical rate of convergence attained by the gradient ascent implementation of the M-step is slower by a $\sqrt{s^*}$ factor than the rate of the maximization implementation. However, our discussion in §A illustrates that, for mixture of regression model, the gradient ascent implementation does not involve estimating the inverse covariance of $\boldsymbol{X}$ in (A.4). Hence, the gradient ascent implementation is more computationally efficient, and is also applicable to the settings in which $\boldsymbol{X}$ has more general covariance structures.

# F  Implications for Specific Models: Inference

To establish the high dimensional inference results for each model, we only need to verify Conditions 4.1-4.4 and determine the key quantities $\zeta^{\mathrm{EM}}$, $\zeta^{\mathrm{G}}$, $\zeta^{\mathrm{T}}$ and $\zeta^{\mathrm{L}}$. In the following, we focus on Gaussian mixture and mixture of regression models.

**Gaussian Mixture Model:** The following lemma verifies Conditions 4.1 and 4.2.

**Lemma F.1.** We have that Conditions 4.1 and 4.2 hold with

$$\zeta^{\mathrm{EM}} = \frac{\sqrt{\widehat{s}} \cdot \Delta^{\mathrm{GMM}}(s^*)}{1 - \exp\big(-C \cdot r^2/2\big)} \cdot \sqrt{\frac{\log d \cdot T}{n}}, \quad \text{and} \quad \zeta^{\mathrm{G}} = \big( \|\boldsymbol{\beta}^*\|_\infty + \sigma \big) \cdot \sqrt{\frac{\log d}{n}},$$

where $\widehat{s}$, $\Delta^{\mathrm{GMM}}(s^*)$, $r$ and $T$ are as defined in Theorem E.3.

*Proof.* See §I.5 for a detailed proof. $\qquad\square$

By our discussion that follows Theorem E.3, we have that $\widehat{s}$ and $\Delta^{\mathrm{GMM}}(s^*)$ are of the same order as $s^*$ and $\sqrt{s^*}$ respectively, and $T$ is roughly of the order $\sqrt{\log n}$. Therefore, $\zeta^{\mathrm{EM}}$ is roughly of the order $s^* \cdot \sqrt{\log d/n \cdot \log n}$. The following lemma verifies Condition 4.3 for Gaussian mixture model.

**Lemma F.2.** We have that Condition 4.3 holds with
$$\zeta^{\mathrm{T}} = \big(\|\boldsymbol{\beta}^*\|_\infty^2 + \sigma^2\big)/\sigma^2 \cdot \sqrt{\frac{\log d}{n}}.$$

*Proof.* See §I.6 for a detailed proof. □

The following lemma establishes Condition 4.4 for Gaussian mixture model.

**Lemma F.3.** We have that Condition 4.4 holds with
$$\zeta^{\mathrm{L}} = \big(\|\boldsymbol{\beta}^*\|_\infty^2 + \sigma^2\big)^{3/2}/\sigma^4 \cdot \big(\log d + \log n\big)^{3/2}.$$

*Proof.* See §I.7 for a detailed proof. □

Equipped with Lemmas F.1-F.3, we establish the inference results for Gaussian mixture model.

**Theorem F.4.** Under Assumption 4.5, we have that for $n \to \infty$, (4.7) holds for Gaussian mixture model.

In fact, for Gaussian mixture model we can make (4.6) in Assumption 4.5 more transparent by plugging in $\zeta^{\mathrm{EM}}$, $\zeta^{\mathrm{G}}$, $\zeta^{\mathrm{T}}$ and $\zeta^{\mathrm{L}}$ specified above. Particularly, for simplicity we assume all quantities except $s_{\mathbf{w}}^*$, $s^*$, $d$ and $n$ are constants. Then we can verify that (4.6) holds if
$$\max\big\{s_{\mathbf{w}}^*, \ s^*\big\}^2 \cdot (s^*)^2 \cdot (\log d)^5 = o\big[n/(\log n)^2\big]. \tag{F.1}$$

According to the discussion following Theorem E.3, we require $s^* \cdot \log d = o(n/\log n)$ for the estimator $\widehat{\boldsymbol{\beta}}$ to be consistent. In comparison, (F.1) illustrates that high dimensional inference requires a higher sample complexity than parameter estimation. In the context of high dimensional generalized linear models, [26, 32] also observe the same phenomenon.

**Mixture of Regression Model:** The following lemma verifies Conditions 4.1 and 4.2. Recall that $\widehat{s}$, $T$, $\Delta_1^{\mathrm{MR}}(s^*)$ and $\Delta_2^{\mathrm{MR}}(s^*)$ are defined in Theorem E.6, while $\sigma$ denotes the standard deviation of the Gaussian noise in mixture of regression model.

**Lemma F.5.** We have that Conditions 4.1 and 4.2 hold with
$$\zeta^{\mathrm{EM}} = \sqrt{\widehat{s}} \cdot \Delta^{\mathrm{MR}}(s^*) \cdot \sqrt{\frac{\log d \cdot T}{n}}, \quad \text{and} \ \ \zeta^{\mathrm{G}} = \max\Big\{\|\boldsymbol{\beta}^*\|_2^2 + \sigma^2, \ 1, \ \sqrt{s^*} \cdot \|\boldsymbol{\beta}^*\|_2\Big\} \cdot \sqrt{\frac{\log d}{n}},$$
where we have $\Delta^{\mathrm{MR}}(s^*) = \Delta_1^{\mathrm{MR}}(s^*)$ for the maximization implementation of the M-step (Algorithm 2), and $\Delta^{\mathrm{MR}}(s^*) = \Delta_2^{\mathrm{MR}}(s^*)$ for the gradient ascent implementation of the M-step (Algorithm 3).

*Proof.* See §I.8 for a detailed proof. □

By our discussion that follows Theorem E.6, we have that $\widehat{s}$ is of the same order as $s^*$. For the maximization implementation of the M-step (Algorithm 2), we have that $\Delta^{\mathrm{MR}}(s^*) = \Delta_1^{\mathrm{MR}}(s^*)$ is of the same order as $\sqrt{s^*}$. Meanwhile, for the gradient ascent implementation in Algorithm 3, we have that $\Delta^{\mathrm{MR}}(s^*) = \Delta_2^{\mathrm{MR}}(s^*)$ is of the same order as $s^*$. Hence, $\zeta^{\mathrm{EM}}$ is of the order $s^* \cdot \sqrt{\log d/n \cdot \log n}$ or $(s^*)^{3/2} \cdot \sqrt{\log d/n \cdot \log n}$ correspondingly, since $T$ is roughly of the order $\sqrt{\log n}$. The next lemma establishes Condition 4.3 for mixture of regression model.

**Lemma F.6.** We have that Condition 4.3 holds with
$$\zeta^{\mathrm{T}} = \big(\log n + \log d\big) \cdot \big[\big(\log n + \log d\big) \cdot \|\boldsymbol{\beta}^*\|_1^2 + \sigma^2\big]/\sigma^2 \cdot \sqrt{\frac{\log d}{n}}.$$

*Proof.* See §I.9 for a detailed proof. □

The following lemma establishes Condition 4.4 for mixture of regression model.

**Lemma F.7.** We have that Condition 4.4 holds with
$$\zeta^{\mathrm{L}} = \big(\|\boldsymbol{\beta}^*\|_1 + \sigma\big)^3 \cdot \big(\log n + \log d\big)^3/\sigma^4.$$

*Proof.* See §I.10 for a detailed proof. □

Equipped with Lemmas F.5-F.7, we are now ready to establish the high dimensional inference results for mixture of regression model.

**Theorem F.8.** For mixture of regression model, under Assumption 4.5, (4.7) holds as $n \to \infty$.

Similar to the discussion that follows Theorem F.4, we can make (4.6) in Assumption 4.5 more explicit by plugging in $\zeta^{\mathrm{EM}}, \zeta^{\mathrm{G}}, \zeta^{\mathrm{T}}$ and $\zeta^{\mathrm{L}}$ specified in Lemmas F.5-F.7. Assuming all quantities except $s_{\mathbf{w}}^*, s^*, d$ and $n$ are constants, we have that (4.6) holds if

$$\max\{s_{\mathbf{w}}^*, \ s^*\}^2 \cdot (s^*)^4 \cdot (\log d)^8 = o\big[n/(\log n)^2\big].$$

In contrast, for high dimensional estimation, we only require $(s^*)^2 \cdot \log d = o(n/\log n)$ to ensure the consistency of $\widehat{\boldsymbol{\beta}}$ by our discussion following Theorem E.6.

# G    Proof of Main Results

We lay out a proof sketch of the main theory. First we prove the results in Theorem 3.4 for parameter estimation and computation. Then we establish the results in Theorem 4.6 for inference.

## G.1    Proof of Results for Computation and Estimation

**Proof of Theorem 3.4:** First we introduce some notations. Recall that the $\mathrm{trunc}(\cdot, \cdot)$ function is defined in (2.3). We define $\overline{\boldsymbol{\beta}}^{(t+0.5)}, \overline{\boldsymbol{\beta}}^{(t+1)} \in \mathbb{R}^d$ as

$$\overline{\boldsymbol{\beta}}^{(t+0.5)} = M\big(\boldsymbol{\beta}^{(t)}\big), \qquad \overline{\boldsymbol{\beta}}^{(t+1)} = \mathrm{trunc}\big(\overline{\boldsymbol{\beta}}^{(t+0.5)}, \widehat{\mathcal{S}}^{(t+0.5)}\big). \tag{G.1}$$

As defined in (3.1) or (3.2), $M(\cdot)$ is the population version M-step with the maximization or gradient ascent implementation. Here $\widehat{\mathcal{S}}^{(t+0.5)}$ denotes the set of index $j$'s with the top $\widehat{s}$ largest $\big|\beta_j^{(t+0.5)}\big|$'s. It is worth noting $\widehat{\mathcal{S}}^{(t+0.5)}$ is calculated based on $\boldsymbol{\beta}^{(t+0.5)}$ in the truncation step (line 6 of Algorithm 4), rather than based on $\overline{\boldsymbol{\beta}}^{(t+0.5)}$ defined in (G.1).

Our goal is to characterize the relationship between $\big\|\boldsymbol{\beta}^{(t+1)} - \boldsymbol{\beta}^*\big\|_2$ and $\big\|\boldsymbol{\beta}^{(t)} - \boldsymbol{\beta}^*\big\|_2$. According to the definition of the truncation step (line 6 of Algorithm 4) and triangle inequality, we have

$$\big\|\boldsymbol{\beta}^{(t+1)} - \boldsymbol{\beta}^*\big\|_2 = \Big\|\mathrm{trunc}\big(\boldsymbol{\beta}^{(t+0.5)}, \widehat{\mathcal{S}}^{(t+0.5)}\big) - \boldsymbol{\beta}^*\Big\|_2$$

$$\leq \Big\|\mathrm{trunc}\big(\boldsymbol{\beta}^{(t+0.5)}, \widehat{\mathcal{S}}^{(t+0.5)}\big) - \mathrm{trunc}\big(\overline{\boldsymbol{\beta}}^{(t+0.5)}, \widehat{\mathcal{S}}^{(t+0.5)}\big)\Big\|_2 + \Big\|\mathrm{trunc}\big(\overline{\boldsymbol{\beta}}^{(t+0.5)}, \widehat{\mathcal{S}}^{(t+0.5)}\big) - \boldsymbol{\beta}^*\Big\|_2$$

$$= \underbrace{\Big\|\mathrm{trunc}\big(\boldsymbol{\beta}^{(t+0.5)}, \widehat{\mathcal{S}}^{(t+0.5)}\big) - \mathrm{trunc}\big(\overline{\boldsymbol{\beta}}^{(t+0.5)}, \widehat{\mathcal{S}}^{(t+0.5)}\big)\Big\|_2}_{(\mathrm{i})} + \underbrace{\big\|\overline{\boldsymbol{\beta}}^{(t+1)} - \boldsymbol{\beta}^*\big\|_2}_{(\mathrm{ii})}, \tag{G.2}$$

where the last equality is obtained from (G.1). According to the definition of the $\mathrm{trunc}(\cdot, \cdot)$ function in (2.3), the two terms within term (i) both have support $\widehat{\mathcal{S}}^{(t+0.5)}$ with cardinality $\widehat{s}$. Thus, we have

$$\Big\|\mathrm{trunc}\big(\boldsymbol{\beta}^{(t+0.5)}, \widehat{\mathcal{S}}^{(t+0.5)}\big) - \mathrm{trunc}\big(\overline{\boldsymbol{\beta}}^{(t+0.5)}, \widehat{\mathcal{S}}^{(t+0.5)}\big)\Big\|_2 = \Big\|\big(\boldsymbol{\beta}^{(t+0.5)} - \overline{\boldsymbol{\beta}}^{(t+0.5)}\big)_{\widehat{\mathcal{S}}^{(t+0.5)}}\Big\|_2$$

$$\leq \sqrt{\widehat{s}} \cdot \Big\|\big(\boldsymbol{\beta}^{(t+0.5)} - \overline{\boldsymbol{\beta}}^{(t+0.5)}\big)_{\widehat{\mathcal{S}}^{(t+0.5)}}\Big\|_\infty$$

$$\leq \sqrt{\widehat{s}} \cdot \big\|\boldsymbol{\beta}^{(t+0.5)} - \overline{\boldsymbol{\beta}}^{(t+0.5)}\big\|_\infty. \tag{G.3}$$

Since we have $\boldsymbol{\beta}^{(t+0.5)} = M_n\big(\boldsymbol{\beta}^{(t)}\big)$ and $\overline{\boldsymbol{\beta}}^{(t+0.5)} = M\big(\boldsymbol{\beta}^{(t)}\big)$, we can further establish an upper bound for the right-hand side by invoking Condition 3.3.

Our subsequent proof will establish an upper bound for term (ii) in (G.2) in two steps. We first characterize the relationship between $\big\|\overline{\boldsymbol{\beta}}^{(t+1)} - \boldsymbol{\beta}^*\big\|_2$ and $\big\|\overline{\boldsymbol{\beta}}^{(t+0.5)} - \boldsymbol{\beta}^*\big\|_2$ and then the relationship between $\big\|\overline{\boldsymbol{\beta}}^{(t+0.5)} - \boldsymbol{\beta}^*\big\|_2$ and $\big\|\boldsymbol{\beta}^{(t)} - \boldsymbol{\beta}^*\big\|_2$. The next lemma accomplishes the first step. Recall that $\widehat{s}$ is the sparsity parameter in Algorithm 4, while $s^*$ is the sparsity level of the true parameter $\boldsymbol{\beta}^*$.

**Lemma G.1.** Suppose that we have

$$\big\|\overline{\boldsymbol{\beta}}^{(t+0.5)} - \boldsymbol{\beta}^*\big\|_2 \leq \kappa \cdot \|\boldsymbol{\beta}^*\|_2 \tag{G.4}$$

for some $\kappa \in (0, 1)$. Assuming that we have

$$\widehat{s} \geq \frac{4 \cdot (1+\kappa)^2}{(1-\kappa)^2} \cdot s^*, \ \ \text{and} \ \ \sqrt{\widehat{s}} \cdot \big\|\boldsymbol{\beta}^{(t+0.5)} - \overline{\boldsymbol{\beta}}^{(t+0.5)}\big\|_\infty \leq \frac{(1-\kappa)^2}{2 \cdot (1+\kappa)} \cdot \|\boldsymbol{\beta}^*\|_2, \tag{G.5}$$

then it holds that

$$\left\|\overline{\boldsymbol{\beta}}^{(t+1)} - \boldsymbol{\beta}^*\right\|_2 \leq \frac{C \cdot \sqrt{s^*}}{\sqrt{1-\kappa}} \cdot \left\|\boldsymbol{\beta}^{(t+0.5)} - \overline{\boldsymbol{\beta}}^{(t+0.5)}\right\|_\infty + \left(1 + 4 \cdot \sqrt{s^*/\widehat{s}}\right)^{1/2} \cdot \left\|\overline{\boldsymbol{\beta}}^{(t+0.5)} - \boldsymbol{\beta}^*\right\|_2.$$
(G.6)

*Proof.* The proof is based on fine-grained analysis of the relationship between $\widehat{\mathcal{S}}^{(t+0.5)}$ and the true support $\mathcal{S}^*$. In particular, we focus on three index sets, namely, $\mathcal{I}_1 = \mathcal{S}^* \backslash \widehat{\mathcal{S}}^{(t+0.5)}$, $\mathcal{I}_2 = \mathcal{S}^* \cap \widehat{\mathcal{S}}^{(t+0.5)}$ and $\mathcal{I}_3 = \widehat{\mathcal{S}}^{(t+0.5)} \backslash \mathcal{S}^*$. Among them, $\mathcal{I}_2$ characterizes the similarity between $\widehat{\mathcal{S}}^{(t+0.5)}$ and $\mathcal{S}^*$, while $\mathcal{I}_1$ and $\mathcal{I}_3$ characterize their difference. The key proof strategy is to represent the three distances in (G.6) with the $\ell_2$-norms of the restrictions of $\overline{\boldsymbol{\beta}}^{(t+0.5)}$ and $\boldsymbol{\beta}^*$ on the three index sets. In particular, we focus on $\left\|\overline{\boldsymbol{\beta}}_{\mathcal{I}_1}^{(t+0.5)}\right\|_2$ and $\left\|\boldsymbol{\beta}_{\mathcal{I}_1}^*\right\|_2$. In order to quantify these $\ell_2$-norms, we establish a fine-grained characterization for the absolute values of $\overline{\boldsymbol{\beta}}^{(t+0.5)}$'s entries that are selected and eliminated within the truncation operation $\overline{\boldsymbol{\beta}}^{(t+1)} \leftarrow \text{trunc}(\overline{\boldsymbol{\beta}}^{(t+0.5)}, \widehat{\mathcal{S}}^{(t+0.5)})$. See §H.1 for a detailed proof. □

Lemma G.1 is central to the proof of Theorem 3.4. In detail, the assumption in (G.4) guarantees $\overline{\boldsymbol{\beta}}^{(t+0.5)}$ is within the basin of attraction. In (G.5), the first assumption ensures the sparsity parameter $\widehat{s}$ in Algorithm 4 is set to be sufficiently large, while second ensures the statistical error is sufficiently small. These assumptions will be verified in the proof of Theorem 3.4. The intuition behind (G.6) is as follows. Let $\overline{\mathcal{S}}^{(t+0.5)} = \text{supp}(\overline{\boldsymbol{\beta}}^{(t+0.5)}, \widehat{s})$, where $\text{supp}(\cdot, \cdot)$ is defined in (2.2). By triangle inequality, the left-hand side of (G.6) satisfies

$$\left\|\overline{\boldsymbol{\beta}}^{(t+1)} - \boldsymbol{\beta}^*\right\|_2 \leq \underbrace{\left\|\overline{\boldsymbol{\beta}}^{(t+1)} - \text{trunc}(\overline{\boldsymbol{\beta}}^{(t+0.5)}, \overline{\mathcal{S}}^{(t+0.5)})\right\|_2}_{\text{(i)}} + \underbrace{\left\|\text{trunc}(\overline{\boldsymbol{\beta}}^{(t+0.5)}, \overline{\mathcal{S}}^{(t+0.5)}) - \boldsymbol{\beta}^*\right\|_2}_{\text{(ii)}}.$$
(G.7)

Intuitively, the two terms on right-hand side of (G.6) reflect terms (i) and (ii) in (G.7) correspondingly. In detail, for term (i) in (G.7), recall that according to (G.1) and line 6 of Algorithm 4 we have

$$\overline{\boldsymbol{\beta}}^{(t+1)} = \text{trunc}(\overline{\boldsymbol{\beta}}^{(t+0.5)}, \widehat{\mathcal{S}}^{(t+0.5)}), \quad \text{where } \widehat{\mathcal{S}}^{(t+0.5)} = \text{supp}(\boldsymbol{\beta}^{(t+0.5)}, \widehat{s}).$$

As the sample size $n$ is sufficiently large, $\overline{\boldsymbol{\beta}}^{(t+0.5)}$ and $\boldsymbol{\beta}^{(t+0.5)}$ are close, since they are attained by the population version and sample version M-steps correspondingly. Hence, $\overline{\mathcal{S}}^{(t+0.5)} = \text{supp}(\overline{\boldsymbol{\beta}}^{(t+0.5)}, \widehat{s})$ and $\widehat{\mathcal{S}}^{(t+0.5)} = \text{supp}(\boldsymbol{\beta}^{(t+0.5)}, \widehat{s})$ should be similar. Thus, in term (i), $\overline{\boldsymbol{\beta}}^{(t+1)} = \text{trunc}(\overline{\boldsymbol{\beta}}^{(t+0.5)}, \widehat{\mathcal{S}}^{(t+0.5)})$ should be close to $\text{trunc}(\overline{\boldsymbol{\beta}}^{(t+0.5)}, \overline{\mathcal{S}}^{(t+0.5)})$ up to some statistical error, which is reflected by the first term on the right-hand side of (G.6).

Also, we turn to quantify the relationship between $\left\|\overline{\boldsymbol{\beta}}^{(t+0.5)} - \boldsymbol{\beta}^*\right\|_2$ in (G.6) and term (ii) in (G.7). The truncation in term (ii) preserves the top $\widehat{s}$ coordinates of $\overline{\boldsymbol{\beta}}^{(t+0.5)}$ with the largest magnitudes while setting others to zero. Intuitively speaking, the truncation incurs additional error to $\overline{\boldsymbol{\beta}}^{(t+0.5)}$'s distance to $\boldsymbol{\beta}^*$. Meanwhile, note that when $\overline{\boldsymbol{\beta}}^{(t+0.5)}$ is close to $\boldsymbol{\beta}^*$, $\overline{\mathcal{S}}^{(t+0.5)}$ is similar to $\mathcal{S}^*$. Therefore, the incurred error can be controlled, because the truncation does not eliminate many relevant entries. In particular, as shown in the second term on the right-hand side of (G.6), such incurred error decays as $\widehat{s}$ increases, since in this case $\widehat{\mathcal{S}}^{(t+0.5)}$ includes more entries. According to the discussion for term (i) in (G.7), $\overline{\mathcal{S}}^{(t+0.5)}$ is similar to $\widehat{\mathcal{S}}^{(t+0.5)}$, which implies that $\overline{\mathcal{S}}^{(t+0.5)}$ should also cover more entries. Thus, fewer relevant entries are wrongly eliminated by the truncation and hence the incurred error is smaller. The extreme case is that, when $\widehat{s} \to \infty$, term (ii) in (G.7) becomes $\left\|\overline{\boldsymbol{\beta}}^{(t+0.5)} - \boldsymbol{\beta}^*\right\|_2$, which indicates that no additional error is incurred by the truncation. Correspondingly, the second term on the right-hand side of (G.6) also becomes $\left\|\overline{\boldsymbol{\beta}}^{(t+0.5)} - \boldsymbol{\beta}^*\right\|_2$.

Next, we turn to characterize the relationship between $\left\|\overline{\boldsymbol{\beta}}^{(t+0.5)} - \boldsymbol{\beta}^*\right\|_2$ and $\left\|\boldsymbol{\beta}^{(t)} - \boldsymbol{\beta}^*\right\|_2$. Recall $\overline{\boldsymbol{\beta}}^{(t+0.5)} = M(\boldsymbol{\beta}^{(t)})$ is defined in (G.1). The next lemma, which is adapted from Theorems 1 and 3 in [2], characterizes the contraction property of the population version M-step defined in (3.1) or (3.2).

**Lemma G.2.** Under the assumptions of Theorem 3.4, the following results hold for $\boldsymbol{\beta}^{(t)} \in \mathcal{B}$.
- For the maximization implementation of the M-step (Algorithm 2), we have

$$\left\|\overline{\boldsymbol{\beta}}^{(t+0.5)} - \boldsymbol{\beta}^*\right\|_2 \leq (\gamma_1/\nu) \cdot \left\|\boldsymbol{\beta}^{(t)} - \boldsymbol{\beta}^*\right\|_2.$$
(G.8)

- For the gradient ascent implementation of the M-step (Algorithm 3), we have

$$\left\|\overline{\boldsymbol{\beta}}^{(t+0.5)} - \boldsymbol{\beta}^*\right\|_2 \leq \left(1 - 2 \cdot \frac{\nu - \gamma_2}{\nu + \mu}\right) \cdot \left\|\boldsymbol{\beta}^{(t)} - \boldsymbol{\beta}^*\right\|_2. \tag{G.9}$$

Here $\gamma_1$, $\gamma_2$, $\mu$ and $\nu$ are defined in Conditions 3.1 and 3.2.

*Proof.* The proof strategy is to first characterize the M-step using $Q(\cdot; \boldsymbol{\beta}^*)$. According to Condition *Concavity-Smoothness*$(\mu, \nu, \mathcal{B})$, $-Q(\cdot; \boldsymbol{\beta}^*)$ is $\nu$-strongly convex and $\mu$-smooth, and thus enjoys desired optimization guarantees. Then Condition *Lipschitz-Gradient*-$1(\gamma_1, \mathcal{B})$ or *Lipschitz-Gradient*-$2(\gamma_2, \mathcal{B})$ is invoked to characterize the difference between $Q(\cdot; \boldsymbol{\beta}^*)$ and $Q(\cdot; \boldsymbol{\beta}^{(t)})$. We provide the proof in §H.2 for the sake of completeness. □

Equipped with Lemmas G.1 and G.2, we are now ready to prove Theorem 3.4.

*Proof.* To unify the subsequent proof for the maximization and gradient implementations of the M-step, we employ $\rho \in (0, 1)$ to denote $\rho_1 := \gamma_1/\nu$ in (G.8) or $\rho_2 := 1 - 2 \cdot (\nu - \gamma_2)/(\nu + \mu)$ in (G.9). In the following we stick to the former one to avoid confusion. The proof for the letter one is exactly the same. By the definitions of $\overline{\boldsymbol{\beta}}^{(t+0.5)}$ and $\boldsymbol{\beta}^{(t+0.5)}$ in (G.1) and Algorithm 4, Condition *Statistical-Error*$(\epsilon, \delta/T, \widehat{s}, n/T, \mathcal{B})$ implies

$$\left\|\boldsymbol{\beta}^{(t+0.5)} - \overline{\boldsymbol{\beta}}^{(t+0.5)}\right\|_\infty = \left\|M_{n/T}(\boldsymbol{\beta}^{(t)}) - M(\boldsymbol{\beta}^{(t)})\right\|_\infty \leq \epsilon$$

holds with probability at least $1 - \delta/T$. Then by taking union bound we have that, the event

$$\mathcal{E} = \left\{\left\|\boldsymbol{\beta}^{(t+0.5)} - \overline{\boldsymbol{\beta}}^{(t+0.5)}\right\|_\infty \leq \epsilon, \text{ for all } t \in \{0, \dots, T-1\}\right\} \tag{G.10}$$

occurs with probability at least $1 - \delta$. Conditioning on $\mathcal{E}$, in the following we prove that

$$\left\|\boldsymbol{\beta}^{(t)} - \boldsymbol{\beta}^*\right\|_2 \leq \frac{(\sqrt{\widehat{s}} + C/\sqrt{1-\kappa} \cdot \sqrt{s^*}) \cdot \epsilon}{1 - \sqrt{\rho}} + \rho^{t/2} \cdot \left\|\boldsymbol{\beta}^{(0)} - \boldsymbol{\beta}^*\right\|_2, \quad \text{for all } t \in \{1, \dots, T\} \tag{G.11}$$

by mathematical induction.

Before we lay out the proof, we first prove $\boldsymbol{\beta}^{(0)} \in \mathcal{B}$. Recall $\boldsymbol{\beta}^{\text{init}}$ is the initialization of Algorithm 4 and $R$ is the radius of the basin of attraction $\mathcal{B}$. By the assumption of Theorem 3.4, we have

$$\left\|\boldsymbol{\beta}^{\text{init}} - \boldsymbol{\beta}^*\right\|_2 \leq R/2. \tag{G.12}$$

Therefore, (G.12) implies $\left\|\boldsymbol{\beta}^{\text{init}} - \boldsymbol{\beta}^*\right\|_2 < \kappa \cdot \|\boldsymbol{\beta}^*\|_2$ since $R = \kappa \cdot \|\boldsymbol{\beta}^*\|_2$. Invoking the auxiliary result in Lemma H.1, we obtain

$$\left\|\boldsymbol{\beta}^{(0)} - \boldsymbol{\beta}^*\right\|_2 \leq \left(1 + 4 \cdot \sqrt{s^*/\widehat{s}}\right)^{1/2} \cdot \left\|\boldsymbol{\beta}^{\text{init}} - \boldsymbol{\beta}^*\right\|_2 \leq \left(1 + 4 \cdot \sqrt{1/4}\right)^{1/2} \cdot R/2 < R. \tag{G.13}$$

Here the second inequality is from (G.12) as well as the assumption in (3.8), which implies $s^*/\widehat{s} \leq (1-\kappa)^2/[4 \cdot (1+\kappa)^2] \leq 1/4$. Thus, (G.13) implies $\boldsymbol{\beta}^{(0)} \in \mathcal{B}$. In the sequel, we prove that (G.11) holds for $t = 1$. By invoking Lemma G.2 and setting $t = 0$ in (G.8), we obtain

$$\left\|\overline{\boldsymbol{\beta}}^{(0.5)} - \boldsymbol{\beta}^*\right\|_2 \leq \rho \cdot \left\|\boldsymbol{\beta}^{(0)} - \boldsymbol{\beta}^*\right\|_2 \leq \rho \cdot R < R = \kappa \cdot \|\boldsymbol{\beta}^*\|_2,$$

where the second inequality is from (G.13). Hence, the assumption in (G.4) of Lemma G.1 holds for $\overline{\boldsymbol{\beta}}^{(0.5)}$. Furthermore, by the assumptions in (3.8) and (3.9) of Theorem 3.4, we can also verify that the assumptions in (G.5) of Lemma G.1 hold conditioning on the event $\mathcal{E}$ defined in (G.10). By invoking Lemma G.1 we have that (G.6) holds for $t = 0$. Further plugging $\left\|\boldsymbol{\beta}^{(t+0.5)} - \overline{\boldsymbol{\beta}}^{(t+0.5)}\right\|_\infty \leq \epsilon$ in (G.10) into (G.6) with $t = 0$, we obtain

$$\left\|\overline{\boldsymbol{\beta}}^{(1)} - \boldsymbol{\beta}^*\right\|_2 \leq \frac{C \cdot \sqrt{s^*}}{\sqrt{1-\kappa}} \cdot \epsilon + \left(1 + 4 \cdot \sqrt{s^*/\widehat{s}}\right)^{1/2} \cdot \left\|\overline{\boldsymbol{\beta}}^{(0.5)} - \boldsymbol{\beta}^*\right\|_2. \tag{G.14}$$

Setting $t = 0$ in (G.8) of Lemma G.2 and then plugging (G.8) into (G.14), we obtain

$$\left\|\overline{\boldsymbol{\beta}}^{(1)} - \boldsymbol{\beta}^*\right\|_2 \leq \frac{C \cdot \sqrt{s^*}}{\sqrt{1-\kappa}} \cdot \epsilon + \left(1 + 4 \cdot \sqrt{s^*/\widehat{s}}\right)^{1/2} \cdot \rho \cdot \left\|\boldsymbol{\beta}^{(0)} - \boldsymbol{\beta}^*\right\|_2. \tag{G.15}$$

For $t = 0$, plugging (G.3) into term (i) in (G.2), and (G.15) into term (ii) in (G.2), and then applying $\left\|\boldsymbol{\beta}^{(t+0.5)} - \overline{\boldsymbol{\beta}}^{(t+0.5)}\right\|_\infty \leq \epsilon$ with $t = 0$ in (G.10), we obtain

$$\left\|\boldsymbol{\beta}^{(1)} - \boldsymbol{\beta}^*\right\|_2 \leq \sqrt{\widehat{s}} \cdot \left\|\boldsymbol{\beta}^{(0.5)} - \overline{\boldsymbol{\beta}}^{(0.5)}\right\|_\infty + \frac{C \cdot \sqrt{s^*}}{\sqrt{1 - \kappa}} \cdot \epsilon + \left(1 + 4 \cdot \sqrt{s^*/\widehat{s}}\right)^{1/2} \cdot \rho \cdot \left\|\boldsymbol{\beta}^{(0)} - \boldsymbol{\beta}^*\right\|_2$$

$$\leq \left(\sqrt{\widehat{s}} + C/\sqrt{1 - \kappa} \cdot \sqrt{s^*}\right) \cdot \epsilon + \left(1 + 4 \cdot \sqrt{s^*/\widehat{s}}\right)^{1/2} \cdot \rho \cdot \left\|\boldsymbol{\beta}^{(0)} - \boldsymbol{\beta}^*\right\|_2. \quad \text{(G.16)}$$

By our assumption that $\widehat{s} \geq 16 \cdot (1/\rho - 1)^{-2} \cdot s^*$ in (3.8), we have $\left(1 + 4 \cdot \sqrt{s^*/\widehat{s}}\right)^{1/2} \leq 1/\sqrt{\rho}$ in (G.16). Hence, from (G.16) we obtain

$$\left\|\boldsymbol{\beta}^{(1)} - \boldsymbol{\beta}^*\right\|_2 \leq \left(\sqrt{\widehat{s}} + C/\sqrt{1 - \kappa} \cdot \sqrt{s^*}\right) \cdot \epsilon + \sqrt{\rho} \cdot \left\|\boldsymbol{\beta}^{(0)} - \boldsymbol{\beta}^*\right\|_2, \quad \text{(G.17)}$$

which implies that (G.11) holds for $t = 1$, since we have $1 - \sqrt{\rho} < 1$ in (G.11).
Suppose we have that (G.11) holds for some $t \geq 1$. By (G.11) we have

$$\left\|\boldsymbol{\beta}^{(t)} - \boldsymbol{\beta}^*\right\|_2 \leq \frac{\left(\sqrt{\widehat{s}} + C/\sqrt{1 - \kappa} \cdot \sqrt{s^*}\right) \cdot \epsilon}{1 - \sqrt{\rho}} + \rho^{t/2} \cdot \left\|\boldsymbol{\beta}^{(0)} - \boldsymbol{\beta}^*\right\|_2$$

$$\leq \left(1 - \sqrt{\rho}\right) \cdot R + \sqrt{\rho} \cdot R = R, \quad \text{(G.18)}$$

where the second inequality is from (G.13) and our assumption $\left(\sqrt{\widehat{s}} + C/\sqrt{1 - \kappa} \cdot \sqrt{s^*}\right) \cdot \epsilon \leq \left(1 - \sqrt{\rho}\right)^2 \cdot R$ in (3.9). Therefore, by (G.18) we have $\boldsymbol{\beta}^{(t)} \in \mathcal{B}$. Then (G.8) in Lemma G.2 implies

$$\left\|\overline{\boldsymbol{\beta}}^{(t+0.5)} - \boldsymbol{\beta}^*\right\|_2 \leq \rho \cdot \left\|\boldsymbol{\beta}^{(t)} - \boldsymbol{\beta}^*\right\|_2 \leq \rho \cdot R < R = \kappa \cdot \|\boldsymbol{\beta}^*\|_2,$$

where the third inequality is from $\rho \in (0, 1)$. Following the same proof for (G.17), we obtain

$$\left\|\boldsymbol{\beta}^{(t+1)} - \boldsymbol{\beta}^*\right\|_2 \leq \left(\sqrt{\widehat{s}} + C/\sqrt{1 - \kappa} \cdot \sqrt{s^*}\right) \cdot \epsilon + \sqrt{\rho} \cdot \left\|\boldsymbol{\beta}^{(t)} - \boldsymbol{\beta}^*\right\|_2$$

$$\leq \left(1 + \frac{\sqrt{\rho}}{1 - \sqrt{\rho}}\right) \cdot \left(\sqrt{\widehat{s}} + C/\sqrt{1 - \kappa} \cdot \sqrt{s^*}\right) \cdot \epsilon + \sqrt{\rho} \cdot \rho^{t/2} \cdot \left\|\boldsymbol{\beta}^{(0)} - \boldsymbol{\beta}^*\right\|_2$$

$$= \frac{\left(\sqrt{\widehat{s}} + C/\sqrt{1 - \kappa} \cdot \sqrt{s^*}\right) \cdot \epsilon}{1 - \sqrt{\rho}} + \rho^{(t+1)/2} \cdot \left\|\boldsymbol{\beta}^{(0)} - \boldsymbol{\beta}^*\right\|_2.$$

Here the second inequality is obtained by plugging in (G.11) for $t$. Hence we have that (G.11) holds for $t + 1$. By induction, we conclude that (G.11) holds conditioning on the event $\mathcal{E}$ defined in (G.10), which occurs with probability at least $1 - \delta$. By plugging the specific definitions of $\rho$ into (G.11), and applying $\left\|\boldsymbol{\beta}^{(0)} - \boldsymbol{\beta}^*\right\|_2 \leq R$ in (G.13) to the right-hand side of (G.11), we obtain (3.10). The results of the gradient descent implementation follows from the same proof with $\rho = \rho_2$. $\qquad\square$

## G.2 Proof of Results for Inference

**Proof of Theorem 4.6:** We establish the asymptotic normality of the decorrelated score statistic defined in (2.7) in two steps. We first prove the asymptotic normality of $\sqrt{n} \cdot S_n(\widehat{\boldsymbol{\beta}}_0, \lambda)$, where $\widehat{\boldsymbol{\beta}}_0$ is defined in (2.7) and $S_n(\cdot, \cdot)$ is defined in (2.5). Then we prove that $-\left[T_n(\widehat{\boldsymbol{\beta}}_0)\right]_{\alpha|\gamma}$ defined in (2.7) is a consistent estimator of $\sqrt{n} \cdot S_n(\widehat{\boldsymbol{\beta}}_0, \lambda)$'s asymptotic variance. The next lemma accomplishes the first step. Recall $I(\boldsymbol{\beta}^*) = -\mathbb{E}_{\boldsymbol{\beta}^*}\left[\nabla^2 \ell_n(\boldsymbol{\beta}^*)\right]/n$ is the Fisher information for $\ell_n(\boldsymbol{\beta}^*)$ defined in (B.3).

**Lemma G.3.** Under the assumptions of Theorem 4.6, we have that for $n \to \infty$,

$$\sqrt{n} \cdot S_n(\widehat{\boldsymbol{\beta}}_0, \lambda) \xrightarrow{D} N\left(0, \left[I(\boldsymbol{\beta}^*)\right]_{\alpha|\gamma}\right),$$

where $\left[I(\boldsymbol{\beta}^*)\right]_{\alpha|\gamma}$ is defined in (4.2).

*Proof.* Our proof consists of two steps. Note that by the definition in (2.5) we have

$$\sqrt{n} \cdot S_n(\widehat{\boldsymbol{\beta}}_0, \lambda) = \sqrt{n} \cdot \left[\nabla_1 Q_n(\widehat{\boldsymbol{\beta}}_0; \widehat{\boldsymbol{\beta}}_0)\right]_\alpha - \sqrt{n} \cdot w(\widehat{\boldsymbol{\beta}}_0, \lambda)^\top \cdot \left[\nabla_1 Q_n(\widehat{\boldsymbol{\beta}}_0; \widehat{\boldsymbol{\beta}}_0)\right]_\gamma. \quad \text{(G.19)}$$

Recall that $\mathbf{w}^* = \left[I(\boldsymbol{\beta}^*)\right]_{\gamma,\gamma}^{-1} \cdot \left[I(\boldsymbol{\beta}^*)\right]_{\gamma,\alpha}$ is defined in (4.1). At the first step, we prove

$$\sqrt{n} \cdot S_n(\widehat{\boldsymbol{\beta}}_0, \lambda) = \sqrt{n} \cdot \left[\nabla_1 Q_n(\boldsymbol{\beta}^*; \boldsymbol{\beta}^*)\right]_\alpha - \sqrt{n} \cdot (\mathbf{w}^*)^\top \cdot \left[\nabla_1 Q_n(\boldsymbol{\beta}^*; \boldsymbol{\beta}^*)\right]_\gamma + o_\mathbb{P}(1). \quad \text{(G.20)}$$

In other words, replacing $\widehat{\boldsymbol{\beta}}_0$ and $w(\widehat{\boldsymbol{\beta}}_0, \lambda)$ in (G.19) with the corresponding population quantities $\boldsymbol{\beta}^*$ and $\mathbf{w}^*$ only introduces an $o_{\mathbb{P}}(1)$ error term. Meanwhile, by Theorem D.1 we have $\nabla_1 Q_n(\boldsymbol{\beta}^*; \boldsymbol{\beta}^*) = \nabla \ell_n(\boldsymbol{\beta}^*)/n$. Recall that $\ell_n(\cdot)$ is the log-likelihood defined in (B.3), which implies that in (G.20)

$$\sqrt{n} \cdot \left[\nabla_1 Q_n(\boldsymbol{\beta}^*; \boldsymbol{\beta}^*)\right]_\alpha - \sqrt{n} \cdot (\mathbf{w}^*)^\top \cdot \left[\nabla_1 Q_n(\boldsymbol{\beta}^*; \boldsymbol{\beta}^*)\right]_{\boldsymbol{\gamma}} = \sqrt{n} \cdot \left[1, -(\mathbf{w}^*)^\top\right] \cdot \nabla \ell_n(\boldsymbol{\beta}^*)/n$$

is a (rescaled) average of $n$ i.i.d. random variables. At the second step, we calculate the mean and variance of each term within this average and invoke the central limit theorem. Finally we combine these two steps by invoking Slutsky's theorem. See §I.3 for a detailed proof. □

The next lemma establishes the consistency of $-\left[T_n(\widehat{\boldsymbol{\beta}}_0)\right]_{\alpha|\boldsymbol{\gamma}}$ for estimating $\left[I(\boldsymbol{\beta}^*)\right]_{\alpha|\boldsymbol{\gamma}}$. Recall that $\left[T_n(\widehat{\boldsymbol{\beta}}_0)\right]_{\alpha|\boldsymbol{\gamma}} \in \mathbb{R}$ and $\left[I(\boldsymbol{\beta}^*)\right]_{\alpha|\boldsymbol{\gamma}} \in \mathbb{R}$ are defined in (2.7) and (4.2) respectively.

**Lemma G.4.** Under the assumptions of Theorem 4.6, we have

$$\left[T_n(\widehat{\boldsymbol{\beta}}_0)\right]_{\alpha|\boldsymbol{\gamma}} + \left[I(\boldsymbol{\beta}^*)\right]_{\alpha|\boldsymbol{\gamma}} = o_{\mathbb{P}}(1). \tag{G.21}$$

*Proof.* For notational simplicity, we abbreviate $w(\widehat{\boldsymbol{\beta}}_0, \lambda)$ in the definition of $\left[T_n(\widehat{\boldsymbol{\beta}}_0)\right]_{\alpha|\boldsymbol{\gamma}}$ as $\widehat{\mathbf{w}}_0$. By (2.7) and (4.3), we have

$$\left[T_n(\widehat{\boldsymbol{\beta}}_0)\right]_{\alpha|\boldsymbol{\gamma}} = (1, -\widehat{\mathbf{w}}_0^\top) \cdot T_n(\widehat{\boldsymbol{\beta}}_0) \cdot (1, -\widehat{\mathbf{w}}_0^\top)^\top, \quad \left[I(\boldsymbol{\beta}^*)\right]_{\alpha|\boldsymbol{\gamma}} = \left[1, -(\mathbf{w}^*)^\top\right] \cdot I(\boldsymbol{\beta}^*) \cdot \left[1, -(\mathbf{w}^*)^\top\right]^\top.$$

First, we establish the relationship between $\widehat{\mathbf{w}}_0$ and $\mathbf{w}^*$ by analyzing the Dantzig selector in (2.6). Meanwhile, by Theorem D.1 we have $\mathbb{E}_{\boldsymbol{\beta}^*}\left[T_n(\boldsymbol{\beta}^*)\right] = -I(\boldsymbol{\beta}^*)$. Then by triangle inequality we have

$$\left|T_n(\widehat{\boldsymbol{\beta}}_0) + I(\boldsymbol{\beta}^*)\right| \le \underbrace{\left|T_n(\widehat{\boldsymbol{\beta}}_0) - T_n(\boldsymbol{\beta}^*)\right|}_{(i)} + \underbrace{\left|T_n(\boldsymbol{\beta}^*) - \mathbb{E}_{\boldsymbol{\beta}^*}\left[T_n(\boldsymbol{\beta}^*)\right]\right|}_{(ii)}.$$

We prove term (i) is $o_{\mathbb{P}}(1)$ by quantifying the Lipschitz continuity of $T_n(\cdot)$ using Condition 4.4. We then prove term (ii) is $o_{\mathbb{P}}(1)$ by concentration analysis. Together with the result on the relationship between $\widehat{\mathbf{w}}_0$ and $\mathbf{w}^*$ we establish (G.21). See §I.4 for a detailed proof. □

Combining Lemmas G.3 and G.4 using Slutsky's theorem, we obtain Theorem 4.6.

# H   Proof of Results for Computation and Estimation

We provide the detailed proof of the main results in §3 for computation and parameter estimation. We first lay out the proof for the general framework, and then the proof for specific models.

## H.1   Proof of Lemma G.1

*Proof.* Recall $\overline{\boldsymbol{\beta}}^{(t+0.5)}$ and $\overline{\boldsymbol{\beta}}^{(t+1)}$ are defined in (G.1). Note that in (G.4) of Lemma G.1 we assume

$$\left\|\overline{\boldsymbol{\beta}}^{(t+0.5)} - \boldsymbol{\beta}^*\right\|_2 \le \kappa \cdot \|\boldsymbol{\beta}^*\|_2, \tag{H.1}$$

which implies

$$(1 - \kappa) \cdot \|\boldsymbol{\beta}^*\|_2 \le \left\|\overline{\boldsymbol{\beta}}^{(t+0.5)}\right\|_2 \le (1 + \kappa) \cdot \|\boldsymbol{\beta}^*\|_2. \tag{H.2}$$

For notational simplicity, we define

$$\overline{\boldsymbol{\theta}} = \overline{\boldsymbol{\beta}}^{(t+0.5)}/\left\|\overline{\boldsymbol{\beta}}^{(t+0.5)}\right\|_2, \quad \boldsymbol{\theta} = \boldsymbol{\beta}^{(t+0.5)}/\left\|\overline{\boldsymbol{\beta}}^{(t+0.5)}\right\|_2, \quad \text{and} \quad \boldsymbol{\theta}^* = \boldsymbol{\beta}^*/\|\boldsymbol{\beta}^*\|_2. \tag{H.3}$$

Note that $\overline{\boldsymbol{\theta}}$ and $\boldsymbol{\theta}^*$ are unit vectors, while $\boldsymbol{\theta}$ is not, since it is obtained by normalizing $\boldsymbol{\beta}^{(t+0.5)}$ with $\left\|\overline{\boldsymbol{\beta}}^{(t+0.5)}\right\|_2$. Recall that the $\text{supp}(\cdot, \cdot)$ function is defined in (2.2). Hence we have

$$\text{supp}(\boldsymbol{\theta}^*) = \text{supp}(\boldsymbol{\beta}^*) = \mathcal{S}^*, \quad \text{and} \quad \text{supp}(\boldsymbol{\theta}, \widehat{s}) = \text{supp}(\boldsymbol{\beta}^{(t+0.5)}, \widehat{s}) = \widehat{\mathcal{S}}^{(t+0.5)}, \tag{H.4}$$

where the last equality follows from line 6 of Algorithm 4. To ease the notation, we define

$$\mathcal{I}_1 = \mathcal{S}^* \setminus \widehat{\mathcal{S}}^{(t+0.5)}, \quad \mathcal{I}_2 = \mathcal{S}^* \cap \widehat{\mathcal{S}}^{(t+0.5)}, \quad \text{and} \quad \mathcal{I}_3 = \widehat{\mathcal{S}}^{(t+0.5)} \setminus \mathcal{S}^*. \tag{H.5}$$

Let $s_1 = |\mathcal{I}_1|$, $s_2 = |\mathcal{I}_2|$ and $s_3 = |\mathcal{I}_3|$ correspondingly. Also, we define $\Delta = \langle \overline{\boldsymbol{\theta}}, \boldsymbol{\theta}^* \rangle$. Note that

$$\Delta = \langle \overline{\boldsymbol{\theta}}, \boldsymbol{\theta}^* \rangle = \sum_{j \in \mathcal{S}^*} \bar{\theta}_j \cdot \theta_j^* = \sum_{j \in \mathcal{I}_1} \bar{\theta}_j \cdot \theta_j^* + \sum_{j \in \mathcal{I}_2} \bar{\theta}_j \cdot \theta_j^* \le \left\|\overline{\boldsymbol{\theta}}_{\mathcal{I}_1}\right\|_2 \cdot \left\|\boldsymbol{\theta}_{\mathcal{I}_1}^*\right\|_2 + \left\|\overline{\boldsymbol{\theta}}_{\mathcal{I}_2}\right\|_2 \cdot \left\|\boldsymbol{\theta}_{\mathcal{I}_2}^*\right\|_2.$$

$$\tag{H.6}$$

Here the first equality is from $\text{supp}(\boldsymbol{\theta}^*) = \mathcal{S}^*$, the second equality is from (H.5) and the last inequality is from Cauchy-Schwarz inequality. Furthermore, from (H.6) we have

$$
\begin{aligned}
\Delta^2 &\leq \left( \left\| \bar{\boldsymbol{\theta}}_{\mathcal{I}_1} \right\|_2 \cdot \left\| \boldsymbol{\theta}^*_{\mathcal{I}_1} \right\|_2 + \left\| \bar{\boldsymbol{\theta}}_{\mathcal{I}_2} \right\|_2 \cdot \left\| \boldsymbol{\theta}^*_{\mathcal{I}_2} \right\|_2 \right)^2 \\
&\leq \left\| \bar{\boldsymbol{\theta}}_{\mathcal{I}_1} \right\|_2^2 \cdot \left( \left\| \boldsymbol{\theta}^*_{\mathcal{I}_1} \right\|_2^2 + \left\| \boldsymbol{\theta}^*_{\mathcal{I}_2} \right\|_2^2 \right) + \left\| \bar{\boldsymbol{\theta}}_{\mathcal{I}_2} \right\|_2^2 \cdot \left( \left\| \boldsymbol{\theta}^*_{\mathcal{I}_1} \right\|_2^2 + \left\| \boldsymbol{\theta}^*_{\mathcal{I}_2} \right\|_2^2 \right) \\
&= \left\| \bar{\boldsymbol{\theta}}_{\mathcal{I}_1} \right\|_2^2 + \left\| \bar{\boldsymbol{\theta}}_{\mathcal{I}_2} \right\|_2^2 \\
&\leq 1 - \left\| \bar{\boldsymbol{\theta}}_{\mathcal{I}_3} \right\|_2^2.
\end{aligned}
\tag{H.7}
$$

To obtain the second inequality, we expand the square and apply $2ab \leq a^2 + b^2$. In the equality and the last inequality of (H.7), we use the fact that $\boldsymbol{\theta}^*$ and $\bar{\boldsymbol{\theta}}$ are both unit vectors.

By (2.2) and (H.4), $\widehat{\mathcal{S}}^{(t+0.5)}$ contains the index $j$'s with the top $\widehat{s}$ largest $\left| \beta_j^{(t+0.5)} \right|$'s. Therefore, we have

$$
\frac{\left\| \boldsymbol{\beta}_{\mathcal{I}_3}^{(t+0.5)} \right\|_2^2}{s_3} = \frac{\sum_{j \in \mathcal{I}_3} \left( \beta_j^{(t+0.5)} \right)^2}{s_3} \geq \frac{\sum_{j \in \mathcal{I}_1} \left( \beta_j^{(t+0.5)} \right)^2}{s_1} = \frac{\left\| \boldsymbol{\beta}_{\mathcal{I}_1}^{(t+0.5)} \right\|_2^2}{s_1},
\tag{H.8}
$$

because from (H.5) we have $\mathcal{I}_3 \subseteq \widehat{\mathcal{S}}^{(t+0.5)}$ and $\mathcal{I}_1 \cap \widehat{\mathcal{S}}^{(t+0.5)} = \emptyset$. Taking square roots of both sides of (H.8) and then dividing them by $\left\| \bar{\boldsymbol{\beta}}^{(t+0.5)} \right\|_2$ (which is nonzero according to (H.2)), by the definition of $\boldsymbol{\theta}$ in (H.3) we obtain

$$
\frac{\left\| \boldsymbol{\theta}_{\mathcal{I}_3} \right\|_2}{\sqrt{s_3}} \geq \frac{\left\| \boldsymbol{\theta}_{\mathcal{I}_1} \right\|_2}{\sqrt{s_1}}.
\tag{H.9}
$$

Equipped with (H.9), we now quantify the relationship between $\left\| \bar{\boldsymbol{\theta}}_{\mathcal{I}_3} \right\|_2$ and $\left\| \bar{\boldsymbol{\theta}}_{\mathcal{I}_1} \right\|_2$. For notational simplicity, let

$$
\widetilde{\epsilon} = 2 \cdot \left\| \bar{\boldsymbol{\theta}} - \boldsymbol{\theta} \right\|_\infty = 2 \cdot \left\| \bar{\boldsymbol{\beta}}^{(t+0.5)} - \boldsymbol{\beta}^{(t+0.5)} \right\|_\infty / \left\| \bar{\boldsymbol{\beta}}^{(t+0.5)} \right\|_2.
\tag{H.10}
$$

Note that we have

$$
\max \left\{ \frac{\left\| \boldsymbol{\theta}_{\mathcal{I}_3} - \bar{\boldsymbol{\theta}}_{\mathcal{I}_3} \right\|_2}{\sqrt{s_3}}, \frac{\left\| \boldsymbol{\theta}_{\mathcal{I}_1} - \bar{\boldsymbol{\theta}}_{\mathcal{I}_1} \right\|_2}{\sqrt{s_1}} \right\} \leq \max \left\{ \left\| \boldsymbol{\theta}_{\mathcal{I}_3} - \bar{\boldsymbol{\theta}}_{\mathcal{I}_3} \right\|_\infty, \left\| \boldsymbol{\theta}_{\mathcal{I}_1} - \bar{\boldsymbol{\theta}}_{\mathcal{I}_1} \right\|_\infty \right\} \leq \left\| \bar{\boldsymbol{\theta}} - \boldsymbol{\theta} \right\|_\infty = \widetilde{\epsilon}/2,
$$

which implies

$$
\begin{aligned}
\frac{\left\| \bar{\boldsymbol{\theta}}_{\mathcal{I}_3} \right\|_2}{\sqrt{s_3}} &\geq \frac{\left\| \boldsymbol{\theta}_{\mathcal{I}_3} \right\|_2}{\sqrt{s_3}} - \frac{\left\| \boldsymbol{\theta}_{\mathcal{I}_3} - \bar{\boldsymbol{\theta}}_{\mathcal{I}_3} \right\|_2}{\sqrt{s_3}} \geq \frac{\left\| \boldsymbol{\theta}_{\mathcal{I}_1} \right\|_2}{\sqrt{s_1}} - \frac{\left\| \boldsymbol{\theta}_{\mathcal{I}_3} - \bar{\boldsymbol{\theta}}_{\mathcal{I}_3} \right\|_2}{\sqrt{s_3}} \\
&\geq \frac{\left\| \bar{\boldsymbol{\theta}}_{\mathcal{I}_1} \right\|_2}{\sqrt{s_1}} - \frac{\left\| \bar{\boldsymbol{\theta}}_{\mathcal{I}_1} - \boldsymbol{\theta}_{\mathcal{I}_1} \right\|_2}{\sqrt{s_1}} - \frac{\left\| \boldsymbol{\theta}_{\mathcal{I}_3} - \bar{\boldsymbol{\theta}}_{\mathcal{I}_3} \right\|_2}{\sqrt{s_3}} \geq \frac{\left\| \bar{\boldsymbol{\theta}}_{\mathcal{I}_1} \right\|_2}{\sqrt{s_1}} - \widetilde{\epsilon},
\end{aligned}
\tag{H.11}
$$

where second inequality is obtained from (H.9), while the first and third are from triangle inequality. Plugging (H.11) into (H.7), we obtain

$$
\Delta^2 \leq 1 - \left\| \bar{\boldsymbol{\theta}}_{\mathcal{I}_3} \right\|_2^2 \leq 1 - \left( \sqrt{s_3/s_1} \cdot \left\| \bar{\boldsymbol{\theta}}_{\mathcal{I}_1} \right\|_2 - \sqrt{s_3} \cdot \widetilde{\epsilon} \right)^2.
$$

Since by definition we have $\Delta = \left\langle \bar{\boldsymbol{\theta}}, \boldsymbol{\theta}^* \right\rangle \in [-1, 1]$, solving for $\left\| \bar{\boldsymbol{\theta}}_{\mathcal{I}_1} \right\|_2$ in the above inequality yields

$$
\left\| \bar{\boldsymbol{\theta}}_{\mathcal{I}_1} \right\|_2 \leq \sqrt{s_1/s_3} \cdot \sqrt{1 - \Delta^2} + \sqrt{s_1} \cdot \widetilde{\epsilon} \leq \sqrt{s^*/\widehat{s}} \cdot \sqrt{1 - \Delta^2} + \sqrt{s^*} \cdot \widetilde{\epsilon}.
\tag{H.12}
$$

Here we employ the fact that $s_1 \leq s^*$ and $s_1/s_3 \leq (s_1 + s_2)/(s_3 + s_2) = s^*/\widehat{s}$, which follows from (H.5) and our assumption in (G.5) that $s^*/\widehat{s} \leq (1 - \kappa)^2 / \left[ 4 \cdot (1 + \kappa)^2 \right] < 1$.

In the following, we prove that the right-hand side of (H.12) is upper bounded by $\Delta$, i.e.,

$$
\sqrt{s^*/\widehat{s}} \cdot \sqrt{1 - \Delta^2} + \sqrt{s^*} \cdot \widetilde{\epsilon} \leq \Delta.
\tag{H.13}
$$

We can verify that a sufficient condition for (H.13) to hold is that

$$
\begin{aligned}
\Delta &\geq \frac{\sqrt{s^*} \cdot \widetilde{\epsilon} + \left[ s^* \cdot \widetilde{\epsilon}^2 - \left( s^*/\widehat{s} + 1 \right) \cdot \left( s^* \cdot \widetilde{\epsilon}^2 - s^*/\widehat{s} \right) \right]^{1/2}}{s^*/\widehat{s} + 1} \\
&= \frac{\sqrt{s^*} \cdot \widetilde{\epsilon} + \left[ -\left( s^* \cdot \widetilde{\epsilon} \right)^2 / \widehat{s} + \left( s^*/\widehat{s} + 1 \right) \cdot \left( s^*/\widehat{s} \right) \right]^{1/2}}{s^*/\widehat{s} + 1},
\end{aligned}
\tag{H.14}
$$

which is obtained by solving for $\Delta$ in (H.13). When we are solving for $\Delta$ in (H.13), we use the fact that $\sqrt{s^*} \cdot \widetilde{\epsilon} \leq \Delta$, which holds because

$$\sqrt{s^*} \cdot \widetilde{\epsilon} \leq \sqrt{\widehat{s}} \cdot \widetilde{\epsilon} = 2 \cdot \frac{\sqrt{\widehat{s}} \cdot \left\| \overline{\boldsymbol{\beta}}^{(t+0.5)} - \boldsymbol{\beta}^{(t+0.5)} \right\|_\infty}{\left\| \overline{\boldsymbol{\beta}}^{(t+0.5)} \right\|_2} \leq \frac{1-\kappa}{1+\kappa} \leq \Delta. \tag{H.15}$$

The first inequality is from our assumption in (G.5) that $s^*/\widehat{s} \leq (1-\kappa)^2/\left[4 \cdot (1+\kappa)^2\right] < 1$. The equality is from the definition of $\widetilde{\epsilon}$ in (H.10). The second inequality follows from our assumption in (G.5) that

$$\sqrt{\widehat{s}} \cdot \left\| \boldsymbol{\beta}^{(t+0.5)} - \overline{\boldsymbol{\beta}}^{(t+0.5)} \right\|_\infty \leq \frac{(1-\kappa)^2}{2 \cdot (1+\kappa)} \cdot \|\boldsymbol{\beta}^*\|_2$$

and the first inequality in (H.2). To prove the last inequality in (H.15), we note that (H.1) implies

$$\left\| \overline{\boldsymbol{\beta}}^{(t+0.5)} \right\|_2^2 + \|\boldsymbol{\beta}^*\|_2^2 - 2 \cdot \left\langle \overline{\boldsymbol{\beta}}^{(t+0.5)}, \boldsymbol{\beta}^* \right\rangle = \left\| \overline{\boldsymbol{\beta}}^{(t+0.5)} - \boldsymbol{\beta}^* \right\|_2^2 \leq \kappa^2 \cdot \|\boldsymbol{\beta}^*\|_2^2.$$

This together with (H.3) implies

$$\Delta = \langle \overline{\boldsymbol{\theta}}, \boldsymbol{\theta}^* \rangle = \frac{\left\langle \overline{\boldsymbol{\beta}}^{(t+0.5)}, \boldsymbol{\beta}^* \right\rangle}{\left\| \overline{\boldsymbol{\beta}}^{(t+0.5)} \right\|_2 \cdot \|\boldsymbol{\beta}^*\|_2} \geq \frac{\left\| \overline{\boldsymbol{\beta}}^{(t+0.5)} \right\|_2^2 + \|\boldsymbol{\beta}^*\|_2^2 - \kappa^2 \cdot \|\boldsymbol{\beta}^*\|_2^2}{2 \cdot \left\| \overline{\boldsymbol{\beta}}^{(t+0.5)} \right\|_2 \cdot \|\boldsymbol{\beta}^*\|_2}$$

$$\geq \frac{(1-\kappa)^2 + 1 - \kappa^2}{2 \cdot (1+\kappa)} = \frac{1-\kappa}{1+\kappa}, \tag{H.16}$$

where in the second inequality we use both sides of (H.2). In summary, we have that (H.15) holds. Now we verify that (H.14) holds. By (H.15) we have

$$\sqrt{\widehat{s}} \cdot \widetilde{\epsilon} \leq \frac{1-\kappa}{1+\kappa} < 1 < \sqrt{(s^* + \widehat{s})/\widehat{s}},$$

which implies $\widetilde{\epsilon} \leq \sqrt{s^* + \widehat{s}}/\widehat{s}$. For the right-hand side of (H.14) we have

$$\frac{\sqrt{s^*} \cdot \widetilde{\epsilon} + \left[ -\left(s^* \cdot \widetilde{\epsilon}\right)^2/\widehat{s} + \left(s^*/\widehat{s} + 1\right) \cdot \left(s^*/\widehat{s}\right) \right]^{1/2}}{s^*/\widehat{s} + 1} \leq \frac{\sqrt{s^*} \cdot \widetilde{\epsilon} + \left[ \left(s^*/\widehat{s} + 1\right) \cdot \left(s^*/\widehat{s}\right) \right]^{1/2}}{s^*/\widehat{s} + 1}$$

$$\leq 2 \cdot \sqrt{s^*/(s^* + \widehat{s})}, \tag{H.17}$$

where the last inequality is obtained by plugging in $\widetilde{\epsilon} \leq \sqrt{s^* + \widehat{s}}/\widehat{s}$. Meanwhile, note that we have

$$2 \cdot \sqrt{s^*/(s^* + \widehat{s})} \leq 2 \cdot \sqrt{1/\left[1 + 4 \cdot (1+\kappa)^2/(1-\kappa)^2\right]} \leq (1-\kappa)/(1+\kappa) \leq \Delta, \tag{H.18}$$

where the first inequality is from our assumption in (G.5) that $s^*/\widehat{s} \leq (1-\kappa)^2/\left[4 \cdot (1+\kappa)^2\right]$, while the last inequality is from (H.16). Combining (H.17) and (H.18), we then obtain (H.14). By (H.14) we further establish (H.13), i.e., the right-hand side of (H.12) is upper bounded by $\Delta$, which implies

$$\left\| \overline{\boldsymbol{\theta}}_{\mathcal{I}_1} \right\|_2 \leq \Delta. \tag{H.19}$$

Furthermore, according to (H.6) we have

$$\Delta \leq \left\| \overline{\boldsymbol{\theta}}_{\mathcal{I}_1} \right\|_2 \cdot \left\| \boldsymbol{\theta}^*_{\mathcal{I}_1} \right\|_2 + \left\| \overline{\boldsymbol{\theta}}_{\mathcal{I}_2} \right\|_2 \cdot \left\| \boldsymbol{\theta}^*_{\mathcal{I}_2} \right\|_2 \leq \left\| \overline{\boldsymbol{\theta}}_{\mathcal{I}_1} \right\|_2 \cdot \left\| \boldsymbol{\theta}^*_{\mathcal{I}_1} \right\|_2 + \sqrt{1 - \left\| \overline{\boldsymbol{\theta}}_{\mathcal{I}_1} \right\|_2^2} \cdot \sqrt{1 - \left\| \boldsymbol{\theta}^*_{\mathcal{I}_1} \right\|_2^2}, \tag{H.20}$$

where in the last inequality we use the fact $\boldsymbol{\theta}^*$ and $\overline{\boldsymbol{\theta}}$ are unit vectors. Now we solve for $\left\| \boldsymbol{\theta}^*_{\mathcal{I}_1} \right\|_2$ in (H.20). According to (H.19) and the fact that $\left\| \boldsymbol{\theta}^*_{\mathcal{I}_1} \right\|_2 \leq \|\boldsymbol{\theta}^*\|_2 = 1$, on the right-hand side of (H.20) we have $\left\| \overline{\boldsymbol{\theta}}_{\mathcal{I}_1} \right\|_2 \cdot \left\| \boldsymbol{\theta}^*_{\mathcal{I}_1} \right\|_2 \leq \left\| \overline{\boldsymbol{\theta}}_{\mathcal{I}_1} \right\|_2 \leq \Delta$. Thus, we have

$$\left( \Delta - \left\| \overline{\boldsymbol{\theta}}_{\mathcal{I}_1} \right\|_2 \cdot \left\| \boldsymbol{\theta}^*_{\mathcal{I}_1} \right\|_2 \right)^2 \leq \left( 1 - \left\| \overline{\boldsymbol{\theta}}_{\mathcal{I}_1} \right\|_2^2 \right) \cdot \left( 1 - \left\| \boldsymbol{\theta}^*_{\mathcal{I}_1} \right\|_2^2 \right).$$

Further by solving for $\left\| \boldsymbol{\theta}^*_{\mathcal{I}_1} \right\|_2$ in the above inequality, we obtain

$$\left\| \boldsymbol{\theta}^*_{\mathcal{I}_1} \right\|_2 \leq \left\| \overline{\boldsymbol{\theta}}_{\mathcal{I}_1} \right\|_2 \cdot \Delta + \sqrt{1 - \left\| \overline{\boldsymbol{\theta}}_{\mathcal{I}_1} \right\|_2^2} \cdot \sqrt{1 - \Delta^2} \leq \left\| \overline{\boldsymbol{\theta}}_{\mathcal{I}_1} \right\|_2 + \sqrt{1 - \Delta^2}$$

$$\leq \left( 1 + \sqrt{s^*/\widehat{s}} \right) \cdot \sqrt{1 - \Delta^2} + \sqrt{s^*} \cdot \widetilde{\epsilon}, \tag{H.21}$$

where in the second inequality we use the fact that $\Delta \leq 1$, which follows from its definition, while in the last inequality we plug in (H.12). Then combining (H.12) and (H.21), we obtain

$$\left\|\bar{\boldsymbol{\theta}}_{\mathcal{I}_1}\right\|_2 \cdot \left\|\boldsymbol{\theta}^*_{\mathcal{I}_1}\right\|_2 \leq \left[\sqrt{s^*/\widehat{s}} \cdot \sqrt{1 - \Delta^2} + \sqrt{s^*} \cdot \widetilde{\epsilon}\right] \cdot \left[\left(1 + \sqrt{s^*/\widehat{s}}\right) \cdot \sqrt{1 - \Delta^2} + \sqrt{s^*} \cdot \widetilde{\epsilon}\right]. \tag{H.22}$$

Note that by (G.1) and the definition of $\bar{\boldsymbol{\theta}}$ in (H.3), we have

$$\bar{\boldsymbol{\beta}}^{(t+1)} = \mathrm{trunc}\big(\bar{\boldsymbol{\beta}}^{(t+0.5)}, \widehat{\mathcal{S}}^{(t+0.5)}\big) = \mathrm{trunc}\big(\bar{\boldsymbol{\theta}}, \widehat{\mathcal{S}}^{(t+0.5)}\big) \cdot \left\|\bar{\boldsymbol{\beta}}^{(t+0.5)}\right\|_2.$$

Therefore, we have

$$\left\langle \bar{\boldsymbol{\beta}}^{(t+1)}/\left\|\bar{\boldsymbol{\beta}}^{(t+0.5)}\right\|_2, \boldsymbol{\beta}^*/\|\boldsymbol{\beta}^*\|_2 \right\rangle = \left\langle \mathrm{trunc}\big(\bar{\boldsymbol{\theta}}, \widehat{\mathcal{S}}^{(t+0.5)}\big), \boldsymbol{\theta}^* \right\rangle = \left\langle \bar{\boldsymbol{\theta}}_{\mathcal{I}_2}, \boldsymbol{\theta}^*_{\mathcal{I}_2} \right\rangle = \left\langle \bar{\boldsymbol{\theta}}, \boldsymbol{\theta}^* \right\rangle - \left\langle \bar{\boldsymbol{\theta}}_{\mathcal{I}_1}, \boldsymbol{\theta}^*_{\mathcal{I}_1} \right\rangle$$

$$\geq \left\langle \bar{\boldsymbol{\theta}}, \boldsymbol{\theta}^* \right\rangle - \left\|\bar{\boldsymbol{\theta}}_{\mathcal{I}_1}\right\|_2 \cdot \left\|\boldsymbol{\theta}^*_{\mathcal{I}_1}\right\|_2,$$

where the second and third equalities follow from (H.5). Let $\bar{\chi} = \left\|\bar{\boldsymbol{\beta}}^{(t+0.5)}\right\|_2 \cdot \|\boldsymbol{\beta}^*\|_2$. Plugging (H.22) into the right-hand side of the above inequality and then multiplying $\bar{\chi}$ on both sides, we obtain

$$\left\langle \bar{\boldsymbol{\beta}}^{(t+1)}, \boldsymbol{\beta}^* \right\rangle \tag{H.23}$$

$$\geq \left\langle \bar{\boldsymbol{\beta}}^{(t+0.5)}, \boldsymbol{\beta}^* \right\rangle$$

$$- \left[\sqrt{s^*/\widehat{s}} \cdot \sqrt{\bar{\chi} \cdot (1 - \Delta^2)} + \sqrt{s^*} \cdot \sqrt{\bar{\chi}} \cdot \widetilde{\epsilon}\right] \cdot \left[\left(1 + \sqrt{s^*/\widehat{s}}\right) \cdot \sqrt{\bar{\chi} \cdot (1 - \Delta^2)} + \sqrt{s^*} \cdot \sqrt{\bar{\chi}} \cdot \widetilde{\epsilon}\right]$$

$$= \left\langle \bar{\boldsymbol{\beta}}^{(t+0.5)}, \boldsymbol{\beta}^* \right\rangle - \left(\sqrt{s^*/\widehat{s}} + s^*/\widehat{s}\right) \cdot \bar{\chi} \cdot (1 - \Delta^2)$$

$$- \left(1 + 2 \cdot \sqrt{s^*/\widehat{s}}\right) \cdot \underbrace{\sqrt{\bar{\chi} \cdot (1 - \Delta^2)}}_{(i)} \cdot \sqrt{s^*} \cdot \underbrace{\sqrt{\bar{\chi}} \cdot \widetilde{\epsilon}}_{(ii)} - \left(\sqrt{s^*} \cdot \sqrt{\bar{\chi}} \cdot \widetilde{\epsilon}\right)^2.$$

For term (i) in (H.23), note that $\sqrt{1 - \Delta^2} \leq \sqrt{2 \cdot (1 - \Delta)}$. By (H.3) and the definition that $\Delta = \left\langle \bar{\boldsymbol{\theta}}, \boldsymbol{\theta}^* \right\rangle$, for term (i) we have

$$\sqrt{\bar{\chi} \cdot (1 - \Delta^2)} \leq \sqrt{2 \cdot \bar{\chi} \cdot (1 - \Delta)} \leq \sqrt{2 \cdot \left\|\bar{\boldsymbol{\beta}}^{(t+0.5)}\right\|_2 \cdot \|\boldsymbol{\beta}^*\|_2 - 2 \cdot \left\langle \bar{\boldsymbol{\beta}}^{(t+0.5)}, \boldsymbol{\beta}^* \right\rangle} \tag{H.24}$$

$$\leq \sqrt{\left\|\bar{\boldsymbol{\beta}}^{(t+0.5)}\right\|_2^2 + \|\boldsymbol{\beta}^*\|_2^2 - 2 \cdot \left\langle \bar{\boldsymbol{\beta}}^{(t+0.5)}, \boldsymbol{\beta}^* \right\rangle} = \left\|\bar{\boldsymbol{\beta}}^{(t+0.5)} - \boldsymbol{\beta}^*\right\|_2.$$

For term (ii) in (H.23), by the definition of $\widetilde{\epsilon}$ in (H.10) we have

$$\sqrt{\bar{\chi}} \cdot \widetilde{\epsilon} = \sqrt{\left\|\bar{\boldsymbol{\beta}}^{(t+0.5)}\right\|_2 \cdot \|\boldsymbol{\beta}^*\|_2} \cdot 2 \cdot \left\|\bar{\boldsymbol{\beta}}^{(t+0.5)} - \boldsymbol{\beta}^{(t+0.5)}\right\|_\infty / \left\|\bar{\boldsymbol{\beta}}^{(t+0.5)}\right\|_2$$

$$= 2 \cdot \left\|\bar{\boldsymbol{\beta}}^{(t+0.5)} - \boldsymbol{\beta}^{(t+0.5)}\right\|_\infty \cdot \sqrt{\|\boldsymbol{\beta}^*\|_2 / \left\|\bar{\boldsymbol{\beta}}^{(t+0.5)}\right\|_2} \leq \frac{2}{\sqrt{1 - \kappa}} \cdot \left\|\bar{\boldsymbol{\beta}}^{(t+0.5)} - \boldsymbol{\beta}^{(t+0.5)}\right\|_\infty, \tag{H.25}$$

where the last inequality is obtained from (H.2). Plugging (H.24) and (H.25) into (H.23), we obtain

$$\left\langle \bar{\boldsymbol{\beta}}^{(t+1)}, \boldsymbol{\beta}^* \right\rangle \geq \left\langle \bar{\boldsymbol{\beta}}^{(t+0.5)}, \boldsymbol{\beta}^* \right\rangle - \left(\sqrt{s^*/\widehat{s}} + s^*/\widehat{s}\right) \cdot \left\|\bar{\boldsymbol{\beta}}^{(t+0.5)} - \boldsymbol{\beta}^*\right\|_2^2 \tag{H.26}$$

$$- \left(1 + 2 \cdot \sqrt{s^*/\widehat{s}}\right) \cdot \left\|\bar{\boldsymbol{\beta}}^{(t+0.5)} - \boldsymbol{\beta}^*\right\|_2 \cdot \frac{2 \cdot \sqrt{s^*}}{\sqrt{1 - \kappa}} \cdot \left\|\bar{\boldsymbol{\beta}}^{(t+0.5)} - \boldsymbol{\beta}^{(t+0.5)}\right\|_\infty$$

$$- \frac{4 \cdot s^*}{1 - \kappa} \cdot \left\|\bar{\boldsymbol{\beta}}^{(t+0.5)} - \boldsymbol{\beta}^{(t+0.5)}\right\|_\infty^2.$$

Meanwhile, according to (G.1) we have that $\bar{\boldsymbol{\beta}}^{(t+1)}$ is obtained by truncating $\bar{\boldsymbol{\beta}}^{(t+0.5)}$, which implies

$$\left\|\bar{\boldsymbol{\beta}}^{(t+1)}\right\|_2^2 + \|\boldsymbol{\beta}^*\|_2^2 \leq \left\|\bar{\boldsymbol{\beta}}^{(t+0.5)}\right\|_2^2 + \|\boldsymbol{\beta}^*\|_2^2. \tag{H.27}$$

Subtracting two times both sides of (H.26) from (H.27), we obtain

$$\left\|\bar{\boldsymbol{\beta}}^{(t+1)} - \boldsymbol{\beta}^*\right\|_2^2 \leq \left(1 + 2 \cdot \sqrt{s^*/\widehat{s}} + 2 \cdot s^*/\widehat{s}\right) \cdot \left\|\bar{\boldsymbol{\beta}}^{(t+0.5)} - \boldsymbol{\beta}^*\right\|_2^2$$

$$+ \left(1 + 2 \cdot \sqrt{s^*/\widehat{s}}\right) \cdot \frac{4 \cdot \sqrt{s^*}}{\sqrt{1 - \kappa}} \cdot \left\|\bar{\boldsymbol{\beta}}^{(t+0.5)} - \boldsymbol{\beta}^{(t+0.5)}\right\|_\infty \cdot \left\|\bar{\boldsymbol{\beta}}^{(t+0.5)} - \boldsymbol{\beta}^*\right\|_2$$

$$+ \frac{8 \cdot s^*}{1 - \kappa} \cdot \left\|\bar{\boldsymbol{\beta}}^{(t+0.5)} - \boldsymbol{\beta}^{(t+0.5)}\right\|_\infty^2.$$

We can easily verify that the above inequality implies

$$\left\|\overline{\boldsymbol{\beta}}^{(t+1)} - \boldsymbol{\beta}^*\right\|_2^2 \leq \left(1 + 2 \cdot \sqrt{s^*/\widehat{s}} + 2 \cdot s^*/\widehat{s}\right) \cdot \left[\left\|\overline{\boldsymbol{\beta}}^{(t+0.5)} - \boldsymbol{\beta}^*\right\|_2 + \frac{2 \cdot \sqrt{s^*}}{\sqrt{1-\kappa}} \cdot \left\|\overline{\boldsymbol{\beta}}^{(t+0.5)} - \boldsymbol{\beta}^{(t+0.5)}\right\|_\infty\right]^2$$

$$+ \frac{8 \cdot s^*}{1-\kappa} \cdot \left\|\overline{\boldsymbol{\beta}}^{(t+0.5)} - \boldsymbol{\beta}^{(t+0.5)}\right\|_\infty^2.$$

Taking square roots of both sides and utilizing the fact that $\sqrt{a^2 + b^2} \leq a + b \ (a, b > 0)$, we obtain

$$\left\|\overline{\boldsymbol{\beta}}^{(t+1)} - \boldsymbol{\beta}^*\right\|_2 \leq \left(1 + 4 \cdot \sqrt{s^*/\widehat{s}}\right)^{1/2} \cdot \left\|\overline{\boldsymbol{\beta}}^{(t+0.5)} - \boldsymbol{\beta}^*\right\|_2 \tag{H.28}$$

$$+ \frac{C \cdot \sqrt{s^*}}{\sqrt{1-\kappa}} \cdot \left\|\overline{\boldsymbol{\beta}}^{(t+0.5)} - \boldsymbol{\beta}^{(t+0.5)}\right\|_\infty,$$

where $C > 0$ is an constant. Here we utilize the fact that $s^*/\widehat{s} \leq \sqrt{s^*/\widehat{s}}$ and

$$1 + 2 \cdot \sqrt{s^*/\widehat{s}} + 2 \cdot s^*/\widehat{s} \leq 5,$$

both of which follow from our assumption that $s^*/\widehat{s} \leq (1-\kappa)^2/\left[4 \cdot (1+\kappa)^2\right] < 1$ in (G.5). By (H.28) we conclude the proof of Lemma G.1. □

## H.2   Proof of Lemma G.2

In the following, we prove (G.8) and (G.9) for the maximization and gradient ascent implementation of the M-step correspondingly.

**Proof of** (G.8): To prove (G.8) for the maximization implementation of the M-step (Algorithm 2), note that by the self-consistency property [18] we have

$$\boldsymbol{\beta}^* = \underset{\boldsymbol{\beta}}{\operatorname{argmax}} \, Q(\boldsymbol{\beta}; \boldsymbol{\beta}^*). \tag{H.29}$$

Hence, $\boldsymbol{\beta}^*$ satisfies the following first-order optimality condition

$$\left\langle \boldsymbol{\beta} - \boldsymbol{\beta}^*, \nabla_1 Q(\boldsymbol{\beta}^*; \boldsymbol{\beta}^*) \right\rangle \leq 0, \quad \text{for all } \boldsymbol{\beta},$$

where $\nabla_1 Q(\cdot, \cdot)$ denotes the gradient taken with respect to the first variable. In particular, it implies

$$\left\langle \overline{\boldsymbol{\beta}}^{(t+0.5)} - \boldsymbol{\beta}^*, \nabla_1 Q(\boldsymbol{\beta}^*; \boldsymbol{\beta}^*) \right\rangle \leq 0. \tag{H.30}$$

Meanwhile, by (G.1) and the definition of $M(\cdot)$ in (3.1), we have

$$\overline{\boldsymbol{\beta}}^{(t+0.5)} = M\left(\boldsymbol{\beta}^{(t)}\right) = \underset{\boldsymbol{\beta}}{\operatorname{argmax}} \, Q(\boldsymbol{\beta}; \boldsymbol{\beta}^{(t)}).$$

Hence we have the following first-order optimality condition

$$\left\langle \boldsymbol{\beta} - \overline{\boldsymbol{\beta}}^{(t+0.5)}, \nabla_1 Q\left(\overline{\boldsymbol{\beta}}^{(t+0.5)}; \boldsymbol{\beta}^{(t)}\right) \right\rangle \leq 0, \quad \text{for all } \boldsymbol{\beta},$$

which implies

$$\left\langle \boldsymbol{\beta}^* - \overline{\boldsymbol{\beta}}^{(t+0.5)}, \nabla_1 Q\left(\overline{\boldsymbol{\beta}}^{(t+0.5)}; \boldsymbol{\beta}^{(t)}\right) \right\rangle \leq 0. \tag{H.31}$$

Combining (H.30) and (H.31), we then obtain

$$\left\langle \boldsymbol{\beta}^* - \overline{\boldsymbol{\beta}}^{(t+0.5)}, -\nabla_1 Q(\boldsymbol{\beta}^*; \boldsymbol{\beta}^*) \right\rangle \leq \left\langle \boldsymbol{\beta}^* - \overline{\boldsymbol{\beta}}^{(t+0.5)}, -\nabla_1 Q\left(\overline{\boldsymbol{\beta}}^{(t+0.5)}; \boldsymbol{\beta}^{(t)}\right) \right\rangle,$$

which implies

$$\left\langle \boldsymbol{\beta}^* - \overline{\boldsymbol{\beta}}^{(t+0.5)}, \nabla_1 Q\left(\overline{\boldsymbol{\beta}}^{(t+0.5)}; \boldsymbol{\beta}^*\right) - \nabla_1 Q(\boldsymbol{\beta}^*; \boldsymbol{\beta}^*) \right\rangle$$
$$\leq \left\langle \boldsymbol{\beta}^* - \overline{\boldsymbol{\beta}}^{(t+0.5)}, \nabla_1 Q\left(\overline{\boldsymbol{\beta}}^{(t+0.5)}; \boldsymbol{\beta}^*\right) - \nabla_1 Q\left(\overline{\boldsymbol{\beta}}^{(t+0.5)}; \boldsymbol{\beta}^{(t)}\right) \right\rangle. \tag{H.32}$$

In the following, we establish upper and lower bounds for both sides of (H.32) correspondingly. By applying Condition *Lipschitz-Gradient*-$1(\gamma_1, \mathcal{B})$, for the right-hand side of (H.32) we have

$$\left\langle \boldsymbol{\beta}^* - \overline{\boldsymbol{\beta}}^{(t+0.5)}, \nabla_1 Q\left(\overline{\boldsymbol{\beta}}^{(t+0.5)}; \boldsymbol{\beta}^*\right) - \nabla_1 Q\left(\overline{\boldsymbol{\beta}}^{(t+0.5)}; \boldsymbol{\beta}^{(t)}\right) \right\rangle$$
$$\leq \left\|\boldsymbol{\beta}^* - \overline{\boldsymbol{\beta}}^{(t+0.5)}\right\|_2 \cdot \left\|\nabla_1 Q\left(\overline{\boldsymbol{\beta}}^{(t+0.5)}; \boldsymbol{\beta}^*\right) - \nabla_1 Q\left(\overline{\boldsymbol{\beta}}^{(t+0.5)}; \boldsymbol{\beta}^{(t)}\right)\right\|_2$$
$$\leq \gamma_1 \cdot \left\|\boldsymbol{\beta}^* - \overline{\boldsymbol{\beta}}^{(t+0.5)}\right\|_2 \cdot \left\|\boldsymbol{\beta}^* - \boldsymbol{\beta}^{(t)}\right\|_2, \tag{H.33}$$

where the last inequality is from (3.3). Meanwhile, for the left-hand side of (H.32), we have

$$Q\big(\overline{\boldsymbol{\beta}}^{(t+0.5)};\boldsymbol{\beta}^*\big) \leq Q(\boldsymbol{\beta}^*;\boldsymbol{\beta}^*) + \big\langle \nabla_1 Q(\boldsymbol{\beta}^*;\boldsymbol{\beta}^*), \overline{\boldsymbol{\beta}}^{(t+0.5)} - \boldsymbol{\beta}^* \big\rangle - \nu/2 \cdot \big\|\overline{\boldsymbol{\beta}}^{(t+0.5)} - \boldsymbol{\beta}^*\big\|_2^2,$$
(H.34)

$$Q(\boldsymbol{\beta}^*;\boldsymbol{\beta}^*) \leq Q\big(\overline{\boldsymbol{\beta}}^{(t+0.5)};\boldsymbol{\beta}^*\big) + \big\langle \nabla_1 Q\big(\overline{\boldsymbol{\beta}}^{(t+0.5)};\boldsymbol{\beta}^*\big), \boldsymbol{\beta}^* - \overline{\boldsymbol{\beta}}^{(t+0.5)} \big\rangle - \nu/2 \cdot \big\|\overline{\boldsymbol{\beta}}^{(t+0.5)} - \boldsymbol{\beta}^*\big\|_2^2$$
(H.35)

by (3.6) in Condition *Concavity-Smoothness*$(\mu,\nu,\mathcal{B})$. By adding (H.34) and (H.35), we obtain

$$\nu \cdot \big\|\overline{\boldsymbol{\beta}}^{(t+0.5)} - \boldsymbol{\beta}^*\big\|_2^2 \leq \big\langle \boldsymbol{\beta}^* - \overline{\boldsymbol{\beta}}^{(t+0.5)}, \nabla_1 Q\big(\overline{\boldsymbol{\beta}}^{(t+0.5)};\boldsymbol{\beta}^*\big) - \nabla_1 Q(\boldsymbol{\beta}^*;\boldsymbol{\beta}^*)\big\rangle. \qquad \text{(H.36)}$$

Plugging (H.33) and (H.36) into (H.32), we obtain

$$\nu \cdot \big\|\overline{\boldsymbol{\beta}}^{(t+0.5)} - \boldsymbol{\beta}^*\big\|_2^2 \leq \gamma_1 \cdot \big\|\boldsymbol{\beta}^* - \overline{\boldsymbol{\beta}}^{(t+0.5)}\big\|_2 \cdot \big\|\boldsymbol{\beta}^* - \boldsymbol{\beta}^{(t)}\big\|_2,$$

which implies (G.8) in Lemma G.2.

**Proof of** (G.9)**:** We turn to prove (G.9). The self-consistency property in (H.29) implies that $\boldsymbol{\beta}^*$ is the maximizer of $Q(\cdot;\boldsymbol{\beta}^*)$. Furthermore, (3.5) and (3.6) in Condition *Concavity-Smoothness*$(\mu,\nu,\mathcal{B})$ ensure that $-Q(\cdot;\boldsymbol{\beta}^*)$ is $\mu$-smooth and $\nu$-strongly convex. By invoking standard optimization results for minimizing strongly convex and smooth objective functions, e.g., in [21], for stepsize $\eta = 2/(\nu+\mu)$, we have

$$\big\|\boldsymbol{\beta}^{(t)} + \eta \cdot \nabla_1 Q(\boldsymbol{\beta}^{(t)};\boldsymbol{\beta}^*) - \boldsymbol{\beta}^*\big\|_2 \leq \Big(\frac{\mu-\nu}{\mu+\nu}\Big) \cdot \big\|\boldsymbol{\beta}^{(t)} - \boldsymbol{\beta}^*\big\|_2, \qquad \text{(H.37)}$$

i.e., the gradient ascent step decreases the distance to $\boldsymbol{\beta}^*$ by a multiplicative factor. Hence, for the gradient ascent implementation of the M-step, i.e., $M(\cdot)$ defined in (3.2), we have

$$
\begin{aligned}
\big\|\overline{\boldsymbol{\beta}}^{(t+0.5)} - \boldsymbol{\beta}^*\big\|_2 &= \big\|M\big(\boldsymbol{\beta}^{(t)}\big) - \boldsymbol{\beta}^*\big\|_2 \\
&= \big\|\boldsymbol{\beta}^{(t)} + \eta \cdot \nabla_1 Q\big(\boldsymbol{\beta}^{(t)};\boldsymbol{\beta}^{(t)}\big) - \boldsymbol{\beta}^*\big\|_2 \\
&\leq \big\|\boldsymbol{\beta}^{(t)} + \eta \cdot \nabla_1 Q\big(\boldsymbol{\beta}^{(t)};\boldsymbol{\beta}^*\big) - \boldsymbol{\beta}^*\big\|_2 + \eta \cdot \big\|\nabla_1 Q\big(\boldsymbol{\beta}^{(t)};\boldsymbol{\beta}^*\big) - \nabla_1 Q\big(\boldsymbol{\beta}^{(t)};\boldsymbol{\beta}^{(t)}\big)\big\|_2 \\
&\leq \Big(\frac{\mu-\nu}{\mu+\nu}\Big) \cdot \big\|\boldsymbol{\beta}^{(t)} - \boldsymbol{\beta}^*\big\|_2 + \eta \cdot \gamma_2 \cdot \big\|\boldsymbol{\beta}^{(t)} - \boldsymbol{\beta}^*\big\|_2, \qquad \text{(H.38)}
\end{aligned}
$$

where the last inequality is from (H.37) and (3.4) in Condition *Lipschitz-Gradient*-$2(\gamma_2,\mathcal{B})$. Plugging $\eta = 2/(\nu+\mu)$ into (H.38), we obtain

$$\big\|\overline{\boldsymbol{\beta}}^{(t+0.5)} - \boldsymbol{\beta}^*\big\|_2 \leq \Big(\frac{\mu-\nu+2\cdot\gamma_2}{\mu+\nu}\Big) \cdot \big\|\boldsymbol{\beta}^{(t)} - \boldsymbol{\beta}^*\big\|_2,$$

which implies (G.9). Thus, we conclude the proof of Lemma G.2.

### H.3   Auxiliary Lemma for Proving Theorem 3.4

The following lemma characterizes the initialization step in line 2 of Algorithm 4.

**Lemma H.1.**  Suppose that we have $\big\|\boldsymbol{\beta}^{\mathrm{init}} - \boldsymbol{\beta}^*\big\|_2 \leq \kappa \cdot \|\boldsymbol{\beta}^*\|_2$ for some $\kappa \in (0,1)$. Assuming that $\widehat{s} \geq 4 \cdot (1+\kappa)^2/(1-\kappa)^2 \cdot s^*$, we have $\big\|\boldsymbol{\beta}^{(0)} - \boldsymbol{\beta}^*\big\|_2 \leq \big(1 + 4 \cdot \sqrt{s^*/\widehat{s}}\big)^{1/2} \cdot \big\|\boldsymbol{\beta}^{\mathrm{init}} - \boldsymbol{\beta}^*\big\|_2.$

*Proof.* Following the same proof of Lemma G.1 with both $\overline{\boldsymbol{\beta}}^{(t+0.5)}$ and $\boldsymbol{\beta}^{(t+0.5)}$ replaced with $\boldsymbol{\beta}^{\mathrm{init}}$, $\overline{\boldsymbol{\beta}}^{(t+1)}$ replaced with $\boldsymbol{\beta}^{(0)}$ and $\widehat{\mathcal{S}}^{(t+0.5)}$ replaced with $\widehat{\mathcal{S}}^{\mathrm{init}}$, we reach the conclusion. $\qquad\square$

### H.4   Proof of Lemma E.2

*Proof.* Recall that $Q(\cdot;\cdot)$ is the expectation of $Q_n(\cdot;\cdot)$. According to (A.2) and (3.1), we have

$$M(\boldsymbol{\beta}) = \mathbb{E}\big[2 \cdot \omega_{\boldsymbol{\beta}}(\boldsymbol{Y}) \cdot \boldsymbol{Y} - \boldsymbol{Y}\big]$$

with $\omega_{\boldsymbol{\beta}}(\cdot)$ being the weight function defined in (A.2), which together with (A.3) implies

$$M_n(\boldsymbol{\beta}) - M(\boldsymbol{\beta}) = \frac{1}{n}\sum_{i=1}^{n}\big[2 \cdot \omega_{\boldsymbol{\beta}}(\mathbf{y}_i) - 1\big] \cdot \mathbf{y}_i - \mathbb{E}\Big\{\big[2 \cdot \omega_{\boldsymbol{\beta}}(\boldsymbol{Y}) - 1\big] \cdot \boldsymbol{Y}\Big\}. \qquad \text{(H.39)}$$

Recall $\mathbf{y}_i$ is the $i$-th realization of $Y$, which follows the mixture distribution. For any $u > 0$, we have

$$\mathbb{E}\left\{\exp\left[u \cdot \left\|M_n(\boldsymbol{\beta}) - M(\boldsymbol{\beta})\right\|_\infty\right]\right\} = \mathbb{E}\left\{\max_{j \in \{1,\ldots,d\}} \exp\left[u \cdot \left|\left[M_n(\boldsymbol{\beta}) - M(\boldsymbol{\beta})\right]_j\right|\right]\right\}$$

$$\leq \sum_{j=1}^d \mathbb{E}\left\{\exp\left[u \cdot \left|\left[M_n(\boldsymbol{\beta}) - M(\boldsymbol{\beta})\right]_j\right|\right]\right\}. \qquad \text{(H.40)}$$

Based on (H.39), we apply the symmetrization result in Lemma J.4 to the right-hand side of (H.40). Then we have

$$\mathbb{E}\left\{\exp\left[u \cdot \left\|M_n(\boldsymbol{\beta}) - M(\boldsymbol{\beta})\right\|_\infty\right]\right\} \leq \sum_{j=1}^d \mathbb{E}\left\{\exp\left[u \cdot \left|\frac{1}{n}\sum_{i=1}^n \xi_i \cdot \left[2 \cdot \omega_{\boldsymbol{\beta}}(\mathbf{y}_i) - 1\right] \cdot y_{i,j}\right|\right]\right\},$$
$$\text{(H.41)}$$

where $\xi_1, \ldots, \xi_n$ are i.i.d. Rademacher random variables that are independent of $\mathbf{y}_1, \ldots, \mathbf{y}_n$. Then we invoke the contraction result in Lemma J.5 by setting

$$f(y_{i,j}) = y_{i,j}, \quad \mathcal{F} = \{f\}, \quad \psi_i(v) = \left[2 \cdot \omega_{\boldsymbol{\beta}}(\mathbf{y}_i) - 1\right] \cdot v, \quad \text{and} \quad \phi(v) = \exp(u \cdot v),$$

where $u$ is the variable of the moment generating function in (H.40). From the definition of $\omega_{\boldsymbol{\beta}}(\cdot)$ in (A.2) we have $\left|2 \cdot \omega_{\boldsymbol{\beta}}(\mathbf{y}_i) - 1\right| \leq 1$, which implies

$$\left|\psi_i(v) - \psi_i(v')\right| \leq \left|\left[2 \cdot \omega_{\boldsymbol{\beta}}(\mathbf{y}_i) - 1\right] \cdot (v - v')\right| \leq |v - v'|, \quad \text{for all } v, v' \in \mathbb{R}.$$

Therefore, by Lemma J.5 we obtain

$$\mathbb{E}\left\{\exp\left[u \cdot \left|\frac{1}{n}\sum_{i=1}^n \xi_i \cdot \left[2 \cdot \omega_{\boldsymbol{\beta}}(\mathbf{y}_i) - 1\right] \cdot y_{i,j}\right|\right]\right\} \leq \mathbb{E}\left\{\exp\left[2 \cdot u \cdot \left|\frac{1}{n}\sum_{i=1}^n \xi_i \cdot y_{i,j}\right|\right]\right\} \qquad \text{(H.42)}$$

for the right-hand side of (H.41), where $j \in \{1, \ldots, d\}$. Here note that in Gaussian mixture model we have $y_{i,j} = z_i \cdot \beta_j^* + v_{i,j}$, where $z_i$ is a Rademacher random variable and $v_{i,j} \sim N(0, \sigma^2)$. Therefore, according to Example 5.8 in [28] we have $\|z_i \cdot \beta_j^*\|_{\psi_2} \leq |\beta_j^*|$ and $\|v_{i,j}\|_{\psi_2} \leq C \cdot \sigma$. Hence by Lemma J.1 we have

$$\|y_{i,j}\|_{\psi_2} = \|z_i \cdot \beta_j^* + v_{i,j}\|_{\psi_2} \leq C \cdot \sqrt{|\beta_j^*|^2 + C' \cdot \sigma^2} \leq C \cdot \sqrt{\|\boldsymbol{\beta}^*\|_\infty^2 + C' \cdot \sigma^2}.$$

Since $|\xi_i \cdot y_{i,j}| = |y_{i,j}|$, $\xi_i \cdot y_{i,j}$ and $y_{i,j}$ have the same $\psi_2$-norm. Because $\xi_i$ is a Rademacher random variable independent of $y_{i,j}$, we have $\mathbb{E}(\xi_i \cdot y_{i,j}) = 0$. By Lemma 5.5 in [28], we obtain

$$\mathbb{E}\left[\exp(u' \cdot \xi_i \cdot y_{i,j})\right] \leq \exp\left[(u')^2 \cdot C \cdot \left(\|\boldsymbol{\beta}^*\|_\infty^2 + C' \cdot \sigma^2\right)\right], \quad \text{for all } u' \in \mathbb{R}. \qquad \text{(H.43)}$$

Hence, for the right-hand side of (H.42) we have

$$\mathbb{E}\left\{\exp\left[2 \cdot u \cdot \left|\frac{1}{n}\sum_{i=1}^n \xi_i \cdot y_{i,j}\right|\right]\right\} \leq \mathbb{E}\left(\max\left\{\exp\left[2 \cdot u \cdot \frac{1}{n}\sum_{i=1}^n \xi_i \cdot y_{i,j}\right], \exp\left[-2 \cdot u \cdot \frac{1}{n}\sum_{i=1}^n \xi_i \cdot y_{i,j}\right]\right\}\right)$$

$$\leq \mathbb{E}\left\{\exp\left[2 \cdot u \cdot \frac{1}{n}\sum_{i=1}^n \xi_i \cdot y_{i,j}\right]\right\} + \mathbb{E}\left\{\exp\left[-2 \cdot u \cdot \frac{1}{n}\sum_{i=1}^n \xi_i \cdot y_{i,j}\right]\right\}$$

$$\leq 2 \cdot \exp\left[C \cdot u^2 \cdot \left(\|\boldsymbol{\beta}^*\|_\infty^2 + C' \cdot \sigma^2\right)/n\right]. \qquad \text{(H.44)}$$

Here the last inequality is obtained by plugging (H.43) with $u' = 2 \cdot u/n$ and $u' = -2 \cdot u/n$ respectively into the two terms. Plugging (H.44) into (H.42) and then into (H.41), we obtain

$$\mathbb{E}\left\{\exp\left[u \cdot \left\|M_n(\boldsymbol{\beta}) - M(\boldsymbol{\beta})\right\|_\infty\right]\right\} \leq 2 \cdot d \cdot \exp\left[C \cdot u^2 \cdot \left(\|\boldsymbol{\beta}^*\|_\infty^2 + C' \cdot \sigma^2\right)/n\right].$$

By Chernoff bound we have that, for all $u > 0$ and $v > 0$,

$$\mathbb{P}\left[\left\|M_n(\boldsymbol{\beta}) - M(\boldsymbol{\beta})\right\|_\infty > v\right] \leq \mathbb{E}\left\{\exp\left[u \cdot \left\|M_n(\boldsymbol{\beta}) - M(\boldsymbol{\beta})\right\|_\infty\right]\right\}/\exp(u \cdot v)$$

$$\leq 2 \cdot \exp\left[C \cdot u^2 \cdot \left(\|\boldsymbol{\beta}^*\|_\infty^2 + C' \cdot \sigma^2\right)/n - u \cdot v + \log d\right].$$

Minimizing the right-hand side over $u$ we obtain

$$\mathbb{P}\left[\left\|M_n(\boldsymbol{\beta}) - M(\boldsymbol{\beta})\right\|_\infty > v\right] \leq 2 \cdot \exp\left\{-n \cdot v^2 \Big/ \left[4 \cdot C \cdot \left(\|\boldsymbol{\beta}^*\|_\infty^2 + C' \cdot \sigma^2\right)\right] + \log d\right\}.$$

Setting the right-hand side to be $\delta$, we have that

$$v = C \cdot \left(\|\boldsymbol{\beta}^*\|_\infty^2 + C' \cdot \sigma^2\right)^{1/2} \cdot \sqrt{\frac{\log d + \log(2/\delta)}{n}} \leq C'' \cdot \left(\|\boldsymbol{\beta}^*\|_\infty + \sigma\right) \cdot \sqrt{\frac{\log d + \log(2/\delta)}{n}}$$

holds for some constants $C$, $C'$ and $C''$, which completes the proof of Lemma E.2. $\qquad\square$

## H.5  Proof of Lemma E.5

In the sequel, we first establish the result for the maximization implementation of the M-step and then for the gradient ascent implementation.

**Maximization Implementation:** For the maximization implementation we need to estimate the inverse covariance matrix $\boldsymbol{\Theta}^* = \boldsymbol{\Sigma}^{-1}$ with the CLIME estimator $\widehat{\boldsymbol{\Theta}}$ defined in (A.7). The following lemma from [7] quantifies the statistical rate of convergence of $\widehat{\boldsymbol{\Theta}}$. Recall that $\|\cdot\|_{1,\infty}$ is defined as the maximum of the row $\ell_1$-norms of a matrix.

**Lemma H.2.** For $\boldsymbol{\Sigma} = \mathbf{I}_d$ and $\lambda^{\mathrm{CLIME}} = C \cdot \sqrt{\log d/n}$ in (A.7), we have that

$$\left\|\widehat{\boldsymbol{\Theta}} - \boldsymbol{\Theta}^*\right\|_{1,\infty} \leq C' \cdot \sqrt{\frac{\log d + \log(1/\delta)}{n}}$$

holds with probability at least $1 - \delta$, where $C$ and $C'$ are positive constants.

*Proof.* See the proof of Theorem 6 in [7] for details. $\qquad\square$

Now we are ready to prove (E.8) of Lemma E.5.

*Proof.* Recall that $Q(\cdot;\cdot)$ is the expectation of $Q_n(\cdot;\cdot)$. According to (A.5) and (3.1), we have

$$M(\boldsymbol{\beta}) = \mathbb{E}\Big\{\big[2 \cdot \omega_{\boldsymbol{\beta}}(\boldsymbol{X},Y) - 1\big] \cdot Y \cdot \boldsymbol{X}\Big\} \tag{H.45}$$

with $\omega_{\boldsymbol{\beta}}(\cdot,\cdot)$ being the weight function defined in (A.5), which together with (A.8) implies

$$M_n(\boldsymbol{\beta}) - M(\boldsymbol{\beta}) = \widehat{\boldsymbol{\Theta}} \cdot \frac{1}{n}\sum_{i=1}^{n}\big[2 \cdot \omega_{\boldsymbol{\beta}}(\mathbf{x}_i,y_i) - 1\big] \cdot y_i \cdot \mathbf{x}_i - \mathbb{E}\Big\{\big[2 \cdot \omega_{\boldsymbol{\beta}}(\boldsymbol{X},Y) - 1\big] \cdot Y \cdot \boldsymbol{X}\Big\}.$$

Here $\widehat{\boldsymbol{\Theta}}$ is the CLIME estimator defined in (A.7). For notational simplicity, we denote

$$\bar{\omega}_i = 2 \cdot \omega_{\boldsymbol{\beta}}(\mathbf{x}_i,y_i) - 1, \quad \text{and } \bar{\omega} = 2 \cdot \omega_{\boldsymbol{\beta}}(\boldsymbol{X},Y) - 1. \tag{H.46}$$

It is worth noting that both $\bar{\omega}_i$ and $\bar{\omega}$ depend on $\boldsymbol{\beta}$. Note that we have

$$\big\|M_n(\boldsymbol{\beta}) - M(\boldsymbol{\beta})\big\|_\infty \tag{H.47}$$

$$\leq \left\|\widehat{\boldsymbol{\Theta}} \cdot \left[\frac{1}{n}\sum_{i=1}^{n}\bar{\omega}_i \cdot y_i \cdot \mathbf{x}_i - \mathbb{E}(\bar{\omega} \cdot Y \cdot \boldsymbol{X})\right]\right\|_\infty + \left\|(\widehat{\boldsymbol{\Theta}} - \mathbf{I}_d) \cdot \mathbb{E}(\bar{\omega} \cdot Y \cdot \boldsymbol{X})\right\|_\infty$$

$$\leq \underbrace{\big\|\widehat{\boldsymbol{\Theta}}\big\|_{1,\infty}}_{(i)} \cdot \underbrace{\left\|\frac{1}{n}\sum_{i=1}^{n}\bar{\omega}_i \cdot y_i \cdot \mathbf{x}_i - \mathbb{E}(\bar{\omega} \cdot Y \cdot \boldsymbol{X})\right\|_\infty}_{(ii)} + \underbrace{\big\|\widehat{\boldsymbol{\Theta}} - \mathbf{I}_d\big\|_{1,\infty}}_{(iii)} \cdot \underbrace{\big\|\mathbb{E}(\bar{\omega} \cdot Y \cdot \boldsymbol{X})\big\|_\infty}_{(iv)}.$$

**Analysis of Term (i):** For term (i) in (H.47), recall that by our model assumption we have $\boldsymbol{\Sigma} = \mathbf{I}_d$, which implies $\boldsymbol{\Theta}^* = \boldsymbol{\Sigma}^{-1} = \mathbf{I}_d$. By Lemma H.2, for a sufficiently large sample size $n$, we have that

$$\big\|\widehat{\boldsymbol{\Theta}}\big\|_{1,\infty} \leq \big\|\widehat{\boldsymbol{\Theta}} - \mathbf{I}_d\big\|_{1,\infty} + \|\mathbf{I}_d\|_{1,\infty} \leq 1/2 + 1 = 3/2 \tag{H.48}$$

holds with probability at least $1 - \delta/4$.

**Analysis of Term (ii):** For term (ii) in (H.47), we have that for $u > 0$,

$$\mathbb{E}\left\{\exp\left[u \cdot \left\|\frac{1}{n}\sum_{i=1}^{n}\bar{\omega}_i \cdot y_i \cdot \mathbf{x}_i - \mathbb{E}(\bar{\omega} \cdot Y \cdot \boldsymbol{X})\right\|_{\infty}\right]\right\}$$

$$= \mathbb{E}\left\{\max_{j\in\{1,\ldots,d\}}\exp\left[u \cdot \left|\frac{1}{n}\sum_{i=1}^{n}\bar{\omega}_i \cdot y_i \cdot x_{i,j} - \mathbb{E}(\bar{\omega} \cdot Y \cdot X_j)\right|\right]\right\}$$

$$\leq \sum_{j=1}^{d}\mathbb{E}\left\{\exp\left[u \cdot \left|\frac{1}{n}\sum_{i=1}^{n}\bar{\omega}_i \cdot y_i \cdot x_{i,j} - \mathbb{E}(\bar{\omega} \cdot Y \cdot X_j)\right|\right]\right\}$$

$$\leq \sum_{j=1}^{d}\mathbb{E}\left\{\exp\left[u \cdot \left|\frac{1}{n}\sum_{i=1}^{n}\xi_i \cdot \bar{\omega}_i \cdot y_i \cdot x_{i,j}\right|\right]\right\}, \tag{H.49}$$

where $\xi_1, \ldots, \xi_n$ are i.i.d. Rademacher random variables. The last inequality follows from Lemma J.4. Furthermore, for the right-hand side of (H.49), we invoke the contraction result in Lemma J.5 by setting

$$f(y_i \cdot x_{i,j}) = y_i \cdot x_{i,j}, \quad \mathcal{F} = \{f\}, \quad \psi_i(v) = \bar{\omega}_i \cdot v, \quad \text{and} \quad \phi(v) = \exp(u \cdot v),$$

where $u$ is the variable of the moment generating function in (H.49). From the definitions in (A.5) and (H.46) we have $|\bar{\omega}_i| = |2 \cdot \omega_{\boldsymbol{\beta}}(\mathbf{x}_i, y_i) - 1| \leq 1$, which implies

$$\left|\psi_i(v) - \psi_i(v')\right| \leq \left|\left[2 \cdot \omega_{\boldsymbol{\beta}}(\mathbf{x}_i, y_i) - 1\right] \cdot (v - v')\right| \leq |v - v'|, \quad \text{for all } v, v' \in \mathbb{R}.$$

By Lemma J.5, we obtain

$$\mathbb{E}\left\{\exp\left[u \cdot \left|\frac{1}{n}\sum_{i=1}^{n}\xi_i \cdot \bar{\omega}_i \cdot y_i \cdot x_{i,j}\right|\right]\right\} \leq \mathbb{E}\left\{\exp\left[2 \cdot u \cdot \left|\frac{1}{n}\sum_{i=1}^{n}\xi_i \cdot y_i \cdot x_{i,j}\right|\right]\right\} \tag{H.50}$$

for $j \in \{1, \ldots, d\}$ on the right-hand side of (H.49). Recall that in mixture of regression model we have $y_i = z_i \cdot \langle\boldsymbol{\beta}^*, \mathbf{x}_i\rangle + v_i$, where $z_i$ is a Rademacher random variable, $v_i \sim N(0, \sigma^2)$, and $\mathbf{x}_i \sim N(\mathbf{0}, \mathbf{I}_d)$. Then by Example 5.8 in [28] we have $\|z_i \cdot \langle\boldsymbol{\beta}^*, \mathbf{x}_i\rangle\|_{\psi_2} = \|\langle\boldsymbol{\beta}^*, \mathbf{x}_i\rangle\|_{\psi_2} \leq C \cdot \|\boldsymbol{\beta}^*\|_2$ and $\|v_{i,j}\|_{\psi_2} \leq C' \cdot \sigma$. By Lemma J.1 we further have

$$\|y_i\|_{\psi_2} = \left\|z_i \cdot \langle\boldsymbol{\beta}^*, \mathbf{x}_i\rangle + v_i\right\|_{\psi_2} \leq \sqrt{C \cdot \|\boldsymbol{\beta}^*\|_2^2 + C' \cdot \sigma^2}.$$

Note that we have $\|x_{i,j}\|_{\psi_2} \leq C''$ since $x_{i,j} \sim N(0,1)$. Therefore, by Lemma J.2 we have

$$\|\xi_i \cdot y_i \cdot x_{i,j}\|_{\psi_1} = \|y_i \cdot x_{i,j}\|_{\psi_1} \leq \max\{C \cdot \|\boldsymbol{\beta}^*\|_2^2 + C' \cdot \sigma^2, \, C''\} \leq C''' \cdot \max\{\|\boldsymbol{\beta}^*\|_2^2 + \sigma^2, \, 1\}.$$

Since $\xi_i$ is a Rademacher random variable independent of $y_i \cdot x_{i,j}$, we have $\mathbb{E}(\xi_i \cdot y_i \cdot x_{i,j}) = 0$. Hence, by Lemma 5.15 in [28], we obtain

$$\mathbb{E}\left[\exp(u' \cdot \xi_i \cdot y_i \cdot x_{i,j})\right] \leq \exp\left[C \cdot (u')^2 \cdot \max\{\|\boldsymbol{\beta}^*\|_2^2 + \sigma^2, \, 1\}^2\right] \tag{H.51}$$

for all $|u'| \leq C'/\max\{\|\boldsymbol{\beta}^*\|_2^2 + \sigma^2, 1\}$. Hence we have

$$\mathbb{E}\left\{\exp\left[2 \cdot u \cdot \left|\frac{1}{n}\sum_{i=1}^{n}\xi_i \cdot y_i \cdot x_{i,j}\right|\right]\right\}$$

$$\leq \mathbb{E}\left(\max\left\{\exp\left[2 \cdot u \cdot \frac{1}{n}\sum_{i=1}^{n}\xi_i \cdot y_i \cdot x_{i,j}\right], \, \exp\left[-2 \cdot u \cdot \frac{1}{n}\sum_{i=1}^{n}\xi_i \cdot y_i \cdot x_{i,j}\right]\right\}\right)$$

$$\leq \mathbb{E}\left\{\exp\left[2 \cdot u \cdot \frac{1}{n}\sum_{i=1}^{n}\xi_i \cdot y_i \cdot x_{i,j}\right]\right\} + \mathbb{E}\left\{\exp\left[-2 \cdot u \cdot \frac{1}{n}\sum_{i=1}^{n}\xi_i \cdot y_i \cdot x_{i,j}\right]\right\}$$

$$\leq 2 \cdot \exp\left[C \cdot u^2 \cdot \max\{\|\boldsymbol{\beta}^*\|_2^2 + \sigma^2, \, 1\}^2/n\right]. \tag{H.52}$$

The last inequality is obtained by plugging (H.51) with $u' = 2 \cdot u/n$ and $u' = -2 \cdot u/n$ correspondingly into the two terms. Here $|u| \leq C' \cdot n/\max\{\|\boldsymbol{\beta}^*\|_2^2 + \sigma^2, 1\}$. Plugging (H.52) into (H.50) and further into (H.49), we obtain

$$\mathbb{E}\left\{\exp\left[u \cdot \left\|\frac{1}{n}\sum_{i=1}^{n}\bar{\omega}_i \cdot y_i \cdot \mathbf{x}_i - \mathbb{E}(\bar{\omega} \cdot Y \cdot \boldsymbol{X})\right\|_{\infty}\right]\right\} \leq 2 \cdot d \cdot \exp\left[C \cdot u^2 \cdot \max\{\|\boldsymbol{\beta}^*\|_2^2 + \sigma^2, \, 1\}^2/n\right].$$

By Chernoff bound we have that, for all $v > 0$ and $|u| \leq C' \cdot n / \max\{\|\boldsymbol{\beta}^*\|_2^2 + \sigma^2, 1\}$,

$$\mathbb{P}\left[\left\|\frac{1}{n}\sum_{i=1}^{n}\overline{\omega}_i \cdot y_i \cdot \mathbf{x}_i - \mathbb{E}(\overline{\omega} \cdot Y \cdot \boldsymbol{X})\right\|_\infty > v\right]$$

$$\leq \mathbb{E}\left\{\exp\left[u \cdot \left\|\frac{1}{n}\sum_{i=1}^{n}\overline{\omega}_i \cdot y_i \cdot \mathbf{x}_i - \mathbb{E}(\overline{\omega} \cdot Y \cdot \boldsymbol{X})\right\|_\infty\right]\right\} \bigg/ \exp(u \cdot v)$$

$$\leq 2 \cdot \exp\left[C \cdot u^2 \cdot \max\{\|\boldsymbol{\beta}^*\|_2^2 + \sigma^2, 1\}^2 \big/ n - u \cdot v + \log d\right].$$

Minimizing over $u$ on its right-hand side we have that, for $0 < v \leq C'' \cdot \max\{\|\boldsymbol{\beta}^*\|_2^2 + \sigma^2, 1\}$,

$$\mathbb{P}\left[\left\|\frac{1}{n}\sum_{i=1}^{n}\overline{\omega}_i \cdot y_i \cdot \mathbf{x}_i - \mathbb{E}(\overline{\omega} \cdot Y \cdot \boldsymbol{X})\right\|_\infty > v\right]$$

$$\leq 2 \cdot \exp\left\{-n \cdot v^2 \big/ \left[4 \cdot C \cdot \max\{\|\boldsymbol{\beta}^*\|_2^2 + \sigma^2, 1\}^2\right] + \log d\right\}.$$

Setting the right-hand side of the above inequality to be $\delta/2$, we have that

$$\left\|\frac{1}{n}\sum_{i=1}^{n}\overline{\omega}_i \cdot y_i \cdot \mathbf{x}_i - \mathbb{E}(\overline{\omega} \cdot Y \cdot \boldsymbol{X})\right\|_\infty \leq v = C \cdot \max\{\|\boldsymbol{\beta}^*\|_2^2 + \sigma^2, 1\} \cdot \sqrt{\frac{\log d + \log(4/\delta)}{n}} \tag{H.53}$$

holds with probability at least $1 - \delta/2$ for a sufficiently large $n$.

**Analysis of Term (iii):** For term (iii) in (H.47), by Lemma H.2 we have

$$\left\|\widehat{\boldsymbol{\Theta}} - \mathbf{I}_d\right\|_{1,\infty} \leq C \cdot \sqrt{\frac{\log d + \log(4/\delta)}{n}} \tag{H.54}$$

with probability at least $1 - \delta/4$ for a sufficiently large $n$.

**Analysis of Term (iv):** For term (iv) in (H.47), recall that by (H.45) and (H.46) we have

$$M(\boldsymbol{\beta}) = \mathbb{E}\left\{[2 \cdot \omega_{\boldsymbol{\beta}}(\boldsymbol{X}, Y) - 1] \cdot Y \cdot \boldsymbol{X}\right\} = \mathbb{E}(\overline{\omega} \cdot Y \cdot \boldsymbol{X}),$$

which implies

$$\left\|\mathbb{E}(\overline{\omega} \cdot Y \cdot \boldsymbol{X})\right\|_\infty = \left\|M(\boldsymbol{\beta})\right\|_\infty \leq \left\|M(\boldsymbol{\beta}) - \boldsymbol{\beta}^*\right\|_2 + \|\boldsymbol{\beta}^*\|_2$$
$$\leq \|\boldsymbol{\beta} - \boldsymbol{\beta}^*\|_2 + \|\boldsymbol{\beta}^*\|_2 \leq (1 + 1/32) \cdot \|\boldsymbol{\beta}^*\|_2, \tag{H.55}$$

where the first inequality follows from triangle inequality and $\|\cdot\|_\infty \leq \|\cdot\|_2$, the second inequality is from the proof of (G.8) in Lemma G.2 with $\overline{\boldsymbol{\beta}}^{(t+0.5)}$ replaced with $\boldsymbol{\beta}$ and the fact that $\gamma_1/\nu < 1$ in (G.8), and the third inequality holds since in Condition *Statistical-Error*$(\epsilon, \delta, s, n, \mathcal{B})$ we suppose that $\boldsymbol{\beta} \in \mathcal{B}$, and for mixture of regression model $\mathcal{B}$ is specified in (E.7).

Plugging (H.48), (H.53), (H.54) and (H.55) into (H.47), by union bound we have that

$$\left\|M_n(\boldsymbol{\beta}) - M(\boldsymbol{\beta})\right\|_\infty$$

$$\leq C \cdot \max\{\|\boldsymbol{\beta}^*\|_2^2 + \sigma^2, 1\} \cdot \sqrt{\frac{\log d + \log(4/\delta)}{n}} + C' \cdot \|\boldsymbol{\beta}^*\|_2 \cdot \sqrt{\frac{\log d + \log(4/\delta)}{n}}$$

$$\leq C'' \cdot \left[\max\{\|\boldsymbol{\beta}^*\|_2^2 + \sigma^2, 1\} + \|\boldsymbol{\beta}^*\|_2\right] \cdot \sqrt{\frac{\log d + \log(4/\delta)}{n}}$$

holds with probability at least $1 - \delta$. Therefore, we conclude the proof of (E.8) in Lemma E.5. $\square$

**Gradient Ascent Implementation:** In the following, we prove (E.9) in Lemma E.5.

*Proof.* Recall that $Q(\cdot; \cdot)$ is the expectation of $Q_n(\cdot; \cdot)$. According to (A.5) and (3.2), we have

$$M(\boldsymbol{\beta}) = \boldsymbol{\beta} + \eta \cdot \mathbb{E}\left[2 \cdot \omega_{\boldsymbol{\beta}}(\boldsymbol{X}, Y) \cdot Y \cdot \boldsymbol{X} - \boldsymbol{\beta}\right]$$

with $\omega_{\boldsymbol{\beta}}(\cdot,\cdot)$ being the weight function defined in (A.5), which together with (A.9) implies

$$\big\|M_n(\boldsymbol{\beta}) - M(\boldsymbol{\beta})\big\|_\infty \tag{H.56}$$

$$= \left\|\eta \cdot \frac{1}{n}\sum_{i=1}^n \big[2\cdot\omega_{\boldsymbol{\beta}}(\mathbf{x}_i,y_i)\cdot y_i\cdot\mathbf{x}_i - \mathbf{x}_i\cdot\mathbf{x}_i^\top\cdot\boldsymbol{\beta}\big] - \eta\cdot\mathbb{E}\big[2\cdot\omega_{\boldsymbol{\beta}}(\boldsymbol{X},Y)\cdot Y\cdot\boldsymbol{X} - \boldsymbol{\beta}\big]\right\|_\infty$$

$$\leq \eta \cdot \underbrace{\left\|\frac{1}{n}\sum_{i=1}^n\big[2\cdot\omega_{\boldsymbol{\beta}}(\mathbf{x}_i,y_i)\cdot y_i\cdot\mathbf{x}_i\big] - \mathbb{E}\big[2\cdot\omega_{\boldsymbol{\beta}}(\boldsymbol{X},Y)\cdot Y\cdot\boldsymbol{X}\big]\right\|_\infty}_{\text{(i)}} + \eta\cdot\underbrace{\left\|\frac{1}{n}\sum_{i=1}^n\mathbf{x}_i\cdot\mathbf{x}_i^\top\cdot\boldsymbol{\beta} - \boldsymbol{\beta}\right\|_\infty}_{\text{(ii)}}.$$

Here $\eta > 0$ denotes the stepsize in Algorithm 3.

**Analysis of Term (i):** For term (i) in (H.56), we redefine $\bar{\omega}_i$ and $\bar{\omega}$ in (H.46) as

$$\bar{\omega}_i = 2\cdot\omega_{\boldsymbol{\beta}}(\mathbf{x}_i,y_i),\quad\text{and }\bar{\omega} = 2\cdot\omega_{\boldsymbol{\beta}}(\boldsymbol{X},Y). \tag{H.57}$$

Note that $|\bar{\omega}_i| = \big|2\cdot\omega_{\boldsymbol{\beta}}(\mathbf{x}_i,y_i)\big| \leq 2$. Following the same way we establish the upper bound of term (ii) in (H.47), under the new definitions of $\bar{\omega}_i$ and $\bar{\omega}$ in (H.57) we have that

$$\left\|\frac{1}{n}\sum_{i=1}^n\big[2\cdot\omega_{\boldsymbol{\beta}}(\mathbf{x}_i,y_i)\cdot y_i\cdot\mathbf{x}_i\big] - \mathbb{E}\big[2\cdot\omega_{\boldsymbol{\beta}}(\boldsymbol{X},Y)\cdot Y\cdot\boldsymbol{X}\big]\right\|_\infty$$

$$\leq C\cdot\max\big\{\|\boldsymbol{\beta}^*\|_2^2 + \sigma^2,\ 1\big\}\cdot\sqrt{\frac{\log d + \log(4/\delta)}{n}}$$

holds with probability at least $1 - \delta/2$.

**Analysis of Term (ii):** For term (ii) in (H.56), we have

$$\left\|\frac{1}{n}\sum_{i=1}^n\mathbf{x}_i\cdot\mathbf{x}_i^\top\cdot\boldsymbol{\beta} - \boldsymbol{\beta}\right\|_\infty \leq \underbrace{\left\|\frac{1}{n}\sum_{i=1}^n\mathbf{x}_i\cdot\mathbf{x}_i^\top - \mathbf{I}_d\right\|_{\infty,\infty}}_{\text{(ii).a}}\cdot\underbrace{\|\boldsymbol{\beta}\|_1}_{\text{(ii).b}}.$$

For term (ii).a, recall by our model assumption we have $\mathbb{E}\big(\boldsymbol{X}\cdot\boldsymbol{X}^\top\big) = \mathbf{I}_d$ and $\mathbf{x}_i$'s are the independent realizations of $\boldsymbol{X}$. Hence we have

$$\left\|\frac{1}{n}\sum_{i=1}^n\mathbf{x}_i\cdot\mathbf{x}_i^\top - \mathbf{I}_d\right\|_{\infty,\infty} \leq \max_{j\in\{1,\dots,d\}}\max_{k\in\{1,\dots,d\}}\left|\frac{1}{n}\sum_{i=1}^n x_{i,j}\cdot x_{i,k} - \mathbb{E}(X_j\cdot X_k)\right|.$$

Since $X_j, X_k \sim N(0,1)$, according to Example 5.8 in [28] we have $\|X_j\|_{\psi_2} = \|X_k\|_{\psi_2} \leq C$. By Lemma J.2, $X_j\cdot X_k$ is a sub-exponential random variable with $\|X_j\cdot X_k\|_{\psi_1} \leq C'$. Moreover, we have $\big\|X_j\cdot X_k - \mathbb{E}(X_j\cdot X_k)\big\|_{\psi_1} \leq C''$ by Lemma J.3. Then by Bernstein's inequality (Proposition 5.16 in [28]) and union bound, we have

$$\mathbb{P}\left[\left\|\frac{1}{n}\sum_{i=1}^n\mathbf{x}_i\cdot\mathbf{x}_i^\top - \mathbf{I}_d\right\|_{\infty,\infty} > v\right] \leq 2\cdot d^2\cdot\exp\big(-C\cdot n\cdot v^2\big)$$

for $0 < v \leq C'$ and a sufficiently large sample size $n$. Setting its right-hand side to be $\delta/2$, we have

$$\left\|\frac{1}{n}\sum_{i=1}^n\mathbf{x}_i\cdot\mathbf{x}_i^\top - \mathbf{I}_d\right\|_{\infty,\infty} \leq C''\cdot\sqrt{\frac{2\cdot\log d + \log(4/\delta)}{n}}$$

holds with probability at least $1 - \delta/2$. For term (ii).b we have $\|\boldsymbol{\beta}\|_1 \leq \sqrt{s}\cdot\|\boldsymbol{\beta}\|_2$, since in Condition *Statistical-Error*$(\epsilon,\delta,s,n,\mathcal{B})$ we assume $\|\boldsymbol{\beta}\|_0 \leq s$. Furthermore, we have $\|\boldsymbol{\beta}\|_2 \leq \|\boldsymbol{\beta}^*\|_2 + \|\boldsymbol{\beta}^* - \boldsymbol{\beta}\|_2 \leq (1+1/32)\cdot\|\boldsymbol{\beta}^*\|_2$, because in Condition *Statistical-Error*$(\epsilon,\delta,s,n,\mathcal{B})$ we assume that $\boldsymbol{\beta}\in\mathcal{B}$, and for mixture of regression model $\mathcal{B}$ is specified in (E.7).

Plugging the above results into the right-hand side of (H.56), by union bound we have that

$$\big\|M_n(\boldsymbol{\beta}) - M(\boldsymbol{\beta})\big\|_\infty \leq \eta\cdot C\cdot\max\big\{\|\boldsymbol{\beta}^*\|_2^2 + \sigma^2,\ 1\big\}\cdot\sqrt{\frac{\log d + \log(4/\delta)}{n}}$$

$$+ \eta\cdot C'\cdot\sqrt{\frac{2\cdot\log d + \log(4/\delta)}{n}}\cdot\sqrt{s}\cdot\|\boldsymbol{\beta}^*\|_2$$

$$\leq \eta\cdot C''\cdot\max\big\{\|\boldsymbol{\beta}^*\|_2^2 + \sigma^2,\ 1,\ \sqrt{s}\cdot\|\boldsymbol{\beta}^*\|_2\big\}\cdot\sqrt{\frac{\log d + \log(4/\delta)}{n}}$$

holds with probability at least $1 - \delta$. Therefore, we conclude the proof of (E.9) in Lemma E.5. $\quad\square$

# I  Proof of Results for Inference

In the following, we provide the detailed proof of the theoretical results for asymptotic inference in §4. We first present the proof of the general results, and then the proof for specific models.

## I.1  Proof of Theorem 2.1

*Proof.* In the sequel we establish the two equations in (2.8) respectively.

**Proof of the First Equation:** According to the definition of the lower bound function $Q_n(\cdot;\cdot)$ in (2.1), we have

$$Q_n(\boldsymbol{\beta}';\boldsymbol{\beta}) = \frac{1}{n}\sum_{i=1}^n \int_{\mathcal{Z}} k_{\boldsymbol{\beta}}(\mathbf{z}\mid\mathbf{y}_i)\cdot\log f_{\boldsymbol{\beta}'}(\mathbf{y}_i,\mathbf{z})\,\mathrm{d}\mathbf{z}. \tag{I.1}$$

Here $k_{\boldsymbol{\beta}}(\mathbf{z}\mid\mathbf{y}_i)$ is the density of the latent variable $\mathbf{Z}$ conditioning on the observed variable $\mathbf{Y}=\mathbf{y}_i$ under the model with parameter $\boldsymbol{\beta}$. Hence we obtain

$$\nabla_1 Q_n(\boldsymbol{\beta};\boldsymbol{\beta}) = \frac{1}{n}\sum_{i=1}^n \int_{\mathcal{Z}} k_{\boldsymbol{\beta}}(\mathbf{z}\mid\mathbf{y}_i)\cdot\frac{\partial f_{\boldsymbol{\beta}}(\mathbf{y}_i,\mathbf{z})/\partial\boldsymbol{\beta}}{f_{\boldsymbol{\beta}}(\mathbf{y}_i,\mathbf{z})}\,\mathrm{d}\mathbf{z} = \frac{1}{n}\sum_{i=1}^n \int_{\mathcal{Z}} \frac{\partial f_{\boldsymbol{\beta}}(\mathbf{y}_i,\mathbf{z})/\partial\boldsymbol{\beta}}{h_{\boldsymbol{\beta}}(\mathbf{y}_i)}\,\mathrm{d}\mathbf{z}, \tag{I.2}$$

where $h_{\boldsymbol{\beta}}(\mathbf{y}_i)$ is the marginal density function of $\mathbf{Y}$ evaluated at $\mathbf{y}_i$, and the second equality follows from the fact that

$$k_{\boldsymbol{\beta}}(\mathbf{z}\mid\mathbf{y}_i) = f_{\boldsymbol{\beta}}(\mathbf{y}_i,\mathbf{z})/h_{\boldsymbol{\beta}}(\mathbf{y}_i), \tag{I.3}$$

since $k_{\boldsymbol{\beta}}(\mathbf{z}\mid\mathbf{y}_i)$ is the conditional density. According to the definition in (B.3), we have

$$\nabla\ell_n(\boldsymbol{\beta}) = \sum_{i=1}^n \frac{\partial\log h_{\boldsymbol{\beta}}(\mathbf{y}_i)}{\partial\boldsymbol{\beta}} = \sum_{i=1}^n \frac{\partial h_{\boldsymbol{\beta}}(\mathbf{y}_i)/\partial\boldsymbol{\beta}}{h_{\boldsymbol{\beta}}(\mathbf{y}_i)} = \sum_{i=1}^n \int_{\mathcal{Z}} \frac{\partial f_{\boldsymbol{\beta}}(\mathbf{y}_i,\mathbf{z})/\partial\boldsymbol{\beta}}{h_{\boldsymbol{\beta}}(\mathbf{y}_i)}\,\mathrm{d}\mathbf{z}, \tag{I.4}$$

where the last equality is from (B.1). Comparing (I.2) and (I.4), we obtain $\nabla_1 Q_n(\boldsymbol{\beta};\boldsymbol{\beta}) = \nabla\ell_n(\boldsymbol{\beta})/n$.

**Proof of the Second Equation:** For the second equation in (D.1), by (I.1) and (I.3) we have

$$Q_n(\boldsymbol{\beta}';\boldsymbol{\beta}) = \frac{1}{n}\sum_{i=1}^n \int_{\mathcal{Z}} \frac{f_{\boldsymbol{\beta}}(\mathbf{y}_i,\mathbf{z})}{h_{\boldsymbol{\beta}}(\mathbf{y}_i)}\cdot\log f_{\boldsymbol{\beta}'}(\mathbf{y}_i,\mathbf{z})\,\mathrm{d}\mathbf{z}.$$

By calculation we obtain

$$\nabla_{1,2}^2 Q_n(\boldsymbol{\beta};\boldsymbol{\beta}) \tag{I.5}$$

$$= \frac{1}{n}\sum_{i=1}^n \int_{\mathcal{Z}} \frac{\partial f_{\boldsymbol{\beta}}(\mathbf{y}_i,\mathbf{z})/\partial\boldsymbol{\beta}}{f_{\boldsymbol{\beta}}(\mathbf{y}_i,\mathbf{z})}\otimes\left\{\frac{\partial f_{\boldsymbol{\beta}}(\mathbf{y}_i,\mathbf{z})/\partial\boldsymbol{\beta}\cdot h_{\boldsymbol{\beta}}(\mathbf{y}_i)}{\big[h_{\boldsymbol{\beta}}(\mathbf{y}_i)\big]^2} - \frac{f_{\boldsymbol{\beta}}(\mathbf{y}_i,\mathbf{z})\cdot\partial h_{\boldsymbol{\beta}}(\mathbf{y}_i)/\partial\boldsymbol{\beta}}{\big[h_{\boldsymbol{\beta}}(\mathbf{y}_i)\big]^2}\right\}\,\mathrm{d}\mathbf{z}.$$

Here $\otimes$ denotes the vector outer product. Note that in (I.5) we have

$$\int_{\mathcal{Z}} \frac{\partial f_{\boldsymbol{\beta}}(\mathbf{y}_i,\mathbf{z})/\partial\boldsymbol{\beta}}{f_{\boldsymbol{\beta}}(\mathbf{y}_i,\mathbf{z})}\otimes\frac{\partial f_{\boldsymbol{\beta}}(\mathbf{y}_i,\mathbf{z})/\partial\boldsymbol{\beta}}{h_{\boldsymbol{\beta}}(\mathbf{y}_i)}\,\mathrm{d}\mathbf{z} = \int_{\mathcal{Z}} \left[\frac{\partial f_{\boldsymbol{\beta}}(\mathbf{y}_i,\mathbf{z})/\partial\boldsymbol{\beta}}{f_{\boldsymbol{\beta}}(\mathbf{y}_i,\mathbf{z})}\right]^{\otimes 2}\cdot\frac{f_{\boldsymbol{\beta}}(\mathbf{y}_i,\mathbf{z})}{h_{\boldsymbol{\beta}}(\mathbf{y}_i)}\,\mathrm{d}\mathbf{z}$$

$$= \int_{\mathcal{Z}} \left[\frac{\partial f_{\boldsymbol{\beta}}(\mathbf{y}_i,\mathbf{z})/\partial\boldsymbol{\beta}}{f_{\boldsymbol{\beta}}(\mathbf{y}_i,\mathbf{z})}\right]^{\otimes 2}\cdot k_{\boldsymbol{\beta}}(\mathbf{z}\mid\mathbf{y}_i)\,\mathrm{d}\mathbf{z}$$

$$= \mathbb{E}_{\boldsymbol{\beta}}\left[\widetilde{S}_{\boldsymbol{\beta}}(\mathbf{Y},\mathbf{Z})^{\otimes 2}\mid\mathbf{Y}=\mathbf{y}_i\right], \tag{I.6}$$

where $\mathbf{v}^{\otimes 2}$ denotes $\mathbf{v}\otimes\mathbf{v}$. Here $\widetilde{S}_{\boldsymbol{\beta}}(\cdot,\cdot)$ is defined as

$$\widetilde{S}_{\boldsymbol{\beta}}(\mathbf{y},\mathbf{z}) = \frac{\partial\log f_{\boldsymbol{\beta}}(\mathbf{y},\mathbf{z})}{\partial\boldsymbol{\beta}} = \frac{\partial f_{\boldsymbol{\beta}}(\mathbf{y},\mathbf{z})/\partial\boldsymbol{\beta}}{f_{\boldsymbol{\beta}}(\mathbf{y},\mathbf{z})} \in \mathbb{R}^d, \tag{I.7}$$

i.e., the score function for the complete likelihood, which involves both the observed variable $\boldsymbol{Y}$ and the latent variable $\boldsymbol{Z}$. Meanwhile, in (I.5) we have

$$\int_{\mathcal{Z}} \frac{\partial f_{\boldsymbol{\beta}}(\mathbf{y}_i, \mathbf{z})/\partial\boldsymbol{\beta}}{f_{\boldsymbol{\beta}}(\mathbf{y}_i, \mathbf{z})} \otimes \frac{f_{\boldsymbol{\beta}}(\mathbf{y}_i, \mathbf{z}) \cdot \partial h_{\boldsymbol{\beta}}(\mathbf{y}_i)/\partial\boldsymbol{\beta}}{\left[h_{\boldsymbol{\beta}}(\mathbf{y}_i)\right]^2} \, \mathrm{d}\mathbf{z} = \left[\int_{\mathcal{Z}} \frac{\partial f_{\boldsymbol{\beta}}(\mathbf{y}_i, \mathbf{z})/\partial\boldsymbol{\beta}}{h_{\boldsymbol{\beta}}(\mathbf{y}_i)} \, \mathrm{d}\mathbf{z}\right] \otimes \frac{\partial h_{\boldsymbol{\beta}}(\mathbf{y}_i)/\partial\boldsymbol{\beta}}{h_{\boldsymbol{\beta}}(\mathbf{y}_i)}$$

$$= \left[\int_{\mathcal{Z}} \frac{\partial f_{\boldsymbol{\beta}}(\mathbf{y}_i, \mathbf{z})/\partial\boldsymbol{\beta}}{h_{\boldsymbol{\beta}}(\mathbf{y}_i)} \, \mathrm{d}\mathbf{z}\right]^{\otimes 2}, \qquad (\text{I.8})$$

where in the last equality we utilize the fact that

$$\int_{\mathcal{Z}} f_{\boldsymbol{\beta}}(\mathbf{y}_i, \mathbf{z}) \, \mathrm{d}\mathbf{z} = h_{\boldsymbol{\beta}}(\mathbf{y}_i), \qquad (\text{I.9})$$

because $h_{\boldsymbol{\beta}}(\cdot)$ is the marginal density function of $\boldsymbol{Y}$. By (I.3) and (I.7), for the right-hand side of (I.8) we have

$$\mathbb{E}_{\boldsymbol{\beta}}\left[\widetilde{S}_{\boldsymbol{\beta}}(\boldsymbol{Y}, \boldsymbol{Z}) \mid \boldsymbol{Y} = \mathbf{y}_i\right] = \int_{\mathcal{Z}} \frac{\partial f_{\boldsymbol{\beta}}(\mathbf{y}_i, \mathbf{z})/\partial\boldsymbol{\beta}}{f_{\boldsymbol{\beta}}(\mathbf{y}_i, \mathbf{z})} \cdot k_{\boldsymbol{\beta}}(\mathbf{z} \mid \mathbf{y}_i) \, \mathrm{d}\mathbf{z} = \int_{\mathcal{Z}} \frac{\partial f_{\boldsymbol{\beta}}(\mathbf{y}_i, \mathbf{z})/\partial\boldsymbol{\beta}}{h_{\boldsymbol{\beta}}(\mathbf{y}_i)} \, \mathrm{d}\mathbf{z}. \quad (\text{I.10})$$

Plugging (I.10) into (I.8) and then plugging (I.6) and (I.8) into (I.5) we obtain

$$\nabla_{1,2}^2 Q_n(\boldsymbol{\beta}; \boldsymbol{\beta}) = \frac{1}{n} \sum_{i=1}^{n} \left( \mathbb{E}_{\boldsymbol{\beta}}\left[\widetilde{S}_{\boldsymbol{\beta}}(\boldsymbol{Y}, \boldsymbol{Z})^{\otimes 2} \mid \boldsymbol{Y} = \mathbf{y}_i\right] - \left\{\mathbb{E}_{\boldsymbol{\beta}}\left[\widetilde{S}_{\boldsymbol{\beta}}(\boldsymbol{Y}, \boldsymbol{Z}) \mid \boldsymbol{Y} = \mathbf{y}_i\right]\right\}^{\otimes 2} \right).$$

Setting $\boldsymbol{\beta} = \boldsymbol{\beta}^*$ in the above equality, we obtain

$$\mathbb{E}_{\boldsymbol{\beta}^*}\left[\nabla_{1,2}^2 Q_n(\boldsymbol{\beta}^*; \boldsymbol{\beta}^*)\right] = \mathbb{E}_{\boldsymbol{\beta}^*}\left\{\text{Cov}_{\boldsymbol{\beta}^*}\left[\widetilde{S}_{\boldsymbol{\beta}^*}(\boldsymbol{Y}, \boldsymbol{Z}) \mid \boldsymbol{Y}\right]\right\}. \qquad (\text{I.11})$$

Meanwhile, for $\boldsymbol{\beta} = \boldsymbol{\beta}^*$, by the property of Fisher information we have

$$I(\boldsymbol{\beta}^*) = \text{Cov}_{\boldsymbol{\beta}^*}\left[\frac{\partial \log h_{\boldsymbol{\beta}^*}(\boldsymbol{Y})}{\partial\boldsymbol{\beta}}\right] = \text{Cov}_{\boldsymbol{\beta}^*}\left\{\mathbb{E}_{\boldsymbol{\beta}^*}\left[\widetilde{S}_{\boldsymbol{\beta}^*}(\boldsymbol{Y}, \boldsymbol{Z}) \mid \boldsymbol{Y}\right]\right\}. \qquad (\text{I.12})$$

Here the last equality is obtained by taking $\boldsymbol{\beta} = \boldsymbol{\beta}^*$ in

$$\frac{\partial \log h_{\boldsymbol{\beta}}(\boldsymbol{Y})}{\partial\boldsymbol{\beta}} = \frac{\partial h_{\boldsymbol{\beta}}(\boldsymbol{Y})}{\partial\boldsymbol{\beta}} \cdot \frac{1}{h_{\boldsymbol{\beta}}(\boldsymbol{Y})} = \int_{\mathcal{Z}} \frac{\partial f_{\boldsymbol{\beta}}(\boldsymbol{Y}, \mathbf{z})/\partial\boldsymbol{\beta}}{h_{\boldsymbol{\beta}}(\boldsymbol{Y})} \, \mathrm{d}\mathbf{z} = \int_{\mathcal{Z}} \frac{\partial f_{\boldsymbol{\beta}}(\boldsymbol{Y}, \mathbf{z})/\partial\boldsymbol{\beta}}{f_{\boldsymbol{\beta}}(\boldsymbol{Y}, \mathbf{z})} \cdot k_{\boldsymbol{\beta}}(\mathbf{z} \mid \boldsymbol{Y}) \, \mathrm{d}\mathbf{z}$$

$$= \int_{\mathcal{Z}} \widetilde{S}_{\boldsymbol{\beta}}(\boldsymbol{Y}, \mathbf{z}) \cdot k_{\boldsymbol{\beta}}(\mathbf{z} \mid \boldsymbol{Y}) \, \mathrm{d}\mathbf{z}$$

$$= \mathbb{E}_{\boldsymbol{\beta}}\left[\widetilde{S}_{\boldsymbol{\beta}}(\boldsymbol{Y}, \boldsymbol{Z}) \mid \boldsymbol{Y}\right],$$

where the second equality follows from (I.9), the third equality follows from (I.3), while the second last equality follows from (I.7). Combining (I.11) and (I.12), by the law of total variance we have

$$I(\boldsymbol{\beta}^*) + \mathbb{E}_{\boldsymbol{\beta}^*}\left[\nabla_{1,2}^2 Q_n(\boldsymbol{\beta}^*; \boldsymbol{\beta}^*)\right] = \text{Cov}_{\boldsymbol{\beta}^*}\left\{\mathbb{E}_{\boldsymbol{\beta}^*}\left[\widetilde{S}_{\boldsymbol{\beta}^*}(\boldsymbol{Y}, \boldsymbol{Z}) \mid \boldsymbol{Y}\right]\right\} + \mathbb{E}_{\boldsymbol{\beta}^*}\left\{\text{Cov}_{\boldsymbol{\beta}^*}\left[\widetilde{S}_{\boldsymbol{\beta}^*}(\boldsymbol{Y}, \boldsymbol{Z}) \mid \boldsymbol{Y}\right]\right\}$$

$$= \text{Cov}_{\boldsymbol{\beta}^*}\left[\widetilde{S}_{\boldsymbol{\beta}^*}(\boldsymbol{Y}, \boldsymbol{Z})\right]. \qquad (\text{I.13})$$

In the following, we prove

$$\mathbb{E}_{\boldsymbol{\beta}^*}\left[\nabla_{1,1}^2 Q_n(\boldsymbol{\beta}^*; \boldsymbol{\beta}^*)\right] = -\text{Cov}_{\boldsymbol{\beta}^*}\left[\widetilde{S}_{\boldsymbol{\beta}^*}(\boldsymbol{Y}, \boldsymbol{Z})\right]. \qquad (\text{I.14})$$

According to (I.1) we have

$$\nabla_{1,1}^2 Q_n(\boldsymbol{\beta}; \boldsymbol{\beta}) = \frac{1}{n} \sum_{i=1}^{n} \int_{\mathcal{Z}} k_{\boldsymbol{\beta}}(\mathbf{z} \mid \mathbf{y}_i) \cdot \frac{\partial^2 \log f_{\boldsymbol{\beta}}(\mathbf{y}_i, \mathbf{z})}{\partial^2\boldsymbol{\beta}} \, \mathrm{d}\mathbf{z} = \frac{1}{n} \sum_{i=1}^{n} \mathbb{E}_{\boldsymbol{\beta}}\left[\frac{\partial^2 \log f_{\boldsymbol{\beta}}(\boldsymbol{Y}, \boldsymbol{Z})}{\partial^2\boldsymbol{\beta}} \,\Big|\, \boldsymbol{Y} = \mathbf{y}_i\right].$$

$$(\text{I.15})$$

Let $\widetilde{\ell}(\boldsymbol{\beta}) = \log f_{\boldsymbol{\beta}}(\boldsymbol{Y}, \boldsymbol{Z})$ be the complete log-likelihood, which involves both the observed variable $\boldsymbol{Y}$ and the latent variable $\boldsymbol{Z}$, and $\widetilde{I}(\boldsymbol{\beta})$ be the corresponding Fisher information. By setting $\boldsymbol{\beta} = \boldsymbol{\beta}^*$ in (I.15) and taking expectation, we obtain

$$\mathbb{E}_{\boldsymbol{\beta}^*}\left[\nabla_{1,1}^2 Q_n(\boldsymbol{\beta}^*; \boldsymbol{\beta}^*)\right] = \mathbb{E}_{\boldsymbol{\beta}^*}\left\{\mathbb{E}_{\boldsymbol{\beta}^*}\left[\frac{\partial^2 \log f_{\boldsymbol{\beta}^*}(\boldsymbol{Y}, \boldsymbol{Z})}{\partial^2\boldsymbol{\beta}} \,\Big|\, \boldsymbol{Y}\right]\right\} = \mathbb{E}_{\boldsymbol{\beta}^*}\left[\nabla^2\widetilde{\ell}(\boldsymbol{\beta}^*)\right] = -\widetilde{I}(\boldsymbol{\beta}^*).$$

$$(\text{I.16})$$

Since $\widetilde{S}_{\boldsymbol{\beta}}(\boldsymbol{Y}, \boldsymbol{Z})$ defined in (I.7) is the score function for the complete log-likelihood $\widetilde{\ell}(\boldsymbol{\beta})$, according to the relationship between the score function and Fisher information, we have

$$\widetilde{I}(\boldsymbol{\beta}^*) = \mathrm{Cov}_{\boldsymbol{\beta}^*}\left[\widetilde{S}_{\boldsymbol{\beta}^*}(\boldsymbol{Y}, \boldsymbol{Z})\right],$$

which together with (I.16) implies (I.14). By further plugging (I.14) into (I.13), we obtain

$$\mathbb{E}_{\boldsymbol{\beta}^*}\left[\nabla_{1,1}^2 Q_n(\boldsymbol{\beta}^*; \boldsymbol{\beta}^*) + \nabla_{1,2}^2 Q_n(\boldsymbol{\beta}^*; \boldsymbol{\beta}^*)\right] = -I(\boldsymbol{\beta}^*),$$

which establishes the first equality of the second equation in (D.1). In addition, the second equality of the second equation in (D.1) follows from the property of Fisher information. Thus, we conclude the proof of Theorem D.1. □

## I.2 Auxiliary Lemmas for Proving Theorem 4.6

In this section, we lay out several lemmas on the Dantzig selector defined in (2.6). The first lemma, which is from [5], characterizes the cone condition for the Dantzig selector.

**Lemma I.1.** Any feasible solution $\mathbf{w}$ in (2.6) satisfies

$$\left\|\left[w(\boldsymbol{\beta}, \lambda) - \mathbf{w}\right]_{\overline{\mathcal{S}_{\mathbf{w}}}}\right\|_1 \leq \left\|\left[w(\boldsymbol{\beta}, \lambda) - \mathbf{w}\right]_{\mathcal{S}_{\mathbf{w}}}\right\|_1,$$

where $w(\boldsymbol{\beta}, \lambda)$ is the minimizer of (2.6), $\mathcal{S}_{\mathbf{w}}$ is the support of $\mathbf{w}$ and $\overline{\mathcal{S}_{\mathbf{w}}}$ is its complement.

*Proof.* See Lemma B.3 in [5] for a detailed proof. □

In the sequel, we focus on analyzing $w(\widehat{\boldsymbol{\beta}}, \lambda)$. The results for $w(\widehat{\boldsymbol{\beta}}_0, \lambda)$ can be obtained similarly. The next lemma characterizes the restricted eigenvalue of $T_n(\widehat{\boldsymbol{\beta}})$, which is defined as

$$\widehat{\rho}_{\min} = \inf_{\mathbf{v} \in \mathcal{C}} \frac{\mathbf{v}^\top \cdot \left[-T_n(\widehat{\boldsymbol{\beta}})\right] \cdot \mathbf{v}}{\|\mathbf{v}\|_2^2}, \quad \text{where } \mathcal{C} = \left\{\mathbf{v} : \left\|\mathbf{v}_{\overline{\mathcal{S}_{\mathbf{w}}^*}}\right\|_1 \leq \left\|\mathbf{v}_{\mathcal{S}_{\mathbf{w}}^*}\right\|_1, \ \mathbf{v} \neq \mathbf{0}\right\}. \quad \text{(I.17)}$$

Here $\mathcal{S}_{\mathbf{w}}^*$ is the support of $\mathbf{w}^*$ defined in (4.1).

**Lemma I.2.** Under Assumption 4.5 and Conditions 4.1, 4.3 and 4.4, for a sufficiently large sample size $n$, we have $\widehat{\rho}_{\min} \geq \rho_{\min}/2 > 0$ with high probability, where $\rho_{\min}$ is specified in (4.4).

*Proof.* By triangle inequality we have

$$\widehat{\rho}_{\min} \geq \inf_{\mathbf{v} \in \mathcal{C}} \frac{\mathbf{v}^\top \cdot \left[-T_n(\widehat{\boldsymbol{\beta}})\right] \cdot \mathbf{v}}{\|\mathbf{v}\|_2^2} \geq \inf_{\mathbf{v} \in \mathcal{C}} \frac{\mathbf{v}^\top \cdot I(\boldsymbol{\beta}^*) \cdot \mathbf{v} - \left|\mathbf{v}^\top \cdot \left[I(\boldsymbol{\beta}^*) + T_n(\widehat{\boldsymbol{\beta}})\right] \cdot \mathbf{v}\right|}{\|\mathbf{v}\|_2^2}$$

$$\geq \underbrace{\inf_{\mathbf{v} \in \mathcal{C}} \frac{\mathbf{v}^\top \cdot I(\boldsymbol{\beta}^*) \cdot \mathbf{v}}{\|\mathbf{v}\|_2^2}}_{\text{(i)}} - \underbrace{\sup_{\mathbf{v} \in \mathcal{C}} \frac{\left|\mathbf{v}^\top \cdot \left[I(\boldsymbol{\beta}^*) + T_n(\widehat{\boldsymbol{\beta}})\right] \cdot \mathbf{v}\right|}{\|\mathbf{v}\|_2^2}}_{\text{(ii)}}, \quad \text{(I.18)}$$

where $\mathcal{C}$ is defined in (I.17).

**Analysis of Term (i):** For term (i) in (I.18), by (4.4) in Assumption 4.5 we have

$$\inf_{\mathbf{v} \in \mathcal{C}} \frac{\mathbf{v}^\top \cdot I(\boldsymbol{\beta}^*) \cdot \mathbf{v}}{\|\mathbf{v}\|_2^2} \geq \inf_{\mathbf{v} \neq \mathbf{0}} \frac{\mathbf{v}^\top \cdot I(\boldsymbol{\beta}^*) \cdot \mathbf{v}}{\|\mathbf{v}\|_2^2} = \lambda_d\left[I(\boldsymbol{\beta}^*)\right] \geq \rho_{\min}. \quad \text{(I.19)}$$

**Analysis of Term (ii):** For term (ii) in (I.18) we have

$$\sup_{\mathbf{v} \in \mathcal{C}} \frac{\left|\mathbf{v}^\top \cdot \left[I(\boldsymbol{\beta}^*) + T_n(\widehat{\boldsymbol{\beta}})\right] \cdot \mathbf{v}\right|}{\|\mathbf{v}\|_2^2} \leq \sup_{\mathbf{v} \in \mathcal{C}} \frac{\|\mathbf{v}\|_1^2 \cdot \left\|I(\boldsymbol{\beta}^*) + T_n(\widehat{\boldsymbol{\beta}})\right\|_{\infty,\infty}}{\|\mathbf{v}\|_2^2}. \quad \text{(I.20)}$$

By the definition of $\mathcal{C}$ in (I.17), for any $\mathbf{v} \in \mathcal{C}$ we have

$$\|\mathbf{v}\|_1 = \left\|\mathbf{v}_{\mathcal{S}_{\mathbf{w}}^*}\right\|_1 + \left\|\mathbf{v}_{\overline{\mathcal{S}_{\mathbf{w}}^*}}\right\|_1 \leq 2 \cdot \left\|\mathbf{v}_{\mathcal{S}_{\mathbf{w}}^*}\right\|_1 \leq 2 \cdot \sqrt{s_{\mathbf{w}}^*} \cdot \left\|\mathbf{v}_{\mathcal{S}_{\mathbf{w}}^*}\right\|_2 \leq 2 \cdot \sqrt{s_{\mathbf{w}}^*} \cdot \|\mathbf{v}\|_2.$$

Therefore, the right-hand side of (I.20) is upper bounded by

$$4 \cdot s_{\mathbf{w}}^* \cdot \left\|I(\boldsymbol{\beta}^*) + T_n(\widehat{\boldsymbol{\beta}})\right\|_{\infty,\infty} \leq \underbrace{4 \cdot s_{\mathbf{w}}^* \cdot \left\|I(\boldsymbol{\beta}^*) + T_n(\boldsymbol{\beta}^*)\right\|_{\infty,\infty}}_{\text{(ii).a}} + \underbrace{4 \cdot s_{\mathbf{w}}^* \cdot \left\|T(\widehat{\boldsymbol{\beta}}) - T_n(\boldsymbol{\beta}^*)\right\|_{\infty,\infty}}_{\text{(ii).b}}.$$

For term (ii).a, by Theorem D.1 and Condition 4.3 we have
$$4 \cdot s_{\mathbf{w}}^* \cdot \left\| I(\boldsymbol{\beta}^*) + T_n(\boldsymbol{\beta}^*) \right\|_{\infty,\infty} = 4 \cdot s_{\mathbf{w}}^* \cdot \left\| T_n(\boldsymbol{\beta}^*) - \mathbb{E}_{\boldsymbol{\beta}^*}\left[T_n(\boldsymbol{\beta}^*)\right] \right\|_{\infty,\infty} = O_{\mathbb{P}}\left(s_{\mathbf{w}}^* \cdot \zeta^{\mathrm{T}}\right) = o_{\mathbb{P}}(1),$$
where the last equality is from (4.6) in Assumption 4.5, since for $\lambda$ specified in (4.5) we have
$$s_{\mathbf{w}}^* \cdot \zeta^{\mathrm{T}} \leq s_{\mathbf{w}}^* \cdot \lambda \leq \max\{\|\mathbf{w}^*\|_1, \ 1\} \cdot s_{\mathbf{w}}^* \cdot \lambda = o(1).$$
For term (ii).b, by Conditions 4.1 and 4.4 we have
$$4 \cdot s_{\mathbf{w}}^* \cdot \left\| T(\widehat{\boldsymbol{\beta}}) - T_n(\boldsymbol{\beta}^*) \right\|_{\infty,\infty} = 4 \cdot s_{\mathbf{w}}^* \cdot O_{\mathbb{P}}(\zeta^{\mathrm{L}}) \cdot \left\| \widehat{\boldsymbol{\beta}} - \boldsymbol{\beta}^* \right\|_1 = O_{\mathbb{P}}\left(s_{\mathbf{w}}^* \cdot \zeta^{\mathrm{L}} \cdot \zeta^{\mathrm{EM}}\right) = o_{\mathbb{P}}(1),$$
where the last equality is also from (4.6) in Assumption 4.5, since for $\lambda$ specified in (4.5) we have
$$s_{\mathbf{w}}^* \cdot \zeta^{\mathrm{L}} \cdot \zeta^{\mathrm{EM}} \leq s_{\mathbf{w}}^* \cdot \lambda \leq \max\{\|\mathbf{w}^*\|_1, \ 1\} \cdot s_{\mathbf{w}}^* \cdot \lambda = o(1).$$
Hence, term (ii) in (I.18) is $o_{\mathbb{P}}(1)$. Since $\rho_{\min}$ is an constant, for a sufficiently large $n$ we have that term (ii) is upper bounded by $\rho_{\min}/2$ with high probability. Further by plugging this and (I.19) into (I.18), we conclude that $\widehat{\rho}_{\min} \geq \rho_{\min}/2$ holds with high probability. $\qquad\square$

The next lemma quantifies the statistical accuracy of $w(\widehat{\boldsymbol{\beta}}, \lambda)$, where $w(\cdot, \cdot)$ is defined in (2.6).

**Lemma I.3.** Under Assumption 4.5 and Conditions 4.1-4.4, for $\lambda$ specified in (4.5) we have that
$$\max\left\{ \left\| w(\widehat{\boldsymbol{\beta}}, \lambda) - \mathbf{w}^* \right\|_1, \ \left\| w(\widehat{\boldsymbol{\beta}}_0, \lambda) - \mathbf{w}^* \right\|_1 \right\} \leq \frac{16 \cdot s_{\mathbf{w}}^* \cdot \lambda}{\rho_{\min}}$$
holds with high probability. Here $\rho_{\min}$ is specified in (4.4), while $\mathbf{w}^*$ and $s_{\mathbf{w}}^*$ are defined (4.1).

*Proof.* For $\lambda$ specified in (4.5), we verify that $\mathbf{w}^*$ is a feasible solution in (2.6) with high probability. For notational simplicity, we define the following event
$$\mathcal{E} = \left\{ \left\| \left[T_n(\widehat{\boldsymbol{\beta}})\right]_{\boldsymbol{\gamma},\alpha} - \left[T_n(\widehat{\boldsymbol{\beta}})\right]_{\boldsymbol{\gamma},\boldsymbol{\gamma}} \cdot \mathbf{w}^* \right\|_{\infty} \leq \lambda \right\}. \tag{I.21}$$
By the definition of $\mathbf{w}^*$ in (4.1), we have $\left[I(\boldsymbol{\beta}^*)\right]_{\boldsymbol{\gamma},\alpha} - \left[I(\boldsymbol{\beta}^*)\right]_{\boldsymbol{\gamma},\boldsymbol{\gamma}} \cdot \mathbf{w}^* = 0$. Hence, we have
$$\begin{aligned}
\left\| \left[T_n(\widehat{\boldsymbol{\beta}})\right]_{\boldsymbol{\gamma},\alpha} - \left[T_n(\widehat{\boldsymbol{\beta}})\right]_{\boldsymbol{\gamma},\boldsymbol{\gamma}} \cdot \mathbf{w}^* \right\|_{\infty} &= \left\| \left[T_n(\widehat{\boldsymbol{\beta}}) + I(\boldsymbol{\beta}^*)\right]_{\boldsymbol{\gamma},\alpha} - \left[T_n(\widehat{\boldsymbol{\beta}}) + I(\boldsymbol{\beta}^*)\right]_{\boldsymbol{\gamma},\boldsymbol{\gamma}} \cdot \mathbf{w}^* \right\|_{\infty} \\
&\leq \left\| T_n(\widehat{\boldsymbol{\beta}}) + I(\boldsymbol{\beta}^*) \right\|_{\infty,\infty} + \left\| T_n(\widehat{\boldsymbol{\beta}}) + I(\boldsymbol{\beta}^*) \right\|_{\infty,\infty} \cdot \|\mathbf{w}^*\|_1,
\end{aligned} \tag{I.22}$$
where the last inequality is from triangle inequality and Hölder's inequality. Note that we have
$$\left\| T_n(\widehat{\boldsymbol{\beta}}) + I(\boldsymbol{\beta}^*) \right\|_{\infty,\infty} \leq \left\| T_n(\boldsymbol{\beta}^*) + I(\boldsymbol{\beta}^*) \right\|_{\infty,\infty} + \left\| T(\widehat{\boldsymbol{\beta}}) - T_n(\boldsymbol{\beta}^*) \right\|_{\infty,\infty}. \tag{I.23}$$
On the right-hand side, by Theorem D.1 and Condition 4.3 we have
$$\left\| T_n(\boldsymbol{\beta}^*) + I(\boldsymbol{\beta}^*) \right\|_{\infty,\infty} = \left\| T_n(\boldsymbol{\beta}^*) - \mathbb{E}_{\boldsymbol{\beta}^*}\left[T_n(\boldsymbol{\beta}^*)\right] \right\|_{\infty,\infty} = O_{\mathbb{P}}(\zeta^{\mathrm{T}}),$$
while by Conditions 4.1 and 4.4 we have
$$\left\| T(\widehat{\boldsymbol{\beta}}) - T_n(\boldsymbol{\beta}^*) \right\|_{\infty,\infty} = O_{\mathbb{P}}(\zeta^{\mathrm{L}}) \cdot \left\| \widehat{\boldsymbol{\beta}} - \boldsymbol{\beta}^* \right\|_1 = O_{\mathbb{P}}(\zeta^{\mathrm{L}} \cdot \zeta^{\mathrm{EM}}).$$
Plugging the above equations into (I.23) and further plugging (I.23) into (I.22), by (4.5) we have
$$\left\| \left[T_n(\widehat{\boldsymbol{\beta}})\right]_{\boldsymbol{\gamma},\alpha} - \left[T_n(\widehat{\boldsymbol{\beta}})\right]_{\boldsymbol{\gamma},\boldsymbol{\gamma}} \cdot \mathbf{w}^* \right\|_{\infty} \leq C \cdot \left(\zeta^{\mathrm{T}} + \zeta^{\mathrm{L}} \cdot \zeta^{\mathrm{EM}}\right) \cdot \left(1 + \|\mathbf{w}^*\|_1\right) = \lambda$$
holds with high probability for a sufficiently large constant $C \geq 1$. In other words, $\mathcal{E}$ occurs with high probability. The subsequent proof will be conditioning on $\mathcal{E}$ and the following event
$$\mathcal{E}' = \left\{ \widehat{\rho}_{\min} \geq \rho_{\min}/2 > 0 \right\}, \tag{I.24}$$
which also occurs with high probability according to Lemma I.2. Here $\widehat{\rho}_{\min}$ is defined in (I.17). For notational simplicity, we denote $w(\widehat{\boldsymbol{\beta}}, \lambda) = \widehat{\mathbf{w}}$. By triangle inequality we have
$$\begin{aligned}
\left\| \left[T_n(\widehat{\boldsymbol{\beta}})\right]_{\boldsymbol{\gamma},\boldsymbol{\gamma}} \cdot (\widehat{\mathbf{w}} - \mathbf{w}^*) \right\|_{\infty} & \\
&\leq \left\| \left[T_n(\widehat{\boldsymbol{\beta}})\right]_{\boldsymbol{\gamma},\alpha} - \left[T_n(\widehat{\boldsymbol{\beta}})\right]_{\boldsymbol{\gamma},\boldsymbol{\gamma}} \cdot \mathbf{w}^* \right\|_{\infty} + \left\| \left[T_n(\widehat{\boldsymbol{\beta}})\right]_{\boldsymbol{\gamma},\boldsymbol{\gamma}} \cdot \widehat{\mathbf{w}} - \left[T_n(\widehat{\boldsymbol{\beta}})\right]_{\boldsymbol{\gamma},\alpha} \right\|_{\infty} \\
&\leq 2 \cdot \lambda,
\end{aligned} \tag{I.25}$$

where the last inequality follows from (2.6) and (I.21). Moreover, by (I.17) and (I.24) we have

$$(\widehat{\mathbf{w}} - \mathbf{w}^*)^\top \cdot \left[-T_n(\widehat{\boldsymbol{\beta}})\right]_{\boldsymbol{\gamma},\boldsymbol{\gamma}} \cdot (\widehat{\mathbf{w}} - \mathbf{w}^*) \geq \widehat{\rho}_{\min} \cdot \|\widehat{\mathbf{w}} - \mathbf{w}^*\|_2^2 \geq \rho_{\min}/2 \cdot \|\widehat{\mathbf{w}} - \mathbf{w}^*\|_2^2. \quad \text{(I.26)}$$

Meanwhile, by Lemma I.1 we have

$$\|\widehat{\mathbf{w}} - \mathbf{w}^*\|_1 = \left\|(\widehat{\mathbf{w}} - \mathbf{w}^*)_{\mathcal{S}_{\mathbf{w}}^*}\right\|_1 + \left\|(\widehat{\mathbf{w}} - \mathbf{w}^*)_{\overline{\mathcal{S}_{\mathbf{w}}^*}}\right\|_1 \leq 2 \cdot \left\|(\widehat{\mathbf{w}} - \mathbf{w}^*)_{\mathcal{S}_{\mathbf{w}}^*}\right\|_1 \leq 2 \cdot \sqrt{s_{\mathbf{w}}^*} \cdot \|\widehat{\mathbf{w}} - \mathbf{w}^*\|_2.$$

Plugging the above inequality into (I.26), we obtain

$$(\widehat{\mathbf{w}} - \mathbf{w}^*)^\top \cdot \left[-T_n(\widehat{\boldsymbol{\beta}})\right]_{\boldsymbol{\gamma},\boldsymbol{\gamma}} \cdot (\widehat{\mathbf{w}} - \mathbf{w}^*) \geq \rho_{\min}/(8 \cdot s_{\mathbf{w}}^*) \cdot \|\widehat{\mathbf{w}} - \mathbf{w}^*\|_1^2. \quad \text{(I.27)}$$

Note that by (I.25), the left-hand side of (I.27) is upper bounded by

$$\|\widehat{\mathbf{w}} - \mathbf{w}^*\|_1 \cdot \left\|\left[T_n(\widehat{\boldsymbol{\beta}})\right]_{\boldsymbol{\gamma},\boldsymbol{\gamma}} \cdot (\widehat{\mathbf{w}} - \mathbf{w}^*)\right\|_\infty \leq \|\widehat{\mathbf{w}} - \mathbf{w}^*\|_1 \cdot 2 \cdot \lambda. \quad \text{(I.28)}$$

By (I.27) and (I.28), we then obtain $\|\widehat{\mathbf{w}} - \mathbf{w}^*\|_1 \leq 16 \cdot s_{\mathbf{w}}^* \cdot \lambda/\rho_{\min}$ conditioning on $\mathcal{E}$ and $\mathcal{E}'$, both of which hold with high probability. Note that the proof for $w(\widehat{\boldsymbol{\beta}}_0, \lambda)$ follows similarly. Therefore, we conclude the proof of Lemma I.3. $\qquad\square$

### I.3 Proof of Lemma G.3

*Proof.* Our proof strategy is as follows. First we prove that

$$\sqrt{n} \cdot S_n(\widehat{\boldsymbol{\beta}}_0, \lambda) = \sqrt{n} \cdot \left[\nabla_1 Q_n(\boldsymbol{\beta}^*; \boldsymbol{\beta}^*)\right]_\alpha - \sqrt{n} \cdot (\mathbf{w}^*)^\top \cdot \left[\nabla_1 Q_n(\boldsymbol{\beta}^*; \boldsymbol{\beta}^*)\right]_{\boldsymbol{\gamma}} + o_{\mathbb{P}}(1), \quad \text{(I.29)}$$

where $\boldsymbol{\beta}^*$ is the true parameter and $\mathbf{w}^*$ is defined in (4.1). We then prove

$$\sqrt{n} \cdot \left[\nabla_1 Q_n(\boldsymbol{\beta}^*; \boldsymbol{\beta}^*)\right]_\alpha - \sqrt{n} \cdot (\mathbf{w}^*)^\top \cdot \left[\nabla_1 Q_n(\boldsymbol{\beta}^*; \boldsymbol{\beta}^*)\right]_{\boldsymbol{\gamma}} \xrightarrow{D} N\left(0, \left[I(\boldsymbol{\beta}^*)\right]_{\alpha|\boldsymbol{\gamma}}\right), \quad \text{(I.30)}$$

where $\left[I(\boldsymbol{\beta}^*)\right]_{\alpha|\boldsymbol{\gamma}}$ is defined in (4.2). Throughout the proof, we abbreviate $w(\widehat{\boldsymbol{\beta}}_0, \lambda)$ as $\widehat{\mathbf{w}}_0$. Also, it is worth noting that our analysis is under the null hypothesis where $\boldsymbol{\beta}^* = \left[\alpha^*, (\boldsymbol{\gamma}^*)^\top\right]^\top$ with $\alpha^* = 0$.

**Proof of (I.29):** For (I.29), by the definition of the decorrelated score function in (2.5) we have

$$S_n(\widehat{\boldsymbol{\beta}}_0, \lambda) = \left[\nabla_1 Q_n(\widehat{\boldsymbol{\beta}}_0; \widehat{\boldsymbol{\beta}}_0)\right]_\alpha - \widehat{\mathbf{w}}_0^\top \cdot \left[\nabla_1 Q_n(\widehat{\boldsymbol{\beta}}_0; \widehat{\boldsymbol{\beta}}_0)\right]_{\boldsymbol{\gamma}}.$$

By the mean-value theorem, we obtain

$$S_n(\widehat{\boldsymbol{\beta}}_0, \lambda) = \overbrace{\left[\nabla_1 Q_n(\boldsymbol{\beta}^*; \boldsymbol{\beta}^*)\right]_\alpha - \widehat{\mathbf{w}}_0^\top \cdot \left[\nabla_1 Q_n(\boldsymbol{\beta}^*; \boldsymbol{\beta}^*)\right]_{\boldsymbol{\gamma}}}^{(i)} \quad \text{(I.31)}$$

$$+ \underbrace{\left[T_n(\boldsymbol{\beta}^\sharp)\right]_{\boldsymbol{\gamma},\alpha}^\top \cdot (\widehat{\boldsymbol{\beta}}_0 - \boldsymbol{\beta}^*) - \widehat{\mathbf{w}}_0^\top \cdot \left[T_n(\boldsymbol{\beta}^\sharp)\right]_{\boldsymbol{\gamma},\boldsymbol{\gamma}} \cdot (\widehat{\boldsymbol{\beta}}_0 - \boldsymbol{\beta}^*)}_{(ii)},$$

where we have $T_n(\boldsymbol{\beta}) = \nabla_{1,1}^2 Q_n(\boldsymbol{\beta}; \boldsymbol{\beta}) + \nabla_{1,2}^2 Q_n(\boldsymbol{\beta}; \boldsymbol{\beta})$ as defined in (2.4), and $\boldsymbol{\beta}^\sharp$ is an intermediate value between $\boldsymbol{\beta}^*$ and $\widehat{\boldsymbol{\beta}}_0$.

**Analysis of Term (i):** For term (i) in (I.31), we have

$$\left[\nabla_1 Q_n(\boldsymbol{\beta}^*; \boldsymbol{\beta}^*)\right]_\alpha - \widehat{\mathbf{w}}_0^\top \cdot \left[\nabla_1 Q_n(\boldsymbol{\beta}^*; \boldsymbol{\beta}^*)\right]_{\boldsymbol{\gamma}}$$

$$= \left[\nabla_1 Q_n(\boldsymbol{\beta}^*; \boldsymbol{\beta}^*)\right]_\alpha - (\mathbf{w}^*)^\top \cdot \left[\nabla_1 Q_n(\boldsymbol{\beta}^*; \boldsymbol{\beta}^*)\right]_{\boldsymbol{\gamma}} + (\mathbf{w}^* - \widehat{\mathbf{w}}_0)^\top \cdot \left[\nabla_1 Q_n(\boldsymbol{\beta}^*; \boldsymbol{\beta}^*)\right]_{\boldsymbol{\gamma}}. \quad \text{(I.32)}$$

For the right-hand side of (I.32), we have

$$(\mathbf{w}^* - \widehat{\mathbf{w}}_0)^\top \cdot \left[\nabla_1 Q_n(\boldsymbol{\beta}^*; \boldsymbol{\beta}^*)\right]_{\boldsymbol{\gamma}} \leq \|\mathbf{w}^* - \widehat{\mathbf{w}}_0\|_1 \cdot \left\|\left[\nabla_1 Q_n(\boldsymbol{\beta}^*; \boldsymbol{\beta}^*)\right]_{\boldsymbol{\gamma}}\right\|_\infty. \quad \text{(I.33)}$$

By Lemma I.3, we have $\|\mathbf{w}^* - \widehat{\mathbf{w}}_0\|_1 = O_{\mathbb{P}}(s_{\mathbf{w}}^* \cdot \lambda)$, where $\lambda$ is specified in (4.5). Meanwhile, we have

$$\left\|\left[\nabla_1 Q_n(\boldsymbol{\beta}^*; \boldsymbol{\beta}^*)\right]_{\boldsymbol{\gamma}}\right\|_\infty \leq \left\|\nabla_1 Q_n(\boldsymbol{\beta}^*; \boldsymbol{\beta}^*)\right\|_\infty = \left\|\nabla_1 Q_n(\boldsymbol{\beta}^*; \boldsymbol{\beta}^*) - \nabla_1 Q(\boldsymbol{\beta}^*; \boldsymbol{\beta}^*)\right\|_\infty$$

$$= O_{\mathbb{P}}(\zeta^G),$$

where the first equality follows from the self-consistency property [18] that $\boldsymbol{\beta}^* = \operatorname{argmax}_{\boldsymbol{\beta}} Q(\boldsymbol{\beta}; \boldsymbol{\beta}^*)$, which gives $\nabla_1 Q(\boldsymbol{\beta}^*; \boldsymbol{\beta}^*) = \mathbf{0}$. Here the last equality is from Condition 4.2. Therefore, (I.33) implies

$$(\mathbf{w}^* - \widehat{\mathbf{w}}_0)^\top \cdot \left[\nabla_1 Q_n(\boldsymbol{\beta}^*; \boldsymbol{\beta}^*)\right]_{\boldsymbol{\gamma}} = O_{\mathbb{P}}(s_{\mathbf{w}}^* \cdot \lambda \cdot \zeta^G) = o_{\mathbb{P}}(1/\sqrt{n}),$$

where the second equality is from $s_{\mathbf{w}}^* \cdot \lambda \cdot \zeta^{\mathrm{G}} = o(1/\sqrt{n})$ in (4.6) of Assumption 4.5. Thus, by (I.32) we conclude that term (i) in (I.31) equals

$$\left[\nabla_1 Q_n(\boldsymbol{\beta}^*; \boldsymbol{\beta}^*)\right]_{\alpha} - (\mathbf{w}^*)^{\top} \cdot \left[\nabla_1 Q_n(\boldsymbol{\beta}^*; \boldsymbol{\beta}^*)\right]_{\gamma} + o_{\mathbb{P}}(1/\sqrt{n}).$$

**Analysis of Term (ii):** By triangle inequality, term (ii) in (I.31) is upper bounded by

$$\underbrace{\left| \left[T_n(\widehat{\boldsymbol{\beta}}_0)\right]_{\gamma,\alpha}^{\top} \cdot (\widehat{\boldsymbol{\beta}}_0 - \boldsymbol{\beta}^*) - \widehat{\mathbf{w}}_0^{\top} \cdot \left[T_n(\widehat{\boldsymbol{\beta}}_0)\right]_{\gamma,\gamma} \cdot (\widehat{\boldsymbol{\beta}}_0 - \boldsymbol{\beta}^*) \right|}_{\text{(ii).a}} \tag{I.34}$$

$$+ \underbrace{\left| \left[T_n(\boldsymbol{\beta}^\sharp)\right]_{\gamma,\alpha}^{\top} \cdot (\widehat{\boldsymbol{\beta}}_0 - \boldsymbol{\beta}^*) - \left[T_n(\widehat{\boldsymbol{\beta}}_0)\right]_{\gamma,\alpha}^{\top} \cdot (\widehat{\boldsymbol{\beta}}_0 - \boldsymbol{\beta}^*) \right|}_{\text{(ii).b}}$$

$$+ \underbrace{\left| \widehat{\mathbf{w}}_0^{\top} \cdot \left[T_n(\widehat{\boldsymbol{\beta}}_0)\right]_{\gamma,\gamma} \cdot (\widehat{\boldsymbol{\beta}}_0 - \boldsymbol{\beta}^*) - \widehat{\mathbf{w}}_0^{\top} \cdot \left[T_n(\boldsymbol{\beta}^\sharp)\right]_{\gamma,\gamma} \cdot (\widehat{\boldsymbol{\beta}}_0 - \boldsymbol{\beta}^*) \right|}_{\text{(ii).c}}.$$

By Hölder's inequality, term (ii).a in (I.34) is upper bounded by

$$\left\|\widehat{\boldsymbol{\beta}}_0 - \boldsymbol{\beta}^*\right\|_1 \cdot \left\| \left[T_n(\widehat{\boldsymbol{\beta}}_0)\right]_{\gamma,\alpha} - \widehat{\mathbf{w}}_0^{\top} \cdot \left[T_n(\widehat{\boldsymbol{\beta}}_0)\right]_{\gamma,\gamma} \right\|_{\infty} = \left\|\widehat{\boldsymbol{\beta}}_0 - \boldsymbol{\beta}^*\right\|_1 \cdot \lambda$$

$$\leq O_{\mathbb{P}}(\zeta^{\mathrm{EM}}) \cdot \lambda = o_{\mathbb{P}}(1/\sqrt{n}), \tag{I.35}$$

where the first inequality holds because $\widehat{\mathbf{w}}_0 = w(\widehat{\boldsymbol{\beta}}_0, \lambda)$ is a feasible solution in (2.6). Meanwhile, Condition 4.1 gives $\left\|\widehat{\boldsymbol{\beta}} - \boldsymbol{\beta}^*\right\|_1 = O_{\mathbb{P}}(\zeta^{\mathrm{EM}})$. Also note that by definition we have $(\widehat{\boldsymbol{\beta}}_0)_\alpha = (\boldsymbol{\beta}^*)_\alpha = 0$, which implies $\left\|\widehat{\boldsymbol{\beta}}_0 - \boldsymbol{\beta}^*\right\|_1 \leq \left\|\widehat{\boldsymbol{\beta}} - \boldsymbol{\beta}^*\right\|_1$. Hence, we have

$$\left\|\widehat{\boldsymbol{\beta}}_0 - \boldsymbol{\beta}^*\right\|_1 = O_{\mathbb{P}}(\zeta^{\mathrm{EM}}), \tag{I.36}$$

which implies the first equality in (I.35). The last equality in (I.35) follows from $\zeta^{\mathrm{EM}} \cdot \lambda = o(1/\sqrt{n})$ in (4.6) of Assumption 4.5. Note that term (ii).b in (I.34) is upper bounded by

$$\left\| \left[T_n(\boldsymbol{\beta}^\sharp)\right]_{\gamma,\alpha} - \left[T_n(\widehat{\boldsymbol{\beta}}_0)\right]_{\gamma,\alpha} \right\|_{\infty} \cdot \left\|\widehat{\boldsymbol{\beta}}_0 - \boldsymbol{\beta}^*\right\|_1 \leq \left\| T_n(\boldsymbol{\beta}^\sharp) - T_n(\widehat{\boldsymbol{\beta}}_0) \right\|_{\infty,\infty} \cdot \left\|\widehat{\boldsymbol{\beta}}_0 - \boldsymbol{\beta}^*\right\|_1. \tag{I.37}$$

For the first term on the right-hand side of (I.37), by triangle inequality we have

$$\left\| T_n(\boldsymbol{\beta}^\sharp) - T_n(\widehat{\boldsymbol{\beta}}_0) \right\|_{\infty,\infty} \leq \left\| T_n(\boldsymbol{\beta}^\sharp) - T_n(\boldsymbol{\beta}^*) \right\|_{\infty,\infty} + \left\| T_n(\widehat{\boldsymbol{\beta}}_0) - T_n(\boldsymbol{\beta}^*) \right\|_{\infty,\infty}.$$

By Condition 4.4, we have

$$\left\| T_n(\widehat{\boldsymbol{\beta}}_0) - T_n(\boldsymbol{\beta}^*) \right\|_{\infty,\infty} = O_{\mathbb{P}}(\zeta^{\mathrm{L}}) \cdot \left\|\widehat{\boldsymbol{\beta}}_0 - \boldsymbol{\beta}^*\right\|_1, \tag{I.38}$$

and

$$\left\| T_n(\boldsymbol{\beta}^\sharp) - T_n(\boldsymbol{\beta}^*) \right\|_{\infty,\infty} = O_{\mathbb{P}}(\zeta^{\mathrm{L}}) \cdot \left\|\boldsymbol{\beta}^\sharp - \boldsymbol{\beta}^*\right\|_1 \leq O_{\mathbb{P}}(\zeta^{\mathrm{L}}) \cdot \left\|\widehat{\boldsymbol{\beta}}_0 - \boldsymbol{\beta}^*\right\|_1, \tag{I.39}$$

where the last inequality in (I.39) holds because $\boldsymbol{\beta}^\sharp$ is defined as an intermediate value between $\boldsymbol{\beta}^*$ and $\widehat{\boldsymbol{\beta}}_0$. Further by plugging (I.36) into (I.38), (I.39) as well as the second term on the right-hand side of (I.37), we have that term (ii).b in (I.34) is $O_{\mathbb{P}}\left[\zeta^{\mathrm{L}} \cdot (\zeta^{\mathrm{EM}})^2\right]$. Moreover, by our assumption in (4.6) of Assumption 4.5 we have

$$\zeta^{\mathrm{L}} \cdot (\zeta^{\mathrm{EM}})^2 \leq \max\{1, \|\mathbf{w}^*\|_1\} \cdot \zeta^{\mathrm{L}} \cdot (\zeta^{\mathrm{EM}})^2 = o(1/\sqrt{n}).$$

Thus, we conclude that term (ii).b is $o_{\mathbb{P}}(1/\sqrt{n})$. Similarly, term (ii).c in (I.34) is upper bounded by

$$\|\widehat{\mathbf{w}}_0\|_1 \cdot \left\| T_n(\boldsymbol{\beta}^\sharp) - T_n(\widehat{\boldsymbol{\beta}}_0) \right\|_{\infty,\infty} \cdot \left\|\widehat{\boldsymbol{\beta}}_0 - \boldsymbol{\beta}^*\right\|_1. \tag{I.40}$$

By triangle inequality and Lemma I.3, the first term in (I.40) is upper bounded by

$$\|\mathbf{w}^*\|_1 + \|\widehat{\mathbf{w}}_0 - \mathbf{w}^*\|_1 = \|\mathbf{w}^*\|_1 + O_{\mathbb{P}}(s_{\mathbf{w}}^* \cdot \lambda).$$

Meanwhile, for the second and third terms in (I.40), by the same analysis for term (ii).b in (I.34) we have

$$\left\| T_n(\boldsymbol{\beta}^\sharp) - T_n(\widehat{\boldsymbol{\beta}}_0) \right\|_{\infty,\infty} \cdot \left\|\widehat{\boldsymbol{\beta}}_0 - \boldsymbol{\beta}^*\right\|_1 = O_{\mathbb{P}}\left[\zeta^{\mathrm{L}} \cdot (\zeta^{\mathrm{EM}})^2\right].$$

By (4.6) in Assumption 4.5, since $s_{\mathbf{w}}^* \cdot \lambda = o(1)$, we have

$$\left(\|\mathbf{w}^*\|_1 + s_{\mathbf{w}}^* \cdot \lambda\right) \cdot \zeta^{\mathrm{L}} \cdot \left(\zeta^{\mathrm{EM}}\right)^2 \leq \left[\max\{1, \ \|\mathbf{w}^*\|_1\} + o(1)\right] \cdot \zeta^{\mathrm{L}} \cdot \left(\zeta^{\mathrm{EM}}\right)^2$$
$$= o(1/\sqrt{n}).$$

Therefore, term (ii).c in (I.34) is $o_{\mathbb{P}}(1/\sqrt{n})$. Hence, by (I.34) we conclude that term (ii) in (I.31) is $o_{\mathbb{P}}(1/\sqrt{n})$. Combining the analysis for terms (i) and (ii) in (I.31), we then obtain (I.29). In the sequel, we turn to prove the second part on asymptotic normality.

**Proof of (I.30):** Note that by Theorem D.1, we have

$$\sqrt{n} \cdot \left[\nabla_1 Q_n(\boldsymbol{\beta}^*; \boldsymbol{\beta}^*)\right]_\alpha - \sqrt{n} \cdot (\mathbf{w}^*)^\top \cdot \left[\nabla_1 Q_n(\boldsymbol{\beta}^*; \boldsymbol{\beta}^*)\right]_\gamma = \sqrt{n} \cdot \left[1, -(\mathbf{w}^*)^\top\right] \cdot \nabla_1 Q_n(\boldsymbol{\beta}^*; \boldsymbol{\beta}^*)$$
$$= \sqrt{n} \cdot \left[1, -(\mathbf{w}^*)^\top\right] \cdot \nabla \ell_n(\boldsymbol{\beta}^*)/n.$$
$$(\text{I.41})$$

Recall that $\ell_n(\boldsymbol{\beta}^*)$ is the log-likelihood function defined in (B.3). Hence, $\left[1, -(\mathbf{w}^*)^\top\right] \cdot \nabla \ell_n(\boldsymbol{\beta}^*)/n$ is the average of $n$ independent random variables. Meanwhile, the score function has mean zero at $\boldsymbol{\beta}^*$, i.e., $\mathbb{E}\left[\nabla \ell_n(\boldsymbol{\beta}^*)\right] = \mathbf{0}$. For the variance of the rescaled average in (I.41), we have

$$\mathrm{Var}\left\{\sqrt{n} \cdot \left[1, -(\mathbf{w}^*)^\top\right] \cdot \nabla \ell_n(\boldsymbol{\beta}^*)/n\right\} = \left[1, -(\mathbf{w}^*)^\top\right] \cdot \mathrm{Cov}\left[\nabla \ell_n(\boldsymbol{\beta}^*)/\sqrt{n}\right] \cdot \left[1, -(\mathbf{w}^*)^\top\right]^\top$$
$$= \left[1, -(\mathbf{w}^*)^\top\right] \cdot I(\boldsymbol{\beta}^*) \cdot \left[1, -(\mathbf{w}^*)^\top\right]^\top.$$

Here the second equality is from the fact that the covariance of the score function equals the Fisher information (up to renormalization). Hence, the variance of each item in the average in (I.41) is

$$\left[1, -(\mathbf{w}^*)^\top\right] \cdot I(\boldsymbol{\beta}^*) \cdot \left[1, -(\mathbf{w}^*)^\top\right]^\top = \left[I(\boldsymbol{\beta}^*)\right]_{\alpha,\alpha} - 2 \cdot (\mathbf{w}^*)^\top \cdot \left[I(\boldsymbol{\beta}^*)\right]_{\gamma,\alpha} + (\mathbf{w}^*)^\top \cdot \left[I(\boldsymbol{\beta}^*)\right]_{\gamma,\gamma} \cdot \mathbf{w}^*$$
$$= \left[I(\boldsymbol{\beta}^*)\right]_{\alpha,\alpha} - \left[I(\boldsymbol{\beta}^*)\right]_{\gamma,\alpha} \cdot \left[I(\boldsymbol{\beta}^*)\right]_{\gamma,\gamma}^{-1} \cdot \left[I(\boldsymbol{\beta}^*)\right]_{\gamma,\alpha}$$
$$= \left[I(\boldsymbol{\beta}^*)\right]_{\alpha|\gamma},$$

where the second and third equalities are from (4.1) and (4.2). Hence, by the central limit theorem we obtain (I.30). Finally, combining (I.29) and (I.30) by invoking Slutsky's theorem, we obtain

$$\sqrt{n} \cdot S_n(\widehat{\boldsymbol{\beta}}_0, \lambda) \xrightarrow{D} N\left(0, \left[I(\boldsymbol{\beta}^*)\right]_{\alpha|\gamma}\right),$$

which concludes the proof of Lemma G.3. $\qquad\square$

## I.4   Proof of Lemma G.4

*Proof.* Throughout the proof, we abbreviate $w(\widehat{\boldsymbol{\beta}}_0, \lambda)$ as $\widehat{\mathbf{w}}_0$. Our proof is under the null hypothesis where $\boldsymbol{\beta}^* = \left[\alpha^*, (\boldsymbol{\gamma}^*)^\top\right]^\top$ with $\alpha^* = 0$. Recall that $\mathbf{w}^*$ is defined in (4.1). Then by the definitions of $\left[T_n(\widehat{\boldsymbol{\beta}}_0)\right]_{\alpha|\gamma}$ and $\left[I(\boldsymbol{\beta}^*)\right]_{\alpha|\gamma}$ in (2.7) and (4.2), we have

$$\left[T_n(\widehat{\boldsymbol{\beta}}_0)\right]_{\alpha|\gamma} = \left(1, -\widehat{\mathbf{w}}_0^\top\right) \cdot T_n(\widehat{\boldsymbol{\beta}}_0) \cdot \left(1, -\widehat{\mathbf{w}}_0^\top\right)^\top$$
$$= \left[T_n(\widehat{\boldsymbol{\beta}}_0)\right]_{\alpha,\alpha} - 2 \cdot \widehat{\mathbf{w}}_0^\top \cdot \left[T_n(\widehat{\boldsymbol{\beta}}_0)\right]_{\gamma,\alpha} + \widehat{\mathbf{w}}_0^\top \cdot \left[T_n(\widehat{\boldsymbol{\beta}}_0)\right]_{\gamma,\gamma} \cdot \widehat{\mathbf{w}}_0,$$
$$\left[I(\boldsymbol{\beta}^*)\right]_{\alpha|\gamma} = \left[I(\boldsymbol{\beta}^*)\right]_{\alpha,\alpha} - \left[I(\boldsymbol{\beta}^*)\right]_{\gamma,\alpha}^\top \cdot \left[I(\boldsymbol{\beta}^*)\right]_{\gamma,\gamma}^{-1} \cdot \left[I(\boldsymbol{\beta}^*)\right]_{\gamma,\alpha}$$
$$= \left[I(\boldsymbol{\beta}^*)\right]_{\alpha,\alpha} - 2 \cdot (\mathbf{w}^*)^\top \cdot \left[I(\boldsymbol{\beta}^*)\right]_{\gamma,\alpha} + (\mathbf{w}^*)^\top \cdot \left[I(\boldsymbol{\beta}^*)\right]_{\gamma,\gamma} \cdot \mathbf{w}^*.$$

By triangle inequality, we have

$$\left|\left[T_n(\widehat{\boldsymbol{\beta}}_0)\right]_{\alpha|\gamma} + \left[I(\boldsymbol{\beta}^*)\right]_{\alpha|\gamma}\right|$$
$$\leq \underbrace{\left|\left[T_n(\widehat{\boldsymbol{\beta}}_0)\right]_{\alpha,\alpha} + \left[I(\boldsymbol{\beta}^*)\right]_{\alpha,\alpha}\right|}_{(i)} + 2 \cdot \underbrace{\left|\widehat{\mathbf{w}}_0^\top \cdot \left[T_n(\widehat{\boldsymbol{\beta}}_0)\right]_{\gamma,\alpha} + (\mathbf{w}^*)^\top \cdot \left[I(\boldsymbol{\beta}^*)\right]_{\gamma,\alpha}\right|}_{(ii)}$$
$$+ \underbrace{\left|\widehat{\mathbf{w}}_0^\top \cdot \left[T_n(\widehat{\boldsymbol{\beta}}_0)\right]_{\gamma,\gamma} \cdot \widehat{\mathbf{w}}_0 + (\mathbf{w}^*)^\top \cdot \left[I(\boldsymbol{\beta}^*)\right]_{\gamma,\gamma} \cdot \mathbf{w}^*\right|}_{(iii)}. \qquad (\text{I.42})$$

**Analysis of Term (i):** For term (i) in (I.42), by Theorem D.1 and triangle inequality we have

$$\left| \left[ T_n(\widehat{\boldsymbol{\beta}}_0) \right]_{\alpha,\alpha} + \left[ I(\boldsymbol{\beta}^*) \right]_{\alpha,\alpha} \right| \leq \underbrace{\left| \left[ T_n(\widehat{\boldsymbol{\beta}}_0) \right]_{\alpha,\alpha} - \left[ T_n(\boldsymbol{\beta}^*) \right]_{\alpha,\alpha} \right|}_{\text{(i).a}} + \underbrace{\left| \left[ T_n(\boldsymbol{\beta}^*) \right]_{\alpha,\alpha} - \left\{ \mathbb{E}_{\boldsymbol{\beta}^*} \left[ T_n(\boldsymbol{\beta}^*) \right] \right\}_{\alpha,\alpha} \right|}_{\text{(i).b}}.$$

(I.43)

For term (i).a in (I.43), by Condition 4.4 we have

$$\left| \left[ T_n(\widehat{\boldsymbol{\beta}}_0) \right]_{\alpha,\alpha} - \left[ T_n(\boldsymbol{\beta}^*) \right]_{\alpha,\alpha} \right| \leq \left\| T_n(\widehat{\boldsymbol{\beta}}_0) - T_n(\boldsymbol{\beta}^*) \right\|_{\infty,\infty}$$

$$= O_{\mathbb{P}}(\zeta^{\mathrm{L}}) \cdot \left\| \widehat{\boldsymbol{\beta}}_0 - \boldsymbol{\beta}^* \right\|_1.$$

(I.44)

Note that we have $(\widehat{\boldsymbol{\beta}}_0)_\alpha = (\boldsymbol{\beta}^*)_\alpha = 0$ by definition, which implies $\left\| \widehat{\boldsymbol{\beta}}_0 - \boldsymbol{\beta}^* \right\|_1 \leq \left\| \widehat{\boldsymbol{\beta}} - \boldsymbol{\beta}^* \right\|_1$. Hence, by Condition 4.1 we have

$$\left\| \widehat{\boldsymbol{\beta}}_0 - \boldsymbol{\beta}^* \right\|_1 = O_{\mathbb{P}}(\zeta^{\mathrm{EM}}).$$

(I.45)

Moreover, combining (I.44) and (I.45), by (4.6) in Assumption 4.5 we have

$$\zeta^{\mathrm{L}} \cdot \zeta^{\mathrm{EM}} \leq \max \left\{ \|\mathbf{w}^*\|_1, \ 1 \right\} \cdot s_{\mathbf{w}}^* \cdot \lambda = o(1)$$

for $\lambda$ specified in (4.5). Hence we obtain

$$\left| \left[ T_n(\widehat{\boldsymbol{\beta}}_0) \right]_{\alpha,\alpha} - \left[ T_n(\boldsymbol{\beta}^*) \right]_{\alpha,\alpha} \right| \leq \left\| T_n(\widehat{\boldsymbol{\beta}}_0) - T_n(\boldsymbol{\beta}^*) \right\|_{\infty,\infty}$$

$$= O_{\mathbb{P}}(\zeta^{\mathrm{L}} \cdot \zeta^{\mathrm{EM}}) = o_{\mathbb{P}}(1).$$

(I.46)

Meanwhile, for term (i).b in (I.43) we have

$$\left| \left[ T_n(\boldsymbol{\beta}^*) \right]_{\alpha,\alpha} - \left\{ \mathbb{E}_{\boldsymbol{\beta}^*} \left[ T_n(\boldsymbol{\beta}^*) \right] \right\}_{\alpha,\alpha} \right| \leq \left\| T_n(\boldsymbol{\beta}^*) - \mathbb{E}_{\boldsymbol{\beta}^*} \left[ T_n(\boldsymbol{\beta}^*) \right] \right\|_{\infty,\infty} = O_{\mathbb{P}}(\zeta^{\mathrm{T}}) = o_{\mathbb{P}}(1),$$

(I.47)

where the second last equality follows from Condition 4.3, while the last equality holds because our assumption in (4.6) of Assumption 4.5 implies

$$\zeta^{\mathrm{T}} \leq \max \left\{ \|\mathbf{w}^*\|_1, \ 1 \right\} \cdot s_{\mathbf{w}}^* \cdot \lambda = o(1)$$

for $\lambda$ specified in (4.5).

**Analysis of Term (ii):** For term (ii) in (I.42), by Theorem D.1 and triangle inequality, we have

$$\left| \widehat{\mathbf{w}}_0^\top \cdot \left[ T_n(\widehat{\boldsymbol{\beta}}_0) \right]_{\boldsymbol{\gamma},\alpha} + (\mathbf{w}^*)^\top \cdot \left[ I(\boldsymbol{\beta}^*) \right]_{\boldsymbol{\gamma},\alpha} \right|$$

(I.48)

$$\leq \underbrace{\left| (\widehat{\mathbf{w}}_0 - \mathbf{w}^*)^\top \cdot \left\{ T_n(\widehat{\boldsymbol{\beta}}_0) - \mathbb{E}_{\boldsymbol{\beta}^*} \left[ T_n(\boldsymbol{\beta}^*) \right] \right\}_{\boldsymbol{\gamma},\alpha} \right|}_{\text{(ii).a}} + \underbrace{\left| (\widehat{\mathbf{w}}_0 - \mathbf{w}^*)^\top \cdot \left\{ \mathbb{E}_{\boldsymbol{\beta}^*} \left[ T_n(\boldsymbol{\beta}^*) \right] \right\}_{\boldsymbol{\gamma},\alpha} \right|}_{\text{(ii).b}}$$

$$+ \underbrace{\left| (\mathbf{w}^*)^\top \cdot \left\{ T_n(\widehat{\boldsymbol{\beta}}_0) - \mathbb{E}_{\boldsymbol{\beta}^*} \left[ T_n(\boldsymbol{\beta}^*) \right] \right\}_{\boldsymbol{\gamma},\alpha} \right|}_{\text{(ii).c}}.$$

By Hölder's inequality, term (ii).a in (I.48) is upper bounded by

$$\|\widehat{\mathbf{w}}_0 - \mathbf{w}^*\|_1 \cdot \left\| \left\{ T_n(\widehat{\boldsymbol{\beta}}_0) - \mathbb{E}_{\boldsymbol{\beta}^*} \left[ T_n(\boldsymbol{\beta}^*) \right] \right\}_{\boldsymbol{\gamma},\alpha} \right\|_\infty \leq \|\widehat{\mathbf{w}}_0 - \mathbf{w}^*\|_1 \cdot \left\| T_n(\widehat{\boldsymbol{\beta}}_0) - \mathbb{E}_{\boldsymbol{\beta}^*} \left[ T_n(\boldsymbol{\beta}^*) \right] \right\|_{\infty,\infty}.$$

By Lemma I.3, we have $\|\widehat{\mathbf{w}}_0 - \mathbf{w}^*\|_1 = O_{\mathbb{P}}(s_{\mathbf{w}}^* \cdot \lambda)$. Meanwhile, we have

$$\left\| T_n(\widehat{\boldsymbol{\beta}}_0) - \mathbb{E}_{\boldsymbol{\beta}^*} \left[ T_n(\boldsymbol{\beta}^*) \right] \right\|_{\infty,\infty} \leq \left\| T_n(\widehat{\boldsymbol{\beta}}_0) - T_n(\boldsymbol{\beta}^*) \right\|_{\infty,\infty} + \left\| T_n(\boldsymbol{\beta}^*) - \mathbb{E}_{\boldsymbol{\beta}^*} \left[ T_n(\boldsymbol{\beta}^*) \right] \right\|_{\infty,\infty} = o_{\mathbb{P}}(1).$$

where the second equality follows from (I.46) and (I.47). Therefore, term (ii).a is $o_{\mathbb{P}}(1)$, since (4.6) in Assumption 4.5 implies $s_{\mathbf{w}}^* \cdot \lambda = o(1)$. Meanwhile, by Hölder's inequality, term (ii).b in (I.48) is upper bounded by

$$\|\widehat{\mathbf{w}}_0 - \mathbf{w}^*\|_1 \cdot \left\| \left\{ \mathbb{E}_{\boldsymbol{\beta}^*} \left[ T_n(\boldsymbol{\beta}^*) \right] \right\}_{\boldsymbol{\gamma},\alpha} \right\|_\infty \leq \|\widehat{\mathbf{w}}_0 - \mathbf{w}^*\|_1 \cdot \left\| \mathbb{E}_{\boldsymbol{\beta}^*} \left[ T_n(\boldsymbol{\beta}^*) \right] \right\|_{\infty,\infty}.$$

(I.49)

By Lemma I.3, we have $\|\widehat{\mathbf{w}}_0 - \mathbf{w}^*\|_1 = O_{\mathbb{P}}(s_{\mathbf{w}}^* \cdot \lambda)$. Meanwhile, we have $\mathbb{E}_{\boldsymbol{\beta}^*} \left[ T_n(\boldsymbol{\beta}^*) \right] = -I(\boldsymbol{\beta}^*)$ by Theorem D.1. Furthermore, (4.4) in Assumption 4.5 implies

$$\left\| I(\boldsymbol{\beta}^*) \right\|_{\infty,\infty} \leq \left\| I(\boldsymbol{\beta}^*) \right\|_2 \leq C,$$

(I.50)

where $C > 0$ is some constant. Therefore, from (I.49) we have that term (ii).b in (I.48) is $O_{\mathbb{P}}(s_{\mathbf{w}}^* \cdot \lambda)$. By (4.6) in Assumption 4.5, we have $s_{\mathbf{w}}^* \cdot \lambda = o(1)$. Thus, we conclude that term (ii).b is $o_{\mathbb{P}}(1)$. For

term (ii).c, we have

$$\left| (\mathbf{w}^*)^\top \cdot \left\{ T_n(\widehat{\boldsymbol{\beta}}_0) - \mathbb{E}_{\boldsymbol{\beta}^*}[T_n(\boldsymbol{\beta}^*)] \right\}_{\boldsymbol{\gamma},\alpha} \right|$$

$$\leq \|\mathbf{w}^*\|_1 \cdot \left\| T_n(\widehat{\boldsymbol{\beta}}_0) - \mathbb{E}_{\boldsymbol{\beta}^*}[T_n(\boldsymbol{\beta}^*)] \right\|_{\infty,\infty}$$

$$\leq \|\mathbf{w}^*\|_1 \cdot \left\| T_n(\widehat{\boldsymbol{\beta}}_0) - T_n(\boldsymbol{\beta}^*) \right\|_{\infty,\infty} + \|\mathbf{w}^*\|_1 \cdot \left\| T_n(\boldsymbol{\beta}^*) - \mathbb{E}_{\boldsymbol{\beta}^*}[T_n(\boldsymbol{\beta}^*)] \right\|_{\infty,\infty}$$

$$= O_{\mathbb{P}}\big( \|\mathbf{w}^*\|_1 \cdot \zeta^{\mathrm{L}} \cdot \zeta^{\mathrm{EM}} \big) + O_{\mathbb{P}}\big( \|\mathbf{w}^*\|_1 \cdot \zeta^{\mathrm{T}} \big) = o_{\mathbb{P}}(1).$$

Here the first and second inequalities are from Hölder's inequality and triangle inequality, the first equality follows from (I.46) and (I.47), and the second equality holds because (4.6) in Assumption 4.5 implies

$$\|\mathbf{w}^*\|_1 \cdot \big( \zeta^{\mathrm{L}} \cdot \zeta^{\mathrm{EM}} + \zeta^{\mathrm{T}} \big) \leq \max\{\|\mathbf{w}^*\|_1, \, 1\} \cdot s_{\mathbf{w}}^* \cdot \lambda = o(1)$$

for $\lambda$ specified in (4.5).

**Analysis of Term (iii):** For term (iii) in (I.42), by (D.1) in Theorem D.1 we have

$$\left| \widehat{\mathbf{w}}_0^\top \cdot [T_n(\widehat{\boldsymbol{\beta}}_0)]_{\boldsymbol{\gamma},\boldsymbol{\gamma}} \cdot \widehat{\mathbf{w}}_0 + (\mathbf{w}^*)^\top \cdot [I(\boldsymbol{\beta}^*)]_{\boldsymbol{\gamma},\boldsymbol{\gamma}} \cdot \mathbf{w}^* \right| \tag{I.51}$$

$$\leq \underbrace{\left| \widehat{\mathbf{w}}_0^\top \cdot \left\{ T_n(\widehat{\boldsymbol{\beta}}_0) - \mathbb{E}_{\boldsymbol{\beta}^*}[T_n(\boldsymbol{\beta}^*)] \right\}_{\boldsymbol{\gamma},\boldsymbol{\gamma}} \cdot \widehat{\mathbf{w}}_0 \right|}_{\text{(iii).a}} + \underbrace{\left| \widehat{\mathbf{w}}_0^\top \cdot [I(\boldsymbol{\beta}^*)]_{\boldsymbol{\gamma},\boldsymbol{\gamma}} \cdot \widehat{\mathbf{w}}_0 - (\mathbf{w}^*)^\top \cdot [I(\boldsymbol{\beta}^*)]_{\boldsymbol{\gamma},\boldsymbol{\gamma}} \cdot \mathbf{w}^* \right|}_{\text{(iii).b}}.$$

For term (iii).a in (I.51), we have

$$\left| \widehat{\mathbf{w}}_0^\top \cdot \left\{ T_n(\widehat{\boldsymbol{\beta}}_0) - \mathbb{E}_{\boldsymbol{\beta}^*}[T_n(\boldsymbol{\beta}^*)] \right\}_{\boldsymbol{\gamma},\boldsymbol{\gamma}} \cdot \widehat{\mathbf{w}}_0 \right| \leq \|\widehat{\mathbf{w}}_0\|_1^2 \cdot \left\| \left\{ T_n(\widehat{\boldsymbol{\beta}}_0) - \mathbb{E}_{\boldsymbol{\beta}^*}[T_n(\boldsymbol{\beta}^*)] \right\}_{\boldsymbol{\gamma},\boldsymbol{\gamma}} \right\|_{\infty,\infty}$$

$$\leq \|\widehat{\mathbf{w}}_0\|_1^2 \cdot \left\| T_n(\widehat{\boldsymbol{\beta}}_0) - \mathbb{E}_{\boldsymbol{\beta}^*}[T_n(\boldsymbol{\beta}^*)] \right\|_{\infty,\infty}. \tag{I.52}$$

For $\|\widehat{\mathbf{w}}_0\|_1$ we have $\|\widehat{\mathbf{w}}_0\|_1^2 \leq \big( \|\mathbf{w}^*\|_1 + \|\widehat{\mathbf{w}}_0 - \mathbf{w}^*\|_1 \big)^2 = \big[ \|\mathbf{w}^*\|_1 + O_{\mathbb{P}}(s_{\mathbf{w}}^* \cdot \lambda) \big]^2$, where the equality holds because by Lemma I.3 we have $\|\widehat{\mathbf{w}}_0 - \mathbf{w}^*\|_1 = O_{\mathbb{P}}(s_{\mathbf{w}}^* \cdot \lambda)$. Meanwhile, on the right-hand side of (I.52) we have

$$\left\| T_n(\widehat{\boldsymbol{\beta}}_0) - \mathbb{E}_{\boldsymbol{\beta}^*}[T_n(\boldsymbol{\beta}^*)] \right\|_{\infty,\infty} \leq \left\| T_n(\widehat{\boldsymbol{\beta}}_0) - T_n(\boldsymbol{\beta}^*) \right\|_{\infty,\infty} + \left\| T_n(\boldsymbol{\beta}^*) - \mathbb{E}_{\boldsymbol{\beta}^*}[T_n(\boldsymbol{\beta}^*)] \right\|_{\infty,\infty}$$

$$= O_{\mathbb{P}}\big( \zeta^{\mathrm{L}} \cdot \zeta^{\mathrm{EM}} + \zeta^{\mathrm{T}} \big).$$

Here the last equality is from (I.46) and (I.47). Hence, term (iii).a in (I.51) is $O_{\mathbb{P}}\big[ \big( \|\mathbf{w}^*\|_1 + s_{\mathbf{w}}^* \cdot \lambda \big)^2 \cdot \big( \zeta^{\mathrm{L}} \cdot \zeta^{\mathrm{EM}} + \zeta^{\mathrm{T}} \big) \big]$. Note that

$$\big( \|\mathbf{w}^*\|_1 + s_{\mathbf{w}}^* \cdot \lambda \big)^2 \cdot \big( \zeta^{\mathrm{L}} \cdot \zeta^{\mathrm{EM}} + \zeta^{\mathrm{T}} \big)$$

$$= \underbrace{\|\mathbf{w}^*\|_1^2 \cdot \big( \zeta^{\mathrm{L}} \cdot \zeta^{\mathrm{EM}} + \zeta^{\mathrm{T}} \big)}_{\text{(i)}} + 2 \cdot \underbrace{s_{\mathbf{w}}^* \cdot \lambda}_{\text{(ii)}} \cdot \underbrace{\|\mathbf{w}^*\|_1 \cdot \big( \zeta^{\mathrm{L}} \cdot \zeta^{\mathrm{EM}} + \zeta^{\mathrm{T}} \big)}_{\text{(iii)}} + \underbrace{(s_{\mathbf{w}}^* \cdot \lambda)^2}_{\text{(ii)}} \cdot \underbrace{\big( \zeta^{\mathrm{L}} \cdot \zeta^{\mathrm{EM}} + \zeta^{\mathrm{T}} \big)}_{\text{(iv)}}.$$

From (4.6) in Assumption 4.5 we have, for $\lambda$ specified in (4.5), terms (i)-(iv) are all upper bounded by $\max\{\|\mathbf{w}^*\|_1, 1\} \cdot s_{\mathbf{w}}^* \cdot \lambda = o(1)$. Hence, we conclude term (iii).a in (I.51) is $o_{\mathbb{P}}(1)$. Also, for term (iii).b in (I.51), we have

$$\left| \widehat{\mathbf{w}}_0^\top \cdot [I(\boldsymbol{\beta}^*)]_{\boldsymbol{\gamma},\boldsymbol{\gamma}} \cdot \widehat{\mathbf{w}}_0 - (\mathbf{w}^*)^\top \cdot [I(\boldsymbol{\beta}^*)]_{\boldsymbol{\gamma},\boldsymbol{\gamma}} \cdot \mathbf{w}^* \right|$$

$$\leq \left| (\widehat{\mathbf{w}}_0 - \mathbf{w}^*)^\top \cdot [I(\boldsymbol{\beta}^*)]_{\boldsymbol{\gamma},\boldsymbol{\gamma}} \cdot (\widehat{\mathbf{w}}_0 - \mathbf{w}^*) \right| + 2 \cdot \left| \mathbf{w}^* \cdot [I(\boldsymbol{\beta}^*)]_{\boldsymbol{\gamma},\boldsymbol{\gamma}} \cdot (\widehat{\mathbf{w}}_0 - \mathbf{w}^*) \right|$$

$$\leq \|\widehat{\mathbf{w}}_0 - \mathbf{w}^*\|_1^2 \cdot \left\| I(\boldsymbol{\beta}^*) \right\|_{\infty,\infty} + 2 \cdot \|\widehat{\mathbf{w}}_0 - \mathbf{w}^*\|_1 \cdot \|\mathbf{w}^*\|_1 \cdot \left\| I(\boldsymbol{\beta}^*) \right\|_{\infty,\infty}$$

$$= O_{\mathbb{P}}\big[ (s_{\mathbf{w}}^* \cdot \lambda)^2 + \|\mathbf{w}^*\|_1 \cdot s_{\mathbf{w}}^* \cdot \lambda \big],$$

where the last equality follows from Lemma I.3 and (I.50). Moreover, by (4.6) in Assumption 4.5 we have $\max\{s_{\mathbf{w}}^* \cdot \lambda, \, \|\mathbf{w}^*\|_1 \cdot s_{\mathbf{w}}^* \cdot \lambda\} \leq \max\{\|\mathbf{w}^*\|_1, \, 1\} \cdot s_{\mathbf{w}}^* \cdot \lambda = o(1)$. Therefore, we conclude that term (iii).b in (I.51) is $o_{\mathbb{P}}(1)$. Combining the above analysis for terms (i)-(iii) in (I.42), we obtain

$$\left| [T_n(\widehat{\boldsymbol{\beta}}_0)]_{\alpha|\boldsymbol{\gamma}} + [I(\boldsymbol{\beta}^*)]_{\alpha|\boldsymbol{\gamma}} \right| = o_{\mathbb{P}}(1).$$

Thus we conclude the proof of Lemma G.4. $\qquad\square$

## I.5   Proof of Lemma F.1

*Proof.* According to Algorithm 4, the final estimator $\widehat{\boldsymbol{\beta}} = \boldsymbol{\beta}^{(T)}$ has $\widehat{s}$ nonzero entries. Meanwhile, we have $\|\boldsymbol{\beta}^*\|_0 = s^* \le \widehat{s}$. Hence, we have $\|\widehat{\boldsymbol{\beta}} - \boldsymbol{\beta}^*\|_1 \le 2 \cdot \sqrt{\widehat{s}} \cdot \|\widehat{\boldsymbol{\beta}} - \boldsymbol{\beta}^*\|_2$. Invoking (E.6) in Theorem E.3, we obtain $\zeta^{\mathrm{EM}}$.

For Gaussian mixture model, the maximization implementation of the M-step takes the form

$$M_n(\boldsymbol{\beta}) = \frac{2}{n} \sum_{i=1}^{n} \omega_{\boldsymbol{\beta}}(\mathbf{y}_i) \cdot \mathbf{y}_i - \frac{1}{n} \sum_{i=1}^{n} \mathbf{y}_i, \quad \text{and} \ \ M(\boldsymbol{\beta}) = \mathbb{E}\big[2 \cdot \omega_{\boldsymbol{\beta}}(\boldsymbol{Y}) \cdot \boldsymbol{Y} - \boldsymbol{Y}\big],$$

where $\omega_{\boldsymbol{\beta}}(\cdot)$ is defined in (A.2). Meanwhile, we have

$$\nabla_1 Q_n(\boldsymbol{\beta}; \boldsymbol{\beta}) = \frac{1}{n} \sum_{i=1}^{n} \big[2 \cdot \omega_{\boldsymbol{\beta}}(\mathbf{y}_i) - 1\big] \cdot \mathbf{y}_i - \boldsymbol{\beta}, \quad \text{and} \ \ \nabla_1 Q(\boldsymbol{\beta}; \boldsymbol{\beta}) = \mathbb{E}\big[2 \cdot \omega_{\boldsymbol{\beta}}(\boldsymbol{Y}) - \boldsymbol{Y}\big] - \boldsymbol{\beta}.$$

Hence, we have $\big\|M_n(\boldsymbol{\beta}) - M(\boldsymbol{\beta})\big\|_\infty = \big\|\nabla_1 Q_n(\boldsymbol{\beta}; \boldsymbol{\beta}) - \nabla_1 Q(\boldsymbol{\beta}; \boldsymbol{\beta})\big\|_\infty$. By setting $\delta = 2/d$ in Lemma E.2, we obtain $\zeta^{\mathrm{G}}$. $\qquad\square$

## I.6   Proof of Lemma F.2

*Proof.* Recall that for Gaussian mixture model we have

$$Q_n(\boldsymbol{\beta}'; \boldsymbol{\beta}) = -\frac{1}{2n} \sum_{i=1}^{n} \Big\{ \omega_{\boldsymbol{\beta}}(\mathbf{y}_i) \cdot \|\mathbf{y}_i - \boldsymbol{\beta}'\|_2^2 + \big[1 - \omega_{\boldsymbol{\beta}}(\mathbf{y}_i)\big] \cdot \|\mathbf{y}_i + \boldsymbol{\beta}'\|_2^2 \Big\},$$

where $\omega_{\boldsymbol{\beta}}(\cdot)$ is defined in (A.2). Hence, by calculation we have

$$\nabla_1 Q_n(\boldsymbol{\beta}'; \boldsymbol{\beta}) = \frac{1}{n} \sum_{i=1}^{n} \big[2 \cdot \omega_{\boldsymbol{\beta}}(\mathbf{y}_i) - 1\big] \cdot \mathbf{y}_i - \boldsymbol{\beta}', \quad \nabla_{1,1}^2 Q_n(\boldsymbol{\beta}'; \boldsymbol{\beta}) = -\mathbf{I}_d, \tag{I.53}$$

$$\nabla_{1,2}^2 Q_n(\boldsymbol{\beta}'; \boldsymbol{\beta}) = \frac{4}{n} \sum_{i=1}^{n} \frac{\mathbf{y}_i \cdot \mathbf{y}_i^\top}{\sigma^2 \cdot \big[1 + \exp\big(-2 \cdot \langle \boldsymbol{\beta}, \mathbf{y}_i \rangle / \sigma^2\big)\big] \cdot \big[1 + \exp\big(2 \cdot \langle \boldsymbol{\beta}, \mathbf{y}_i \rangle / \sigma^2\big)\big]}. \tag{I.54}$$

For notational simplicity we define

$$\nu_{\boldsymbol{\beta}}(\mathbf{y}) = \frac{4}{\sigma^2 \cdot \big[1 + \exp\big(-2 \cdot \langle \boldsymbol{\beta}, \mathbf{y} \rangle / \sigma^2\big)\big] \cdot \big[1 + \exp\big(2 \cdot \langle \boldsymbol{\beta}, \mathbf{y} \rangle / \sigma^2\big)\big]}. \tag{I.55}$$

Then by the definition of $T_n(\cdot)$ in (2.4), from (I.53) and (I.54) we have

$$\big\{T_n(\boldsymbol{\beta}^*) - \mathbb{E}_{\boldsymbol{\beta}^*}\big[T_n(\boldsymbol{\beta}^*)\big]\big\}_{j,k} = \frac{1}{n} \sum_{i=1}^{n} \nu_{\boldsymbol{\beta}^*}(\mathbf{y}_i) \cdot y_{i,j} \cdot y_{i,k} - \mathbb{E}_{\boldsymbol{\beta}^*}\big[\nu_{\boldsymbol{\beta}^*}(\boldsymbol{Y}) \cdot Y_j \cdot Y_k\big].$$

Applying the symmetrization result in Lemma J.4 to the right-hand side, we have that for $u > 0$,

$$\mathbb{E}_{\boldsymbol{\beta}^*}\Big\{\exp\Big[u \cdot \Big|\big\{T_n(\boldsymbol{\beta}^*) - \mathbb{E}_{\boldsymbol{\beta}^*}\big[T_n(\boldsymbol{\beta}^*)\big]\big\}_{j,k}\Big|\Big]\Big\} \le \mathbb{E}_{\boldsymbol{\beta}^*}\Big\{\exp\Big[u \cdot \Big|\frac{1}{n} \sum_{i=1}^{n} \xi_i \cdot \nu_{\boldsymbol{\beta}^*}(\mathbf{y}_i) \cdot y_{i,j} \cdot y_{i,k}\Big|\Big]\Big\}, \tag{I.56}$$

where $\xi_1, \ldots, \xi_n$ are i.i.d. Rademacher random variables that are independent of $\mathbf{y}_1, \ldots, \mathbf{y}_n$. Then we invoke the contraction result in Lemma J.5 by setting

$$f(y_{i,j} \cdot y_{i,k}) = y_{i,j} \cdot y_{i,k}, \quad \mathcal{F} = \{f\}, \quad \psi_i(v) = \nu_{\boldsymbol{\beta}^*}(\mathbf{y}_i) \cdot v, \quad \text{and} \ \ \phi(v) = \exp(u \cdot v),$$

where $u$ is the variable of the moment generating function in (I.56). By the definition in (I.55) we have $\big|\nu_{\boldsymbol{\beta}^*}(\mathbf{y}_i)\big| \le 4/\sigma^2$, which implies

$$\big|\psi_i(v) - \psi_i(v')\big| \le \big|\nu_{\boldsymbol{\beta}^*}(\mathbf{y}_i) \cdot (v - v')\big| \le 4/\sigma^2 \cdot |v - v'|, \quad \text{for all } v, v' \in \mathbb{R}.$$

Therefore, applying the contraction result in Lemma J.5 to the right-hand side of (I.56), we obtain

$$\mathbb{E}_{\boldsymbol{\beta}^*}\Big\{\exp\Big[u \cdot \Big|\big\{T_n(\boldsymbol{\beta}^*) - \mathbb{E}_{\boldsymbol{\beta}^*}\big[T_n(\boldsymbol{\beta}^*)\big]\big\}_{j,k}\Big|\Big]\Big\} \le \mathbb{E}_{\boldsymbol{\beta}^*}\Big\{\exp\Big[u \cdot 4/\sigma^2 \cdot \Big|\frac{1}{n} \sum_{i=1}^{n} \xi_i \cdot y_{i,j} \cdot y_{i,k}\Big|\Big]\Big\}. \tag{I.57}$$

Note that $\mathbb{E}_{\boldsymbol{\beta}^*}(\xi_i \cdot y_{i,j} \cdot y_{i,k}) = 0$, since $\xi_i$ is a Rademacher random variable independent of $y_{i,j} \cdot y_{i,k}$. Recall that in Gaussian mixture model we have $y_{i,j} = z_i \cdot \beta_j^* + v_{i,j}$, where $z_i$ is a Rademacher

random variable and $v_{i,j} \sim N(0, \sigma^2)$. Hence, by Example 5.8 in [28], we have $\|z_i \cdot \beta_j^*\|_{\psi_2} \leq |\beta_j^*|$ and $\|v_{i,j}\|_{\psi_2} \leq C \cdot \sigma$. Therefore, by Lemma J.1 we have

$$\|y_{i,j}\|_{\psi_2} = \|z_i \cdot \beta_j^* + v_{i,j}\|_{\psi_2} \leq C' \cdot \sqrt{|\beta_j^*|^2 + C'' \cdot \sigma^2} \leq C' \cdot \sqrt{\|\boldsymbol{\beta}^*\|_\infty^2 + C'' \cdot \sigma^2}. \quad \text{(I.58)}$$

Since $|\xi_i \cdot y_{i,j} \cdot y_{i,k}| = |y_{i,j} \cdot y_{i,k}|$, by definition $\xi_i \cdot y_{i,j} \cdot y_{i,k}$ and $y_{i,j} \cdot y_{i,k}$ have the same $\psi_1$-norm. By Lemma J.2 we have

$$\|\xi_i \cdot y_{i,j} \cdot y_{i,k}\|_{\psi_1} \leq C \cdot \max\{\|y_{i,j}\|_{\psi_2}^2, \ \|y_{i,k}\|_{\psi_2}^2\} \leq C' \cdot (\|\boldsymbol{\beta}^*\|_\infty^2 + C'' \cdot \sigma^2).$$

Then by Lemma 5.15 in [28], we obtain

$$\mathbb{E}_{\boldsymbol{\beta}^*}\left[\exp(u' \cdot \xi_i \cdot y_{i,j} \cdot y_{i,k})\right] \leq \exp\left[(u')^2 \cdot C \cdot (\|\boldsymbol{\beta}^*\|_\infty^2 + C' \cdot \sigma^2)\right] \quad \text{(I.59)}$$

for all $|u'| \leq C''/(\|\boldsymbol{\beta}^*\|_\infty^2 + C' \cdot \sigma^2)$. Note that on the right-hand side of (I.57), we have

$$\mathbb{E}_{\boldsymbol{\beta}^*}\left\{\exp\left[u \cdot 4/\sigma^2 \cdot \left|\frac{1}{n}\sum_{i=1}^n \xi_i \cdot y_{i,j} \cdot y_{i,k}\right|\right]\right\}$$

$$\leq \mathbb{E}_{\boldsymbol{\beta}^*}\left(\max\left\{\exp\left[u \cdot 4/\sigma^2 \cdot \frac{1}{n}\sum_{i=1}^n \xi_i \cdot y_{i,j} \cdot y_{i,k}\right], \ \exp\left[-u \cdot 4/\sigma^2 \cdot \frac{1}{n}\sum_{i=1}^n \xi_i \cdot y_{i,j} \cdot y_{i,k}\right]\right\}\right)$$

$$\leq \mathbb{E}_{\boldsymbol{\beta}^*}\left\{\exp\left[u \cdot 4/\sigma^2 \cdot \frac{1}{n}\sum_{i=1}^n \xi_i \cdot y_{i,j} \cdot y_{i,k}\right]\right\} + \mathbb{E}_{\boldsymbol{\beta}^*}\left\{\exp\left[-u \cdot 4/\sigma^2 \cdot \frac{1}{n}\sum_{i=1}^n \xi_i \cdot y_{i,j} \cdot y_{i,k}\right]\right\}. \tag{I.60}$$

By plugging (I.59) into the right-hand side of (I.60) with $u' = u \cdot 4/(\sigma^2 \cdot n)$ and $u' = -u \cdot 4/(\sigma^2 \cdot n)$, from (I.57) we have that for any $j, k \in \{1, \dots, d\}$,

$$\mathbb{E}_{\boldsymbol{\beta}^*}\left\{\exp\left[u \cdot \left|\{T_n(\boldsymbol{\beta}^*) - \mathbb{E}_{\boldsymbol{\beta}^*}[T_n(\boldsymbol{\beta}^*)]\}_{j,k}\right|\right]\right\} \tag{I.61}$$

$$\leq 2 \cdot \exp\left[C \cdot u^2/n \cdot (\|\boldsymbol{\beta}^*\|_\infty^2 + C' \cdot \sigma^2)^2/\sigma^4\right].$$

Therefore, by Chernoff bound we have that, for all $v > 0$ and $|u| \leq C''/(\|\boldsymbol{\beta}^*\|_\infty^2 + C' \cdot \sigma^2)$,

$$\mathbb{P}\left[\left\|T_n(\boldsymbol{\beta}^*) - \mathbb{E}_{\boldsymbol{\beta}^*}[T_n(\boldsymbol{\beta}^*)]\right\|_{\infty,\infty} > v\right]$$

$$\leq \mathbb{E}_{\boldsymbol{\beta}^*}\left\{\exp\left[u \cdot \left\|T_n(\boldsymbol{\beta}^*) - \mathbb{E}_{\boldsymbol{\beta}^*}[T_n(\boldsymbol{\beta}^*)]\right\|_{\infty,\infty}\right]\right\}\Big/\exp(u \cdot v)$$

$$\leq \sum_{j=1}^d \sum_{k=1}^d \mathbb{E}_{\boldsymbol{\beta}^*}\left\{\exp\left[u \cdot \left|\{T_n(\boldsymbol{\beta}^*) - \mathbb{E}_{\boldsymbol{\beta}^*}[T_n(\boldsymbol{\beta}^*)]\}_{j,k}\right|\right]\right\}\Big/\exp(u \cdot v)$$

$$\leq 2 \cdot \exp\left[C \cdot u^2/n \cdot (\|\boldsymbol{\beta}^*\|_\infty^2 + C' \cdot \sigma^2)^2/\sigma^4 - u \cdot v + 2 \cdot \log d\right],$$

where the last inequality is obtained from (I.61). By minimizing its right-hand side over $u$, we conclude that for $0 < v \leq C'' \cdot (\|\boldsymbol{\beta}^*\|_\infty^2 + C' \cdot \sigma^2)$,

$$\mathbb{P}\left[\left\|T_n(\boldsymbol{\beta}^*) - \mathbb{E}_{\boldsymbol{\beta}^*}[T_n(\boldsymbol{\beta}^*)]\right\|_{\infty,\infty} > v\right] \leq 2 \cdot \exp\left\{-n \cdot v^2\Big/\left[C \cdot (\|\boldsymbol{\beta}^*\|_\infty^2 + C' \cdot \sigma^2)^2/\sigma^4\right] + 2 \cdot \log d\right\}.$$

Setting the right-hand side to be $\delta$, we have that

$$\left\|T_n(\boldsymbol{\beta}^*) - \mathbb{E}_{\boldsymbol{\beta}^*}[T_n(\boldsymbol{\beta}^*)]\right\|_{\infty,\infty} \leq v = C \cdot (\|\boldsymbol{\beta}^*\|_\infty^2 + C' \cdot \sigma^2)/\sigma^2 \cdot \sqrt{\frac{\log(2/\delta) + 2 \cdot \log d}{n}}$$

holds with probability at least $1 - \delta$. By setting $\delta = 2/d$, we conclude the proof of Lemma F.2. $\square$

## I.7 Proof of Lemma F.3

*Proof.* For any $j, k \in \{1, \ldots, d\}$, by the mean-value theorem we have

$$\left\| T_n(\boldsymbol{\beta}) - T_n(\boldsymbol{\beta}^*) \right\|_{\infty,\infty} = \max_{j,k \in \{1,\ldots,d\}} \left| \left[ T_n(\boldsymbol{\beta}) \right]_{j,k} - \left[ T_n(\boldsymbol{\beta}^*) \right]_{j,k} \right| \tag{I.62}$$

$$= \max_{j,k \in \{1,\ldots,d\}} \left| (\boldsymbol{\beta} - \boldsymbol{\beta}^*)^\top \cdot \nabla \left[ T_n(\boldsymbol{\beta}^\sharp) \right]_{j,k} \right|$$

$$\leq \| \boldsymbol{\beta} - \boldsymbol{\beta}^* \|_1 \cdot \max_{j,k \in \{1,\ldots,d\}} \left\| \nabla \left[ T_n(\boldsymbol{\beta}^\sharp) \right]_{j,k} \right\|_\infty,$$

where $\boldsymbol{\beta}^\sharp$ is an intermediate value between $\boldsymbol{\beta}$ and $\boldsymbol{\beta}^*$. According to (I.53), (I.54) and the definition of $T_n(\cdot)$ in (2.4), by calculation we have

$$\nabla \left[ T_n(\boldsymbol{\beta}^\sharp) \right]_{j,k} = \frac{1}{n} \sum_{i=1}^n \bar{\nu}_{\boldsymbol{\beta}^\sharp}(\mathbf{y}_i) \cdot y_{i,j} \cdot y_{i,k} \cdot \mathbf{y}_i,$$

where

$$\bar{\nu}_{\boldsymbol{\beta}}(\mathbf{y}) = \frac{8/\sigma^4}{\left[ 1 + \exp\left( -2 \cdot \langle \boldsymbol{\beta}, \mathbf{y} \rangle / \sigma^2 \right) \right] \cdot \left[ 1 + \exp\left( 2 \cdot \langle \boldsymbol{\beta}, \mathbf{y} \rangle / \sigma^2 \right) \right]^2} \tag{I.63}$$

$$- \frac{8/\sigma^4}{\left[ 1 + \exp\left( -2 \cdot \langle \boldsymbol{\beta}, \mathbf{y} \rangle / \sigma^2 \right) \right]^2 \cdot \left[ 1 + \exp\left( 2 \cdot \langle \boldsymbol{\beta}, \mathbf{y} \rangle / \sigma^2 \right) \right]}.$$

For notational simplicity, we define the following event

$$\mathcal{E} = \left\{ \| \mathbf{y}_i \|_\infty \leq \tau, \ \text{for all } i = 1, \ldots, n \right\},$$

where $\tau > 0$ will be specified later. By maximal inequality we have

$$\mathbb{P}\left\{ \left\| \nabla \left[ T_n(\boldsymbol{\beta}^\sharp) \right]_{j,k} \right\|_\infty > v \right\} \leq d \cdot \mathbb{P}\left( \left\{ \left| \nabla \left[ T_n(\boldsymbol{\beta}^\sharp) \right]_{j,k} \right| \right\}_l > v \right)$$

$$= d \cdot \mathbb{P}\left[ \left| \frac{1}{n} \sum_{i=1}^n \bar{\nu}_{\boldsymbol{\beta}^\sharp}(\mathbf{y}_i) \cdot y_{i,j} \cdot y_{i,k} \cdot y_{i,l} \right| > v \right]. \tag{I.64}$$

Let $\bar{\mathcal{E}}$ be the complement of $\mathcal{E}$. On the right-hand side of (I.64) we have

$$\mathbb{P}\left[ \left| \frac{1}{n} \sum_{i=1}^n \bar{\nu}_{\boldsymbol{\beta}^\sharp}(\mathbf{y}_i) \cdot y_{i,j} \cdot y_{i,k} \cdot y_{i,l} \right| > v \right] = \underbrace{\mathbb{P}\left[ \left| \frac{1}{n} \sum_{i=1}^n \bar{\nu}_{\boldsymbol{\beta}^\sharp}(\mathbf{y}_i) \cdot y_{i,j} \cdot y_{i,k} \cdot y_{i,l} \right| > v, \mathcal{E} \right]}_{\text{(i)}} + \underbrace{\mathbb{P}(\bar{\mathcal{E}})}_{\text{(ii)}}. \tag{I.65}$$

**Analysis of Term (i):** For term (i) in (I.65), we have

$$\mathbb{P}\left[ \left| \frac{1}{n} \sum_{i=1}^n \bar{\nu}_{\boldsymbol{\beta}^\sharp}(\mathbf{y}_i) \cdot y_{i,j} \cdot y_{i,k} \cdot y_{i,l} \right| > v, \mathcal{E} \right]$$

$$= \mathbb{P}\left[ \left| \frac{1}{n} \sum_{i=1}^n \bar{\nu}_{\boldsymbol{\beta}^\sharp}(\mathbf{y}_i) \cdot y_{i,j} \cdot y_{i,k} \cdot y_{i,l} \cdot \mathbb{1}\{ \| \mathbf{y}_i \|_\infty \leq \tau \} \right| > v, \mathcal{E} \right]$$

$$\leq \mathbb{P}\left[ \left| \frac{1}{n} \sum_{i=1}^n \bar{\nu}_{\boldsymbol{\beta}^\sharp}(\mathbf{y}_i) \cdot y_{i,j} \cdot y_{i,k} \cdot y_{i,l} \cdot \mathbb{1}\{ \| \mathbf{y}_i \|_\infty \leq \tau \} \right| > v \right]$$

$$\leq \sum_{i=1}^n \mathbb{P}\left[ \left| \bar{\nu}_{\boldsymbol{\beta}^\sharp}(\mathbf{y}_i) \cdot y_{i,j} \cdot y_{i,k} \cdot y_{i,l} \cdot \mathbb{1}\{ \| \mathbf{y}_i \|_\infty \leq \tau \} \right| > v \right],$$

where the last inequality is from union bound. By (I.63) we have $|\bar{\nu}_{\boldsymbol{\beta}^\sharp}(\mathbf{y}_i)| \leq 16/\sigma^4$. Thus we obtain

$$\mathbb{P}\left[ \left| \bar{\nu}_{\boldsymbol{\beta}^\sharp}(\mathbf{y}_i) \cdot y_{i,j} \cdot y_{i,k} \cdot y_{i,l} \cdot \mathbb{1}\{ \| \mathbf{y}_i \|_\infty \leq \tau \} \right| > v \right] \leq \mathbb{P}\left[ \left| y_{i,j} \cdot y_{i,k} \cdot y_{i,l} \cdot \mathbb{1}\{ \| \mathbf{y}_i \|_\infty \leq \tau \} \right| > \sigma^4/16 \cdot v \right].$$

Taking $v = 16 \cdot \tau^3/\sigma^4$, we have that the right-hand side is zero and hence term (i) in (I.65) is zero.

**Analysis of Term (ii):** Meanwhile, for term (ii) in (I.65) by maximal inequality we have

$$\mathbb{P}(\bar{\mathcal{E}}) = \mathbb{P}\left( \max_{i \in \{1,\ldots,n\}} \| \mathbf{y}_i \|_\infty > \tau \right) \leq n \cdot \mathbb{P}(\| \mathbf{y}_i \|_\infty > \tau) \leq n \cdot d \cdot \mathbb{P}(|y_{i,j}| > \tau).$$

Furthermore, by (I.58) in the proof of Lemma F.1, we have that $y_{i,j}$ is sub-Gaussian with $\|y_{i,j}\|_{\psi_2} = C \cdot \sqrt{\|\boldsymbol{\beta}^*\|_\infty^2 + C' \cdot \sigma^2}$. Therefore, by Lemma 5.5 in [28] we have

$$\mathbb{P}(\bar{\mathcal{E}}) \le n \cdot d \cdot \mathbb{P}(|y_{i,j}| > \tau) \le n \cdot d \cdot 2 \cdot \exp\left[-C \cdot \tau^2 / (\|\boldsymbol{\beta}^*\|_\infty^2 + C' \cdot \sigma^2)\right].$$

To ensure the right-hand side is upper bounded by $\delta$, we set $\tau$ to be

$$\tau = C \cdot \sqrt{\|\boldsymbol{\beta}^*\|_\infty^2 + C' \cdot \sigma^2} \cdot \sqrt{\log d + \log n + \log(2/\delta)}. \tag{I.66}$$

Finally, by (I.64), (I.65) and maximal inequality we have

$$\mathbb{P}\left\{\max_{j,k \in \{1,\dots,d\}} \left\|\nabla\big[T_n(\boldsymbol{\beta}^\sharp)\big]_{j,k}\right\|_\infty > v\right\} \le d^2 \cdot d \cdot \mathbb{P}\left[\left|\frac{1}{n}\sum_{i=1}^n \bar{\nu}_{\boldsymbol{\beta}^\sharp}(\mathbf{y}_i) \cdot y_{i,j} \cdot y_{i,k} \cdot y_{i,l}\right| > v\right] \le d^3 \cdot \delta$$

for $v = 16 \cdot \tau^3/\sigma^4$ with $\tau$ specified in (I.66). By setting $\delta = 2 \cdot d^{-4}$ and plugging (I.66) into (I.62), we conclude the proof of Lemma F.3. $\qquad\square$

## I.8  Proof of Lemma F.5

By the same proof of Lemma F.1 in §I.5, we obtain $\zeta^{\mathrm{EM}}$ by invoking Theorem E.6. To obtain $\zeta^{\mathrm{G}}$, note that for the gradient implementation of the M-step (Algorithm 3), we have

$$M_n(\boldsymbol{\beta}) = \boldsymbol{\beta} + \eta \cdot \nabla_1 Q_n(\boldsymbol{\beta};\boldsymbol{\beta}), \quad \text{and} \ \ M(\boldsymbol{\beta}) = \boldsymbol{\beta} + \eta \cdot \nabla_1 Q(\boldsymbol{\beta};\boldsymbol{\beta}).$$

Hence, we obtain $\|\nabla_1 Q_n(\boldsymbol{\beta}^*;\boldsymbol{\beta}^*) - \nabla_1 Q(\boldsymbol{\beta}^*;\boldsymbol{\beta}^*)\|_\infty = 1/\eta \cdot \|M_n(\boldsymbol{\beta}^*) - M(\boldsymbol{\beta}^*)\|_\infty$. Setting $\delta = 4/d$ and $s = s^*$ in (E.9) of Lemma E.5, we then obtain $\zeta^{\mathrm{G}}$.

## I.9  Proof of Lemma F.6

*Proof.* Recall that for mixture of regression model, we have

$$Q_n(\boldsymbol{\beta}';\boldsymbol{\beta}) = -\frac{1}{2n}\sum_{i=1}^n \Big\{\omega_{\boldsymbol{\beta}}(\mathbf{x}_i, y_i) \cdot \big(y_i - \langle \mathbf{x}_i, \boldsymbol{\beta}'\rangle\big)^2 + \big[1 - \omega_{\boldsymbol{\beta}}(\mathbf{x}_i, y_i)\big] \cdot \big(y_i + \langle \mathbf{x}_i, \boldsymbol{\beta}'\rangle\big)^2\Big\},$$

where $\omega_{\boldsymbol{\beta}}(\cdot)$ is defined in (A.5). Hence, by calculation we have

$$\nabla_1 Q_n(\boldsymbol{\beta}',\boldsymbol{\beta}) = \frac{1}{n}\sum_{i=1}^n \big[2 \cdot \omega_{\boldsymbol{\beta}}(\mathbf{x}_i, y_i) \cdot y_i \cdot \mathbf{x}_i - \mathbf{x}_i \cdot \mathbf{x}_i^\top \cdot \boldsymbol{\beta}'\big], \quad \nabla_{1,1}^2 Q_n(\boldsymbol{\beta}',\boldsymbol{\beta}) = -\frac{1}{n}\sum_{i=1}^n \mathbf{x}_i \cdot \mathbf{x}_i^\top, \tag{I.67}$$

$$\nabla_{1,2}^2 Q_n(\boldsymbol{\beta}',\boldsymbol{\beta}) = \frac{4}{n}\sum_{i=1}^n \frac{y_i^2 \cdot \mathbf{x}_i \cdot \mathbf{x}_i^\top}{\sigma^2 \cdot \big[1 + \exp\big(-2 \cdot y_i \cdot \langle \boldsymbol{\beta}, \mathbf{x}_i\rangle/\sigma^2\big)\big] \cdot \big[1 + \exp\big(2 \cdot y_i \cdot \langle \boldsymbol{\beta}, \mathbf{x}_i\rangle/\sigma^2\big)\big]}. \tag{I.68}$$

For notational simplicity we define

$$\nu_{\boldsymbol{\beta}}(\mathbf{x}, y) = \frac{4}{\sigma^2 \cdot \big[1 + \exp\big(-2 \cdot y \cdot \langle \boldsymbol{\beta}, \mathbf{x}\rangle/\sigma^2\big)\big] \cdot \big[1 + \exp\big(2 \cdot y \cdot \langle \boldsymbol{\beta}, \mathbf{x}\rangle/\sigma^2\big)\big]}. \tag{I.69}$$

Then by the definition of $T_n(\cdot)$ in (2.4), from (I.67) and (I.68) we have

$$\big\{T_n(\boldsymbol{\beta}^*) - \mathbb{E}_{\boldsymbol{\beta}^*}\big[T_n(\boldsymbol{\beta}^*)\big]\big\}_{j,k} = \frac{1}{n}\sum_{i=1}^n \nu_{\boldsymbol{\beta}^*}(\mathbf{x}_i, y_i) \cdot x_{i,j} \cdot x_{i,k} \cdot y_i^2 - \mathbb{E}_{\boldsymbol{\beta}^*}\big[\nu_{\boldsymbol{\beta}^*}(Y, \boldsymbol{X}) \cdot X_j \cdot X_k \cdot Y_i^2\big].$$

Applying the symmetrization result in Lemma J.4 to the right-hand side, we have that for $u > 0$,

$$\mathbb{E}_{\boldsymbol{\beta}^*}\left\{\exp\left[u \cdot \left|\big\{T_n(\boldsymbol{\beta}^*) - \mathbb{E}_{\boldsymbol{\beta}^*}\big[T_n(\boldsymbol{\beta}^*)\big]\big\}_{j,k}\right|\right]\right\}$$

$$\le \mathbb{E}_{\boldsymbol{\beta}^*}\left\{\exp\left[u \cdot \left|\frac{1}{n}\sum_{i=1}^n \xi_i \cdot \nu_{\boldsymbol{\beta}^*}(\mathbf{x}_i, y_i) \cdot x_{i,j} \cdot x_{i,k} \cdot y_i^2\right|\right]\right\}, \tag{I.70}$$

where $\xi_1,\dots,\xi_n$ are i.i.d. Rademacher random variables, which are independent of $\mathbf{x}_1,\dots,\mathbf{x}_n$ and $y_1,\dots,y_n$. Then we invoke the contraction result in Lemma J.5 by setting

$$f\big(x_{i,j} \cdot x_{i,k} \cdot y_i^2\big) = x_{i,j} \cdot x_{i,k} \cdot y_i^2, \quad \mathcal{F} = \{f\}, \quad \psi_i(v) = \nu_{\boldsymbol{\beta}^*}(\mathbf{x}_i, y_i) \cdot v, \quad \text{and} \ \ \phi(v) = \exp(u \cdot v),$$

where $u$ is the variable of the moment generating function in (I.70). By the definition in (I.69) we have $|\nu_{\boldsymbol{\beta}^*}(\mathbf{x}_i, y_i)| \le 4/\sigma^2$, which implies

$$\big|\psi_i(v) - \psi_i(v')\big| \le \big|\nu_{\boldsymbol{\beta}^*}(\mathbf{x}_i, y_i) \cdot (v - v')\big| \le 4/\sigma^2 \cdot |v - v'|, \quad \text{for all } v, v' \in \mathbb{R}.$$

Therefore, applying Lemma J.5 to the right-hand side of (I.70), we obtain

$$
\mathbb{E}_{\boldsymbol{\beta}^*}\left\{\exp\left[u\cdot\left|\left\{T_n(\boldsymbol{\beta}^*)-\mathbb{E}_{\boldsymbol{\beta}^*}\left[T_n(\boldsymbol{\beta}^*)\right]\right\}_{j,k}\right|\right]\right\}
$$

$$
\leq \mathbb{E}_{\boldsymbol{\beta}^*}\left\{\exp\left[u\cdot 4/\sigma^2\cdot\left|\frac{1}{n}\sum_{i=1}^{n}\xi_i\cdot x_{i,j}\cdot x_{i,k}\cdot y_i^2\right|\right]\right\}. \tag{I.71}
$$

For notational simplicity, we define the following event

$$
\mathcal{E}=\left\{\|\mathbf{x}_i\|_\infty\leq\tau,\ \text{for all}\ i=1,\ldots,n\right\}.
$$

Let $\bar{\mathcal{E}}$ be the complement of $\mathcal{E}$. We consider the following tail probability

$$
\mathbb{P}\left[4/\sigma^2\cdot\left|\frac{1}{n}\sum_{i=1}^{n}\xi_i\cdot x_{i,j}\cdot x_{i,k}\cdot y_i^2\right|>v\right]\leq\underbrace{\mathbb{P}\left[4/\sigma^2\cdot\left|\frac{1}{n}\sum_{i=1}^{n}\xi_i\cdot x_{i,j}\cdot x_{i,k}\cdot y_i^2\right|>v,\mathcal{E}\right]}_{\text{(i)}}+\underbrace{\mathbb{P}(\bar{\mathcal{E}})}_{\text{(ii)}}.
$$

$$\tag{I.72}$$

**Analysis of Term (i):** For term (i) in (I.72), we have

$$
\mathbb{P}\left[4/\sigma^2\cdot\left|\frac{1}{n}\sum_{i=1}^{n}\xi_i\cdot x_{i,j}\cdot x_{i,k}\cdot y_i^2\right|>v,\mathcal{E}\right]=\mathbb{P}\left[4/\sigma^2\cdot\left|\frac{1}{n}\sum_{i=1}^{n}\xi_i\cdot x_{i,j}\cdot x_{i,k}\cdot y_i^2\cdot\mathbb{1}\left\{\|\mathbf{x}_i\|_\infty\leq\tau\right\}\right|>v,\mathcal{E}\right]
$$

$$
\leq\mathbb{P}\left[4/\sigma^2\cdot\left|\frac{1}{n}\sum_{i=1}^{n}\xi_i\cdot x_{i,j}\cdot x_{i,k}\cdot y_i^2\cdot\mathbb{1}\left\{\|\mathbf{x}_i\|_\infty\leq\tau\right\}\right|>v\right].
$$

Here note that $\mathbb{E}_{\boldsymbol{\beta}^*}\left(\xi_i\cdot x_{i,j}\cdot x_{i,k}\cdot y_i^2\cdot\mathbb{1}\left\{\|\mathbf{x}_i\|_\infty\leq\tau\right\}\right)=0$, because $\xi_i$ is a Rademacher random variable independent of $\mathbf{x}_i$ and $y_i$. Recall that for mixture of regression model we have $y_i=z_i\cdot\langle\boldsymbol{\beta}^*,\mathbf{x}_i\rangle+v_i$, where $z_i$ is a Rademacher random variable, $\mathbf{x}_i\sim N(0,\mathbf{I}_d)$ and $v_i\sim N(0,\sigma^2)$. According to Example 5.8 in [28], we have $\left\|z_i\cdot\langle\boldsymbol{\beta}^*,\mathbf{x}_i\rangle\cdot\mathbb{1}\left\{\|\mathbf{x}_i\|_\infty\leq\tau\right\}\right\|_{\psi_2}=\left\|\langle\boldsymbol{\beta}^*,\mathbf{x}_i\rangle\cdot\mathbb{1}\left\{\|\mathbf{x}_i\|_\infty\leq\tau\right\}\right\|_{\psi_2}\leq\tau\cdot\|\boldsymbol{\beta}^*\|_1$ and $\left\|v_i\cdot\mathbb{1}\left\{\|\mathbf{x}_i\|_\infty\leq\tau\right\}\right\|_{\psi_2}\leq\|v_i\|_{\psi_2}\leq C\cdot\sigma$. Hence, by Lemma J.1 we have

$$
\left\|y_i\cdot\mathbb{1}\left\{\|\mathbf{x}_i\|_\infty\leq\tau\right\}\right\|_{\psi_2}=\left\|z_i\cdot\langle\boldsymbol{\beta}^*,\mathbf{x}_i\rangle\cdot\mathbb{1}\left\{\|\mathbf{x}_i\|_\infty\leq\tau\right\}+v_i\cdot\mathbb{1}\left\{\|\mathbf{x}_i\|_\infty\leq\tau\right\}\right\|_{\psi_2}
$$

$$
\leq C\cdot\sqrt{\tau^2\cdot\|\boldsymbol{\beta}^*\|_1^2+C'\cdot\sigma^2}. \tag{I.73}
$$

By the definition of $\psi_1$-norm, we have $\left\|\xi_i\cdot x_{i,j}\cdot x_{i,k}\cdot y_i^2\cdot\mathbb{1}\left\{\|\mathbf{x}_i\|_\infty\leq\tau\right\}\right\|_{\psi_1}\leq\tau^2\cdot\left\|y_i^2\cdot\mathbb{1}\left\{\|\mathbf{x}_i\|_\infty\leq\tau\right\}\right\|_{\psi_1}$. Further by applying Lemma J.2 to its right-hand side with $Z_1=Z_2=y_i\cdot\mathbb{1}\left\{\|\mathbf{x}_i\|_\infty\leq\tau\right\}$, we obtain

$$
\left\|\xi_i\cdot x_{i,j}\cdot x_{i,k}\cdot y_i^2\cdot\mathbb{1}\left\{\|\mathbf{x}_i\|_\infty\leq\tau\right\}\right\|_{\psi_1}\leq C\cdot\tau^2\cdot\left\|y_i\cdot\mathbb{1}\left\{\|\mathbf{x}_i\|_\infty\leq\tau\right\}\right\|_{\psi_2}^2
$$

$$
\leq C'\cdot\tau^2\cdot\left(\tau^2\cdot\|\boldsymbol{\beta}^*\|_1^2+C''\cdot\sigma^2\right),
$$

where the last inequality follows from (I.73). Therefore, by Bernstein's inequality (Proposition 5.16 in [28]), we obtain

$$
\mathbb{P}\left[4/\sigma^2\cdot\left|\frac{1}{n}\sum_{i=1}^{n}\xi_i\cdot x_{i,j}\cdot x_{i,k}\cdot y_i^2\cdot\mathbb{1}\left\{\|\mathbf{x}_i\|_\infty\leq\tau\right\}\right|>v\right]
$$

$$
\leq 2\cdot\exp\left[-\frac{C\cdot n\cdot v^2\cdot\sigma^4}{\tau^4\cdot\left(\tau^2\cdot\|\boldsymbol{\beta}^*\|_1^2+C'\cdot\sigma^2\right)^2}\right], \tag{I.74}
$$

for all $0\leq v\leq C\cdot\tau^2\cdot\left(\|\boldsymbol{\beta}^*\|_1^2+C'\cdot\sigma^2\right)$ and a sufficiently large sample size $n$.

**Analysis of Term (ii):** For term (ii) in (I.72), by union bound we have

$$
\mathbb{P}(\bar{\mathcal{E}})=\mathbb{P}\left(\max_{1\leq i\leq n}\|\mathbf{x}_i\|_\infty>\tau\right)\leq n\cdot\mathbb{P}\left(\|\mathbf{x}_i\|_\infty>\tau\right)\leq n\cdot d\cdot\mathbb{P}\left(|x_{i,j}|>\tau\right).
$$

Moreover, $x_{i,j}$ is sub-Gaussian with $\|x_{i,j}\|_{\psi_2}=C$. Thus, by Lemma 5.5 in [28] we have

$$
\mathbb{P}(\bar{\mathcal{E}})\leq n\cdot d\cdot 2\cdot\exp\left(-C'\cdot\tau^2\right)=2\cdot\exp\left(-C'\cdot\tau^2+\log n+\log d\right). \tag{I.75}
$$

Plugging (I.74) and (I.75) into (I.72), we obtain

$$\mathbb{P}\left[4/\sigma^2 \cdot \left|\frac{1}{n}\sum_{i=1}^{n}\xi_i \cdot x_{i,j} \cdot x_{i,k} \cdot y_i^2\right| > v\right] \tag{I.76}$$

$$\leq 2 \cdot \exp\left[-\frac{C \cdot n \cdot v^2 \cdot \sigma^4}{\tau^4 \cdot \left(\tau^2 \cdot \|\boldsymbol{\beta}^*\|_1^2 + C' \cdot \sigma^2\right)^2}\right] + 2 \cdot \exp\left(-C'' \cdot \tau^2 + \log n + \log d\right).$$

Note that (I.71) is obtained by applying Lemmas J.4 and J.5 with $\phi(v) = \exp(u \cdot v)$. Since Lemmas J.4 and J.5 allow any increasing convex function $\phi(\cdot)$, similar results hold correspondingly. Hence, applying Panchenko's theorem in Lemma J.6 to (I.71), from (I.76) we have

$$\mathbb{P}\left[\left|\left\{T_n(\boldsymbol{\beta}^*) - \mathbb{E}_{\boldsymbol{\beta}^*}\left[T_n(\boldsymbol{\beta}^*)\right]\right\}_{j,k}\right| > v\right] \leq 2 \cdot e \cdot \exp\left[-\frac{C \cdot n \cdot v^2 \cdot \sigma^4}{\tau^4 \cdot \left(\tau^2 \cdot \|\boldsymbol{\beta}^*\|_1^2 + C' \cdot \sigma^2\right)^2}\right]$$

$$+ 2 \cdot e \cdot \exp\left(-C'' \cdot \tau^2 + \log n + \log d\right).$$

Furthermore, by union bound we have

$$\mathbb{P}\left[\left\|T_n(\boldsymbol{\beta}^*) - \mathbb{E}_{\boldsymbol{\beta}^*}\left[T_n(\boldsymbol{\beta}^*)\right]\right\|_{\infty,\infty} > v\right] \leq \sum_{j=1}^{d}\sum_{k=1}^{d}\mathbb{P}\left[\left|\left\{T_n(\boldsymbol{\beta}^*) - \mathbb{E}_{\boldsymbol{\beta}^*}\left[T_n(\boldsymbol{\beta}^*)\right]\right\}_{j,k}\right| > v\right]$$

$$\leq 2 \cdot e \cdot \exp\left[-\frac{C \cdot n \cdot v^2 \cdot \sigma^4}{\tau^4 \cdot \left(\tau^2 \cdot \|\boldsymbol{\beta}^*\|_1^2 + C' \cdot \sigma^2\right)^2} + 2 \cdot \log d\right]$$

$$+ 2 \cdot e \cdot \exp\left(-C'' \cdot \tau^2 + \log n + 3 \cdot \log d\right). \tag{I.77}$$

To ensure the right-hand side is upper bounded by $\delta$, we set the second term on the right-hand side of (I.77) to be $\delta/2$. Then we obtain

$$\tau = C \cdot \sqrt{\log n + 3 \cdot \log d + \log(4 \cdot e/\delta)}.$$

Let the first term on the right-hand side of (I.77) be upper bounded by $\delta/2$ and plugging in $\tau$, we then obtain

$$v = C \cdot \left[\log n + 3 \cdot \log d + \log(4 \cdot e/\delta)\right]$$

$$\cdot \left\{\left[\log n + 3 \cdot \log d + \log(4 \cdot e/\delta)\right] \cdot \|\boldsymbol{\beta}^*\|_1^2 + C' \cdot \sigma^2\right\}/\sigma^2 \cdot \sqrt{\frac{\log(4 \cdot e/\delta) + 2 \cdot \log d}{n}}.$$

Therefore, by setting $\delta = 4 \cdot e/d$ we have that

$$\left\|T_n(\boldsymbol{\beta}^*) - \mathbb{E}_{\boldsymbol{\beta}^*}\left[T_n(\boldsymbol{\beta}^*)\right]\right\|_{\infty,\infty} \leq v$$

$$= C \cdot \left(\log n + 4 \cdot \log d\right) \cdot \left[\left(\log n + 4 \cdot \log d\right) \cdot \|\boldsymbol{\beta}^*\|_1^2 + C' \cdot \sigma^2\right]/\sigma^2 \cdot \sqrt{\frac{\log d}{n}}$$

holds with probability at least $1 - 4 \cdot e/d$, which completes the proof of Lemma F.6. $\qquad\square$

### I.10 Proof of Lemma F.7

*Proof.* For any $j, k \in \{1, \ldots, d\}$, by the mean-value theorem we have

$$\left\|T_n(\boldsymbol{\beta}) - T_n(\boldsymbol{\beta}^*)\right\|_{\infty,\infty} = \max_{j,k\in\{1,\ldots,d\}}\left|\left[T_n(\boldsymbol{\beta})\right]_{j,k} - \left[T_n(\boldsymbol{\beta}^*)\right]_{j,k}\right| \tag{I.78}$$

$$= \max_{j,k\in\{1,\ldots,d\}}\left|(\boldsymbol{\beta} - \boldsymbol{\beta}^*)^\top \cdot \nabla\left[T_n(\boldsymbol{\beta}^\sharp)\right]_{j,k}\right|$$

$$\leq \|\boldsymbol{\beta} - \boldsymbol{\beta}^*\|_1 \cdot \max_{j,k\in\{1,\ldots,d\}}\left\|\nabla\left[T_n(\boldsymbol{\beta}^\sharp)\right]_{j,k}\right\|_\infty,$$

where $\boldsymbol{\beta}^\sharp$ is an intermediate value between $\boldsymbol{\beta}$ and $\boldsymbol{\beta}^*$. According to (I.67), (I.68) and the definition of $T_n(\cdot)$ in (2.4), by calculation we have

$$\nabla\left[T_n(\boldsymbol{\beta}^\sharp)\right]_{j,k} = \frac{1}{n}\sum_{i=1}^{n}\bar{\nu}_{\boldsymbol{\beta}^\sharp}(\mathbf{x}_i, y_i) \cdot y_i^3 \cdot x_{i,j} \cdot x_{i,k} \cdot \mathbf{x}_i,$$

where

$$\bar{\nu}_{\boldsymbol{\beta}}(\mathbf{x}, y) = \frac{8/\sigma^4}{\left[1 + \exp\left(-2 \cdot y \cdot \langle \boldsymbol{\beta}, \mathbf{x} \rangle / \sigma^2\right)\right] \cdot \left[1 + \exp\left(2 \cdot y \cdot \langle \boldsymbol{\beta}, \mathbf{x} \rangle / \sigma^2\right)\right]^2}$$
$$- \frac{8/\sigma^4}{\left[1 + \exp\left(-2 \cdot y \cdot \langle \boldsymbol{\beta}, \mathbf{x} \rangle / \sigma^2\right)\right]^2 \cdot \left[1 + \exp\left(2 \cdot y \cdot \langle \boldsymbol{\beta}, \mathbf{x} \rangle / \sigma^2\right)\right]}. \tag{I.79}$$

For notational simplicity, we define the following events

$$\mathcal{E} = \left\{\|\mathbf{x}_i\|_\infty \leq \tau, \text{ for all } i = 1, \ldots, n\right\}, \quad \text{and } \mathcal{E}' = \left\{|v_i| \leq \tau', \text{ for all } i = 1, \ldots, n\right\},$$

where $\tau > 0$ and $\tau' > 0$ will be specified later. By union bound we have

$$\mathbb{P}\left\{\left\|\nabla\left[T_n(\boldsymbol{\beta}^\sharp)\right]_{j,k}\right\|_\infty > v\right\} \leq d \cdot \mathbb{P}\left(\left\{\left|\nabla\left[T_n(\boldsymbol{\beta}^\sharp)\right]_{j,k}\right|\right\}_l > v\right)$$
$$= d \cdot \mathbb{P}\left[\left|\frac{1}{n}\sum_{i=1}^n \bar{\nu}_{\boldsymbol{\beta}^\sharp}(\mathbf{x}_i, y_i) \cdot y_i^3 \cdot x_{i,j} \cdot x_{i,k} \cdot x_{i,l}\right| > v\right]. \tag{I.80}$$

Let $\bar{\mathcal{E}}$ and $\bar{\mathcal{E}'}$ be the complement of $\mathcal{E}$ and $\mathcal{E}'$ respectively. On the right-hand side we have

$$\mathbb{P}\left[\left|\frac{1}{n}\sum_{i=1}^n \bar{\nu}_{\boldsymbol{\beta}^\sharp}(\mathbf{x}_i, y_i) \cdot y_i^3 \cdot x_{i,j} \cdot x_{i,k} \cdot x_{i,l}\right| > v\right] = \underbrace{\mathbb{P}\left[\left|\frac{1}{n}\sum_{i=1}^n \bar{\nu}_{\boldsymbol{\beta}^\sharp}(\mathbf{x}_i, y_i) \cdot y_i^3 \cdot x_{i,j} \cdot x_{i,k} \cdot x_{i,l}\right| > v, \mathcal{E}, \mathcal{E}'\right]}_{\text{(i)}}$$
$$+ \underbrace{\mathbb{P}(\bar{\mathcal{E}})}_{\text{(ii)}} + \underbrace{\mathbb{P}(\bar{\mathcal{E}'})}_{\text{(iii)}}. \tag{I.81}$$

**Analysis of Term (i):** For term (i) in (I.81), we have

$$\mathbb{P}\left[\left|\frac{1}{n}\sum_{i=1}^n \bar{\nu}_{\boldsymbol{\beta}^\sharp}(\mathbf{x}_i, y_i) \cdot y_i^3 \cdot x_{i,j} \cdot x_{i,k} \cdot x_{i,l}\right| > v, \mathcal{E}, \mathcal{E}'\right]$$
$$= \mathbb{P}\left[\left|\frac{1}{n}\sum_{i=1}^n \bar{\nu}_{\boldsymbol{\beta}^\sharp}(\mathbf{x}_i, y_i) \cdot y_i^3 \cdot x_{i,j} \cdot x_{i,k} \cdot x_{i,l} \cdot \mathbb{1}\left\{\|\mathbf{x}_i\|_\infty \leq \tau\right\} \cdot \mathbb{1}\left\{|v_i| \leq \tau'\right\}\right| > v, \mathcal{E}, \mathcal{E}'\right]$$
$$\leq \mathbb{P}\left[\left|\frac{1}{n}\sum_{i=1}^n \bar{\nu}_{\boldsymbol{\beta}^\sharp}(\mathbf{x}_i, y_i) \cdot y_i^3 \cdot x_{i,j} \cdot x_{i,k} \cdot x_{i,l} \cdot \mathbb{1}\left\{\|\mathbf{x}_i\|_\infty \leq \tau\right\} \cdot \mathbb{1}\left\{|v_i| \leq \tau'\right\}\right| > v\right].$$

To avoid confusion, note that $v_i$ is the noise in mixture of regression model, while $v$ appears in the tail bound. By applying union bound to the right-hand side of the above inequality, we have

$$\mathbb{P}\left[\left|\frac{1}{n}\sum_{i=1}^n \bar{\nu}_{\boldsymbol{\beta}^\sharp}(\mathbf{x}_i, y_i) \cdot y_i^3 \cdot x_{i,j} \cdot x_{i,k} \cdot x_{i,l}\right| > v, \mathcal{E}, \mathcal{E}'\right]$$
$$\leq \sum_{i=1}^n \mathbb{P}\left[\left|\bar{\nu}_{\boldsymbol{\beta}^\sharp}(\mathbf{x}_i, y_i) \cdot y_i^3 \cdot x_{i,j} \cdot x_{i,k} \cdot x_{i,l} \cdot \mathbb{1}\left\{\|\mathbf{x}_i\|_\infty \leq \tau\right\} \cdot \mathbb{1}\left\{|v_i| \leq \tau'\right\}\right| > v\right].$$

By (I.79) we have $\left|\bar{\nu}_{\boldsymbol{\beta}^\sharp}(\mathbf{x}_i, y_i)\right| \leq 16/\sigma^4$. Hence, we obtain

$$\mathbb{P}\left[\left|\bar{\nu}_{\boldsymbol{\beta}^\sharp}(\mathbf{x}_i, y_i) \cdot y_i^3 \cdot x_{i,j} \cdot x_{i,k} \cdot x_{i,l} \cdot \mathbb{1}\left\{\|\mathbf{x}_i\|_\infty \leq \tau\right\} \cdot \mathbb{1}\left\{|v_i| \leq \tau'\right\}\right| > v\right]$$
$$\leq \mathbb{P}\left[\left|y_i^3 \cdot x_{i,j} \cdot x_{i,k} \cdot x_{i,l} \cdot \mathbb{1}\left\{\|\mathbf{x}_i\|_\infty \leq \tau\right\} \cdot \mathbb{1}\left\{|v_i| \leq \tau'\right\}\right| > \sigma^4/16 \cdot v\right]. \tag{I.82}$$

Recall that in mixture of regression model we have $y_i = z_i \cdot \langle \boldsymbol{\beta}^*, \mathbf{x}_i \rangle + v_i$, where $z_i$ is a Rademacher random variable, $\mathbf{x}_i \sim N(0, \mathbf{I}_d)$ and $v_i \sim N(0, \sigma^2)$. Hence, we have

$$\left|y_i^3 \cdot \mathbb{1}\left\{\|\mathbf{x}_i\|_\infty \leq \tau\right\} \cdot \mathbb{1}\left\{|v_i| \leq \tau'\right\}\right| \leq \left(\left|z_i \cdot \langle \boldsymbol{\beta}^*, \mathbf{x}_i \rangle \cdot \mathbb{1}\left\{\|\mathbf{x}_i\|_\infty \leq \tau\right\}\right| + \left|v_i \cdot \mathbb{1}\left\{|v_i| \leq \tau'\right\}\right|\right)^3$$
$$\leq \left(\tau \cdot \|\boldsymbol{\beta}^*\|_1 + \tau'\right)^3,$$
$$\left|x_{i,j} \cdot x_{i,k} \cdot x_{i,l} \cdot \mathbb{1}\left\{\|\mathbf{x}_i\|_\infty \leq \tau\right\}\right| \leq \left|x_{i,j} \cdot \mathbb{1}\left\{\|\mathbf{x}_i\|_\infty \leq \tau\right\}\right|^3 \leq \tau^3.$$

Taking $v = 16 \cdot \left(\tau \cdot \|\boldsymbol{\beta}^*\|_1 + \tau'\right)^3 \cdot \tau^3/\sigma^4$, we have that the right-hand side of (I.82) is zero. Hence term (i) in (I.81) is zero.

**Analysis of Term (ii):** For term (ii) in (I.81), by union bound we have

$$\mathbb{P}(\bar{\mathcal{E}}) = \mathbb{P}\Big(\max_{i\in\{1,\dots,n\}} \|\mathbf{x}_i\|_\infty > \tau\Big) \leq n \cdot \mathbb{P}\big(\|\mathbf{x}_i\|_\infty > \tau\big) \leq n \cdot d \cdot \mathbb{P}\big(|x_{i,j}| > \tau\big).$$

Moreover, we have that $x_{i,j}$ is sub-Gaussian with $\|x_{i,j}\|_{\psi_2} = C$. Therefore, by Lemma 5.5 in [28] we have

$$\mathbb{P}(\bar{\mathcal{E}}) \leq n \cdot d \cdot \mathbb{P}\big(|x_{i,j}| > \tau\big) \leq n \cdot d \cdot 2 \cdot \exp\big(-C' \cdot \tau^2\big).$$

**Analysis of Term (iii):** Since $v_i$ is sub-Gaussian with $\|v_i\|_{\psi_2} = C \cdot \sigma$, by Lemma 5.5 in [28] and union bound, for term (iii) in (I.81) we have

$$\mathbb{P}(\bar{\mathcal{E}'}) \leq n \cdot \mathbb{P}\big(|v_i| > \tau'\big) \leq n \cdot 2 \cdot \exp\big(-C' \cdot \tau'^2/\sigma^2\big).$$

To ensure the right-hand side of (I.81) is upper bounded by $\delta$, we set $\tau$ and $\tau'$ to be

$$\tau = C \cdot \sqrt{\log d + \log n + \log(4/\delta)}, \quad \text{and} \quad \tau' = C' \cdot \sigma \cdot \sqrt{\log n + \log(4/\delta)} \tag{I.83}$$

to ensure terms (ii) and (iii) are upper bounded by $\delta/2$ correspondingly. Finally, by (I.80), (I.81) and union bound we have

$$\mathbb{P}\Big\{\max_{j,k\in\{1,\dots,d\}} \big\|\nabla[T_n(\boldsymbol{\beta}^\sharp)]_{j,k}\big\|_\infty > v\Big\}$$

$$\leq d^2 \cdot d \cdot \mathbb{P}\bigg[\bigg|\frac{1}{n}\sum_{i=1}^n \bar{\nu}_{\boldsymbol{\beta}^\sharp}(\mathbf{x}_i, y_i) \cdot y_i^3 \cdot x_{i,j} \cdot x_{i,k} \cdot x_{i,l}\bigg| > v\bigg] \leq d^3 \cdot \delta$$

for $v = 16 \cdot \big(\tau \cdot \|\boldsymbol{\beta}^*\|_1 + \tau'\big)^3 \cdot \tau^3/\sigma^4$ with $\tau$ and $\tau'$ specified in (I.83). Then by setting $\delta = 4 \cdot d^{-4}$ and plugging it into (I.83), we have

$$v = 16 \cdot \Big[C \cdot \sqrt{5 \cdot \log d + \log n} \cdot \|\boldsymbol{\beta}^*\|_1 + C' \cdot \sigma \cdot \sqrt{4 \cdot \log d + \log n}\Big]^3 \cdot \Big[C \cdot \sqrt{5 \cdot \log d + \log n}\Big]^3 \Big/ \sigma^4$$

$$\leq C'' \cdot \big(\|\boldsymbol{\beta}^*\|_1 + C''' \cdot \sigma\big)^3 \cdot \big(5 \cdot \log d + \log n\big)^3,$$

which together with (I.78) concludes the proof of Lemma F.7. □

# J Auxiliary Results

In this section, we lay out several auxiliary lemmas. Lemmas J.1-J.3 provide useful properties of sub-Gaussian random variables. Lemmas J.4 and J.5 establish the symmetrization and contraction results. Lemma J.6 is Panchenko's theorem. For more details of these results, see [6, 28].

**Lemma J.1.** Let $Z_1, \dots, Z_k$ be the $k$ independent zero-mean sub-Gaussian random variables, for $Z = \sum_{j=1}^k Z_j$ we have $\|Z\|_{\psi_2}^2 \leq C \cdot \sum_{j=1}^k \|Z_j\|_{\psi_2}^2$, where $C > 0$ is an constant.

**Lemma J.2.** For $Z_1$ and $Z_2$ being two sub-Gaussian random variables, $Z_1 \cdot Z_2$ is a sub-exponential random variable with

$$\|Z_1 \cdot Z_2\|_{\psi_1} \leq C \cdot \max\big\{\|Z_1\|_{\psi_2}^2, \|Z_2\|_{\psi_2}^2\big\},$$

where $C > 0$ is an constant.

**Lemma J.3.** For $Z$ being sub-Gaussian or sub-exponential, it holds that $\|Z - \mathbb{E}Z\|_{\psi_2} \leq 2 \cdot \|Z\|_{\psi_2}$ or $\|Z - \mathbb{E}Z\|_{\psi_1} \leq 2 \cdot \|Z\|_{\psi_1}$ correspondingly.

**Lemma J.4.** Let $\mathbf{z}_1, \dots, \mathbf{z}_n$ be the $n$ independent realizations of the random vector $\boldsymbol{Z} \in \mathcal{Z}$ and $\mathcal{F}$ be a function class defined on $\mathcal{Z}$. For any increasing convex function $\phi(\cdot)$ we have

$$\mathbb{E}\bigg\{\phi\bigg[\sup_{f\in\mathcal{F}}\bigg|\sum_{i=1}^n f(\mathbf{z}_i) - \mathbb{E}\boldsymbol{Z}\bigg|\bigg]\bigg\} \leq \mathbb{E}\bigg\{\phi\bigg[\sup_{f\in\mathcal{F}}\bigg|\sum_{i=1}^n \xi_i \cdot f(\mathbf{z}_i)\bigg|\bigg]\bigg\},$$

where $\xi_1, \dots, \xi_n$ are i.i.d. Rademacher random variables that are independent of $\mathbf{z}_1, \dots, \mathbf{z}_n$.

**Lemma J.5.** Let $\mathbf{z}_1, \dots, \mathbf{z}_n$ be the $n$ independent realizations of the random vector $\boldsymbol{Z} \in \mathcal{Z}$ and $\mathcal{F}$ be a function class defined on $\mathcal{Z}$. We consider the Lipschitz functions $\psi_i(\cdot)$ $(i = 1, \dots, n)$ that satisfy

$$\big|\psi_i(v) - \psi_i(v')\big| \leq L \cdot |v - v'|, \quad \text{for all } v, v' \in \mathbb{R},$$

and $\psi_i(0) = 0$. For any increasing convex function $\phi(\cdot)$ we have

$$\mathbb{E}\bigg\{\phi\bigg[\bigg|\sup_{f\in\mathcal{F}}\sum_{i=1}^n \xi_i \cdot \psi_i\big[f(\mathbf{z}_i)\big]\bigg|\bigg]\bigg\} \leq \mathbb{E}\bigg\{\phi\bigg[2 \cdot \bigg|L \cdot \sup_{f\in\mathcal{F}}\sum_{i=1}^n \xi_i \cdot f(\mathbf{z}_i)\bigg|\bigg]\bigg\},$$

where $\xi_1, \dots, \xi_n$ are i.i.d. Rademacher random variables that are independent of $\mathbf{z}_1, \dots, \mathbf{z}_n$.

**Lemma J.6.** Suppose that $Z_1$ and $Z_2$ are two random variables that satisfy $\mathbb{E}\big[\phi(Z_2)\big] \leq \mathbb{E}\big[\phi(Z_1)\big]$ for any increasing convex function $\phi(\cdot)$. Assuming that $\mathbb{P}(Z_1 \geq v) \leq C \cdot \exp(-C' \cdot v^\alpha)$ $(\alpha \geq 1)$ holds for all $v \geq 0$, we have $\mathbb{P}(Z_2 \geq v) \leq C \cdot \exp(1 - C' \cdot v^\alpha)$.