[Reviews · NeurIPS 2015]

Submitted by Assigned_Reviewer_1

The paper studies EM in a high-dimensional setting. Convergence results are established and inference procedure developed.

Authors should discuss under what conditions can we find the initial solution, \beta^{\rm init} that satisfies the requirement in line 60 on page 2.

There are two algorithms proposed. Is there a practical guidance on which one to use?

How does one choose $\hat s$ and $\lamdba$ in practice? How sensitive are results on the choice of these tuning parameters? Is there a way to choose them in a data dependent way? In the existing literature on high-dimensional inference, choice of the tuning parameters is of utmost importance to obtaining good sampling properties.

The decorrelated score test is the same as in [*]. This fact should be cited and not swept under the rug. Is there anything new that needs to be developed in order to apply the existing inference procedure in the current context?

The main theoretical results hold under a large number of technical conditions. The authors should discuss whether these condition/assumptions are satisfied for the considered latent variable models.

[*] A General Theory of Hypothesis Tests and Confidence Regions for Sparse High Dimensional Models. Yang Ning, Han Liu
Summary: The paper extends [2] to a high-dimensional setting. It also applies the work of Ning and Liu [*] to the problem of inference. The paper combines existing results in a clever way. However, it should be made more clear what is exactly novel in this paper and why is application of existing tools nontrivial. The submitted paper is a "trailer" for the long supplementary material. This trailer should contain more information on the technical challenges required to establish the main results.

[*] A General Theory of Hypothesis Tests and Confidence Regions for Sparse High Dimensional Models. Yang Ning, Han Liu

Submitted by Assigned_Reviewer_2

In this paper, the authors propose a modification of the EM algorithm for the case that the true parameter is sparse and high-dimensional. The authors show some conditions for its convergence to the global solution and derive the rate of convergence. Also, a statistical test framework is constructed, which enables to check the sparsity of the estimator.

The paper is well organized. The related work is sufficiently summarized. However, the contents are very dense and some parts (mainly, section 3 and 4) are hard to follow. More simplification would be helpful for readers.

- Although in line 066 the authors argue that sqrt(s*log(d/n)) is minimax-optimal, I couldn't find the proof of this from the main manuscript (I didn't read the supplementary material.) Can you prove this clearly?

- In Eq.(3.8), ^s depends on s*, but we don't know s* in real situations. How should ^s be chosen in practice? Also, how can we find an appropriate initial value beta^init satisfying ||beta^init - beta*||_2 being less than R/2 (see line 275) in practice?

- I didn't understand why the truncation step in Algorithm 1 is important. Can you explain what does happen if we skip the truncation step in your EM algorithm?

****** After rebuttal ******

Thank you for your response. It was satisfactory for me and I increased my score from 6 to 7.
Summary: A solid stats paper, but not easy to read for a ML

researcher. Also, the usefulness of the proposed method for real

applications is somewhat questionable.

Submitted by Assigned_Reviewer_3

A sparse EM procedure is proposed which modifies classical EM by truncating the parameter vector at each step to a desired sparsity level. Also proposed is an asymptotically normal statistic based on a decorrelated score for testing whether a specific entry of the parameter vector is non-zero.

The guarantee provided for the sparse EM states that the estimate converges geometrically to the true value (in the number of iterations), up to a statistical error term which is like the square root of the sparsity level and the infinity norm of the statistical error in the M-step, which is typically only logarithmic in the ambient dimension.

This follows on the heels of recent similar results for EM in the low-dimensional setting. As in the previous work, it is assumed that the Q function is sufficiently smooth and nearly quadratic in a certain basin of attraction, and that the algorithm is initialized in the same basin. The new condition is the infinity norm control on the error in estimating M. This condition is required only for sparse vectors, but for many models estimating M at all possible vectors is just as easy as at sparse vectors (this holds for the Gaussian mixture example considered, and for the non-gradient version of the mixture of regression model). This raises a question -- is the truncated EM algorithm necessary? Could a similar guarantee be obtained, for instance, by performing ordinary EM under the same conditions, followed by a single truncation step? EDIT: the authors' observation that linear regression is a special case here seems to be a convincing argument against a single truncation step.

There may be some related work in sparse optimization that perhaps should be discussed (like "Sparse online learning via truncated gradient" Langford et al 2009, just as an example).

The experimental section should state what initialization was used.
Summary: A sparse EM variant is proposed for parameter estimation in high dimensional settings. The convergence guarantees provided are an important result for high dimensional learning, although the importance of the sparse variant of EM to the obtained results is not certain.

Author Feedback
Author rebuttal: We thank all the reviewers for their valuable comments and suggestions. We will revise accordingly in the final version.

Response to Reviewer_1

The conditions on the initial solution can be satisfied in two ways: (i) Do multiple random initializations and use the initial solution leading to largest Q function. This is a heuristic that practitioners often do for EM algorithm; (ii) Use result of tensor methods, e.g., [9], as an initial solution, for which we can obtain rigorous theoretical guarantees.

We would like to suggest the use of gradient ascent implementation for two reasons: (i) For complex models, it can be hard to derive the closed form of the exact M-step; (ii) For high dimensional models, the exact M-step is not always well defined.

Thanks for pointing out [*] and we will cite it. (In this version we didn't cite [*] because when we finish our paper in 2014 [*] is still in progress.) It is nontrivial to apply the idea of [*] to EM due to both the lack of convexity and the existence of latent variables, which make the approximation of Fisher information and score function particularly difficult. To resolve this issue, our key observation is Theorem 2.1, which connects the gradient and generalized Hessian of Q function with score function and Fisher information. It is actually quite interesting that the Hessian of Q needs to be corrected in order to approximate the Fisher information.

The technical conditions are presented to simplify the presentation. We have verified all the conditions for each model in Section E due to space limits.

Response to Reviewer_2

We will add more discussion and simplify the presentation in the final version. The minimax lower bound of estimating sparse vectors is \sqrt{s^* log d/n} and can been proved following similar arguments in [**]. We will add it to the final version.

[**] Garvesh, Wainwright, Yu (2009). Minimax rates of estimation for high-dimensional linear regression over Lq-balls.

In our paper, \hat{s} is a parameter larger than s^*. In practice, we can use AIC/BIC/cross validation to choose \hat{s}. The initial solution condition can be satisfied by either multiple random initialization, or using the solution of tensor methods as the initial solution.

The truncation step preserves the top \hat{s} coordinates of beta^(t + 0.5), by which we enforce the sparsity of beta^t. If the truncation is skipped, the noise and error accumulate across all coordinates, which leads to a \sqrt{d/n} rate, resulting in high estimation error when d >> n.

Response to Reviewer_3

It is unlikely that a single truncation step would succeed. Consider a degenerate setting of mixed regression where there is only one cluster. Then it reduces to linear regression. In this case iterative thresholding succeeds while a single thresholding step generally fails (unless the design is identity). Similar arguments apply to our setting.

Thanks for pointing out Langford et al. 2009. We will cite it with discussion in the final version. In the experiment, we use random initializations. We will add more details.

Response to Assigned_Reviewer_4

Thank you for the supportive comments.

We have also discussed Mixture of Regression Model, though the details are deferred to appendix due to space limitation.

Response to Assigned_Reviewer_5

The reviewer's main concern appears to be that the appendix is too long. We thank the reviewer for this important comment. However, the appendix is 35 pages rather than 80 pages. We would like to point out that NIPS papers often have long proofs in the appendix and our main text is self-contained.

Response to Assigned_Reviewer_6

We thank the reviewer for the affirmative comments.